# RAViG-Bench: A Benchmark for Retrieval-Augmented Visually-rich Generation with Multi-modal Automated Evaluation

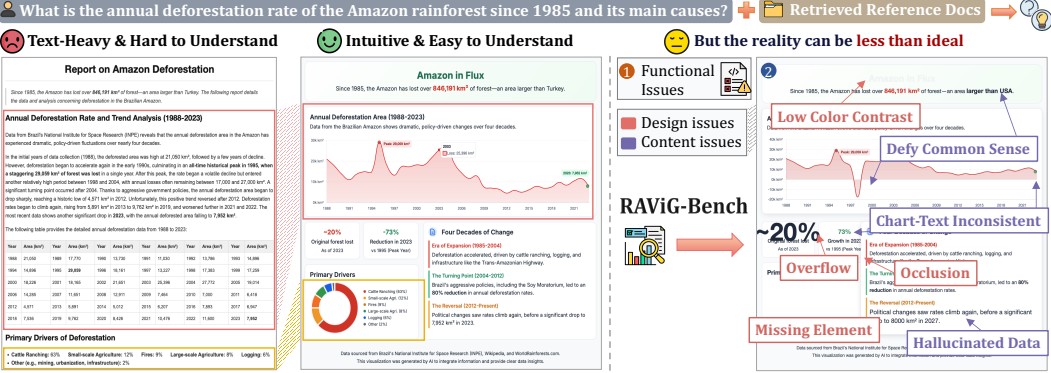

Figure 1: Compared with traditional text-heavy answers (left), RAViG responses (middle) are more intuitive and user-friendly for queries that entail presenting substantial data. However, such outputs may still suffer from various issues in practice (right), which our benchmark is designed to detect.

## ABSTRACT

Retrieval-Augmented Visually-rich Generation (RAViG) extends RAG by integrating textual explanations with multiple visual elements in a well-structured layout. Despite its growing adoption, no existing benchmark offers a holistic evaluation of RAViG. Current RAG benchmarks focus on text-only generation, while natural language to visualization (NL2VIS) benchmarks focus on "show-data-as-chart" style queries and do not follow the RAG paradigm. To address this deficiency, we present RAViG-Bench, the first comprehensive benchmark specifically designed for RAViG. The benchmark features a diverse collection of authentic user queries, each paired with real-world web retrievals to simulate realistic RAViG scenarios. Besides, we introduce a novel multi-modal automated evaluation framework that holistically assesses the quality of RAViG outputs. This framework scrutinizes the generated content by evaluating the functionality, design quality, and content quality of both textual and visual components. Our extensive experiments on leading commercial and open-source LLMs provide a comprehensive analysis of their current capabilities, highlighting significant limitations and charting key directions for future research in this emergent area.

## 1 INTRODUCTION

Recently, Retrieval-Augmented Generation (RAG) that produces outputs comprising illustrative visual elements and explanatory text has seen widespread adoption within enterprise Large Language Models (LLMs) (Google, 2025; Shanghai Mita Network Technology Co., Ltd., 2025; Perplexity AI, Inc., 2025). This trend is driven by the increasing demand for statistical analysis in fields such as meteorology, finance, and public health. For instance, when queried for an analysis of the annual deforestation rate of the Amazon rainforest, a capable model retrieves relevant weather information and constructs a visually rich report. As depicted in the middle of Fig. 1, such a report features a line chart depicting deforestation area over time and a pie chart for sales distribution, with explanatory text embedded within the information panel and all elements organized into a user-friendly layout. Consequently, we term this paradigm, which integrates illustrative visual elements with explanatory text into the generation process, as Retrieval-Augmented Visually-rich Generation (RAViG).

| Benchmark | Knowledge Source | Output Modality | Design Eval | | Content Eval | | GT-Free Eval | Real-World Fidelity |
|---|---|---|---|---|---|---|---|---|
| | | | Chart | Chart-Text | Text | Chart-Text | | |
| nvBench | In-Query Data | Single Chart | ✓ | ✗ | ✗ | ✗ | ✗ | ✗ |
| VISEval | In-Query Data | Single Chart | ✓ | ✗ | ✗ | ✗ | ✗ | ✗ |
| RAGAs | Web Retrieval | Plain Text | ✗ | ✗ | ✓ | ✗ | ✓ | ✗ |
| CRAG | Web Retrieval | Plain Text | ✗ | ✗ | ✓ | ✗ | ✗ | ✗ |
| GaRAGe | Web Retrieval | Plain Text | ✗ | ✗ | ✓ | ✗ | ✗ | ✗ |
| **RAViG-Bench** | Web Retrieval | Rich Content | ✓ | ✓ | ✓ | ✓ | ✓ | ✓ |

Table 1: **Comparison of RAViG-Bench with existing evaluation benchmarks.**

However, a significant gap exists in the literature: the lack of a dedicated benchmark for the evaluation of RAViG. As shown in Tab. 1, prevailing benchmarks are inadequate, since they either focus on text-only RAG or on the "show-data-as-chart" question-answering paradigm. Existing RAG benchmarks lack the necessary data to support visually-rich generation and the multi-modal metrics required to evaluate the interplay between visual and textual contents. Moreover, their query data, often derived from knowledge graphs or LLM generations, deviates significantly from real-world distributions (Yang et al., 2024a; ES et al., 2024; Sorodoc et al., 2025). On the other hand, benchmarks for natural language to visualization (NL2VIS) primarily evaluate the visual fidelity of generated charts, but they operate under a different premise where chart data is provided alongside the natural language query (Chen et al., 2025; Luo et al., 2025). In such settings, the data for visualization is pre-defined, reducing the task to one of direct data-to-chart translation. Furthermore, these benchmarks typically focus on single-chart outputs, failing to address the composite nature of RAViG, which involves an integrated presentation of both text and multiple visualizations.

The need for a dedicated RAViG benchmark is further underscored by several significant and recurrent challenges observed in existing LLMs. 1) Functional Failures: We choose HTML for visually-rich generation due to its widespread adoption, cross-platform compatibility, and high extensibility for integrating multi-modal content. However, LLMs sometimes produce syntactically incorrect or incomplete HTML, preventing the render of a visually-rich webpage. 2) Design Deficiencies: The generated visual elements often violate fundamental principles of user interface design, thereby impairing readability and user acquisition of information. As depicted in Fig. 1, common issues include low color contrast that renders text or icons illegible, text overflow, and empty content boxes to element rendering failure. 3) Content Hallucination: LLMs are prone to hallucination, generating content that is factually incorrect or inconsistent. As shown in Fig. 1, a generated line chart might contradict the accompanying textual summary, or data presented in tables may be fabricated.

Motivated by these observations, we posit the necessity of a benchmark designed for RAViG. Such a benchmark should feature diverse, real-world queries that genuinely necessitate visually-rich answers. Furthermore, it requires a robust multi-modal evaluation mechanism capable of assessing not only the individual quality of visual and textual components but also their interaction.

To this end, we introduce the first dataset for Retrieval-Augmented Visually-rich Generation. Our dataset is characterized by three key attributes: 1) Realism. The majority of our queries are drawn from the NQ dataset (Kwiatkowski et al., 2019), which comprises real user questions to the Google Search engine. An automated selection strategy is employed to identify queries that are genuinely suitable for visually-rich responses. 2) Diversity. The dataset spans six distinct query categories derived from a wide range of real-world scenarios. To mitigate the inherent imbalance in the distribution of real-world queries, we augment the dataset with synthetic examples, preserving the original distribution while ensuring an adequate amount for each category. 3) Curation. For each query, we collect the top-10 web search results from a real search engine, retaining the inherent noise of realistic retrieval. Subsequently, human annotators confirm the suitability of each query for visually-rich generation by jointly considering the query type and the content of its references.

Second, we propose a novel, automatic multi-modal evaluation framework that assesses the generated visually-rich answers along three critical dimensions: 1) Functionality Validation. We employ syntax checkers to verify whether the generated output is well-formed and can be rendered without errors. 2) Design Quality Assessment. Drawing from an analysis of common failure cases, we identify five prevalent design issues that hinder information acquisition. We develop a hybrid evaluation paradigm that combines an "MLLM-as-a-Judge" approach with rule-based methods to automatically detect these visual defects. 3) Content Quality Evaluation. We extend traditional RAG evaluation

metrics from text-only generation to holistically assess both textual and visual content across three dimensions: Reasonableness, Comprehensiveness, and Faithfulness. For this coordinated assessment, we employ an LLM-based judge assisted by a "claim extraction and verification" strategy, allowing fine-grained and accurate evaluation.

To summarize, our contributions are summarized as follows:

- We propose the first dataset for Retrieval-Augmented Visually-rich Generation (RAViG), featuring real-world queries, diverse categories, and web-sourced retrievals to simulate realistic scenarios.

- We present a comprehensive, automated multi-modal evaluation framework for RAViG that assesses the interplay of textual and visual elements across functionality, design, and content quality.

- We benchmark leading commercial and open-source LLMs, providing a comprehensive analysis that reveals their current limitations and outlines key directions for future research.

## 2 RELATED WORK

Existing RAG benchmarks mainly evaluate textual outputs, measuring aspects such as accuracy, faithfulness, and hallucination (Gan et al., 2025; Sorodoc et al., 2025; Yang et al., 2024a; Saad-Falcon et al., 2024; Liu et al., 2023; Niu et al., 2024; Katsis et al., 2025; ES et al., 2024; Chen et al., 2024), with some extending to multi-modal understanding (Zhan et al., 2025; Strich et al., 2025; Masry et al., 2022; Dong et al., 2025; Ma et al., 2024; Ding et al., 2024; Wu et al., 2025) or code generation (Liu et al., 2024b; Wang et al., 2025). Among these, MRAMG-Bench (Yu et al., 2025) is most similar to our setting, but it is essentially a text-only generation task, as its answers combine model-generated text with retrieval-sourced images. Existing benchmarks, however, rarely assess an LLM's ability to autonomously design and generate visually rich responses that integrate retrieved knowledge. In parallel, NL2VIS benchmarks focus on converting structured queries into single chart from predefined data (Cheng et al., 2023; Vázquez, 2024; Li et al., 2024; Luo et al., 2021a; 2025; Chen et al., 2025; Podo et al., 2024; Pan et al., 2025; Song et al., 2024; Joseph et al., 2025), limiting the model to direct data-to-chart translation. Our setting is more open-ended and realistic: given a query, the model must decide when and how to combine multiple visualizations with text into a coherent answer. To our knowledge, no existing benchmark performs a comprehensive, end-to-end evaluation of this capability. See Appendix A for a detailed discussion.

## 3 RAViG-BENCH DATASET

The RAViG-Bench dataset aims to provide a challenging and realistic benchmark for retrieval-augmented visually-rich generation. It is constructed through a four-stage pipeline: 1) selecting queries from a real-world dataset, 2) synthesizing data to address underrepresented categories, 3) creating reference documents from a real search engine, and 4) conducting a final human review (Sec. 3.1). We also perform detailed statistical analyses to further characterize the dataset (Sec. 3.2).

### 3.1 DATASET CONSTRUCTION

**Automated Query Selection.** To better capture real user needs, we derived tasks from the Natural Questions (NQ) dataset (Kwiatkowski et al., 2019), which contains over 300,000 genuine Google search queries and serves as a high-quality real-world data resource. We removed non-English queries and then employed Gemini-2.5-pro (Comanici et al., 2025) with a detailed prompt to identify queries suitable for visually-rich responses. The prompt leverages query keywords, type, and semantic intent to apply explicit selection criteria, identifying queries that request for quantitative or statistical data (e.g., population, price), trend or time-series analysis, quantitative comparisons (e.g., market share), or proportions and compositions. A query is suitable only when the data contains explicit numerical attributes and its visual form would aid comprehension. For each query, the model outputs a binary suitability decision with justification.

**Automated Query Synthesis.** We classified all queries into six categories reflecting common data visualization intents: trend (changes over time), proportion (part-to-whole relationships), comparison (contrasting distinct items), distribution (how data spreads over a range), ranking (ordering by a metric), and details (retrieving raw or specific information). Detailed descriptions and examples are provided in Appendix E. To mitigate the highly skewed distribution of real-world queries, we augmented each type with an equal number of synthetic queries, ensuring adequate samples per category without substantially altering the original distribution. As LLMs are now widely employed for data synthesis (Yuan et al., 2021; Chen et al., 2022), we employed Gemini-2.5-pro to generate

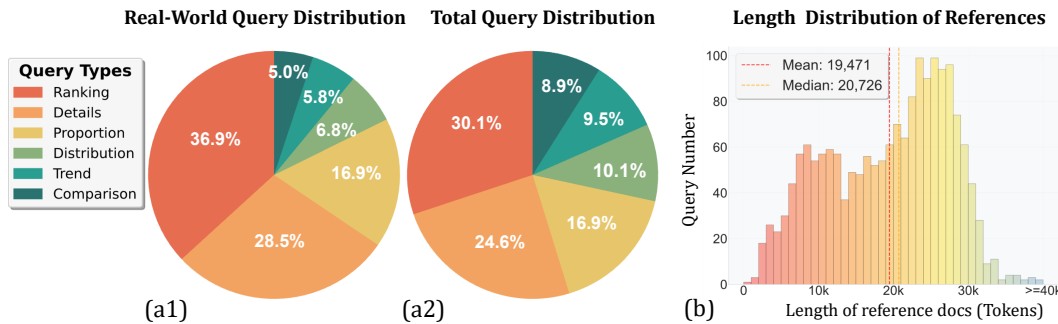

Figure 2: **Dataset statistics.** (a1&a2) Distribution of query categories. (b) Length distribution of reference documents.

queries that met the aforementioned criteria. These automated stages yielded an initial query set that captures real-world scenarios and is inherently suitable for visualization.

**Reference Documents Retrieval.** To simulate realistic RAG scenarios, we paired each query with the contents of the top 10 web pages returned by the Google Search API for response generation. Apart from manually removing content that is overtly biased or harmful, all other documents are retained without filtering, preserving the natural noise distribution in real search environments. This setup creates a noisy yet structured information environment that challenges models to perform cross-document synthesis and selective integration. Each document is represented in a standardized key–value schema containing fields such as page title, synopsis, content, and additional metadata, as detailed in Tab. 4 in Appendix B. This multi-granularity representation provides rich contextual information to support model reasoning.

**Final Human Review.** The suitability of producing a visually enriched answer depends not only on the query type but also on the content of its reference documents. For example, when asked "What are the global smartphone sales trends over the past five years?", purely descriptive news would warrant a textual summary, whereas aggregated time-series sales data would justify a line chart. Therefore, to refine dataset quality, we first used Gemini-2.5-pro to summarize the quantitative content within each query's references, after which human annotators validated query suitability by jointly considering the query and the content of its references. This process yielded 1,798 queries for the RAViG-Bench dataset, including 1,198 real-world queries and 600 synthetic queries.

## 3.2 DATASET STATISTIC

**Query Category Distribution.** We analyzed the distribution of queries across six categories. Real-world queries from the NQ dataset follow the original distribution shown in Fig. 2(a1). After augmentation with synthetic queries, the final distribution is presented in Fig. 2(a2). This process retains the key characteristics of the real-world data, ensuring the realistic fidelity of RAViG-Bench.

**Reference Document Characteristics.** As the primary information source for answer generation, the length of reference documents directly influences task difficulty by determining both the amount of information a model must process and the size of the required context window. As shown in Fig. 2(b), lengths range from a few thousand to over 40k tokens, presenting a broad spectrum of context scales. This diversity not only affects computational requirements, but also brings varying levels of information density, challenging models to adapt summarization and reasoning strategies when handling both concise and highly verbose documents.

## 4 RAViG-BENCH EVALUATION FRAMEWORK

We propose an automated, multi-stage, and multi-modal framework to evaluate LLMs on RAViG. First, it validates functionality to ensure appropriate model responses and complete, well-formed HTML (Sec. 4.1). Passed responses are then assessed for design quality (Sec. 4.2), checking for visual issues that might impair readability. Finally, efficient responses that pass the first two stages are scored for content quality (Sec. 4.3). We then report the agreement between our evaluation results and human expert judgments, demonstrating the effectiveness of our framework (Sec. 4.4).

Figure 3: **RAViG-Bench Overview.** Given analytical queries and retrieved references (left), LLMs generates visually-rich responses (middle), which we then evaluate in three stages (right): functionality and design checks filter out invalid or flawed responses, while the content stage assigns a quality score to the remaining well-designed responses.

## 4.1 FUNCTIONALITY VALIDATION

We developed a multi-step process to assess the functionality of LLMs. For each query, we first check responsiveness, requiring the model to produce an output within at most five attempts. Next, we verify the instruction adherence by ensuring the output contains only HTML code without extraneous conversational text. We then validate the structural integrity of the HTML output using the lxml library to detect issues such as mismatched tags, truncation, or missing DOCTYPE declarations. Code that passes these checks is rendered in a sandbox environment simulating a real-world, web-based QA scenario of an LLM via the Chromium engine with an 800-pixel-wide fixed viewport. An output is deemed functionally valid only if it renders successfully; in such cases, we also capture a screenshot of the rendered result as visual evidence for subsequent design quality assessment.

## 4.2 DESIGN QUALITY ASSESSMENT

Based on extensive case analysis, we classify the visual defects that may impede user acquisition of information into five types (Fig. 4). Inspired by related "MLLM-as-a-Judge" work (Chen et al., 2025), we employ GPT-4o to detect missing elements and occlusion. For oversized elements, overflow, and low color contrast, where LLMs are prone to significant hallucination, we adopt rule-based evaluations. A design is deemed "passed" only if all five defects are absent.

**Oversized Element** occurs when components such as charts or icons are rendered at an excessively large scale (Fig. 4(a)), disrupting layout and harming usability. For example, an oversized chart may require excessive vertical scrolling, while an oversized icon can visually unbalance the page. Detection is performed in a Playwright sandbox, programmatically measuring rendered component size. A chart is flagged if its rendered height exceeds twice the viewport height; an icon is flagged if its rendered size exceeds the code-specified dimensions, or if no size is specified, which causes rendering at a large intrinsic size.

**Missing Element** occurs when elements defined in the HTML fail to render and remain invisible to the user (Fig. 4(b)), typically due to layout or formatting errors. Detection is performed by providing GPT-4o with both the HTML code and its rendered screenshot, prompting it to compare the two to identify elements present in code but absent from the screenshot. Furthermore, we include few-shot instances in the prompt to help the model better understand the task.

**Occlusion** occurs when overlapping elements hinder readability (Fig. 4(c)). When evaluating an entire page screenshot directly, the model's attention can be diluted, causing it to miss localized or minor overlaps (Fig. 4(c), top). To mitigate this, we adopt a divide-and-conquer strategy: HTML <h1>/<h2>tags are used to segment the page into semantic modules. Screenshots of modules from the same response are then processed as a group, with GPT-4o examining each module individually for occlusion. A response is classified as defective if any module contains an occlusion issue. This modular approach improves detection accuracy by focusing the model on smaller, localized regions. Few-shot examples are also included in the prompt.

**Overflow** occurs when a container's content exceeds its boundaries, resulting in truncation or occlusion of adjacent components (Fig. 4(d)). Unlike perceptual issues, detecting overflow requires

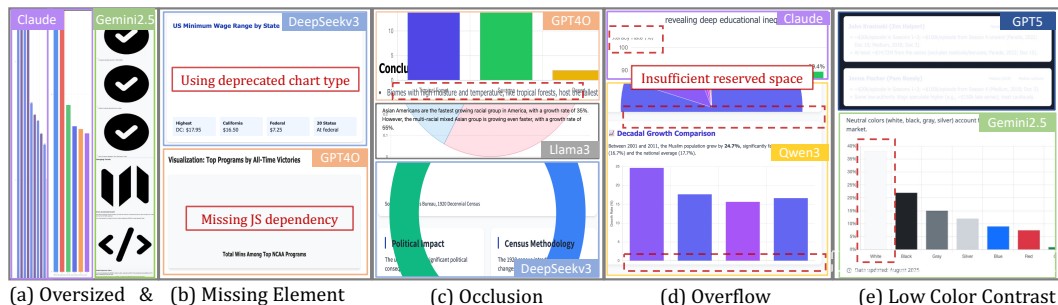

| (a) Oversized & | (b) Missing Element | (c) Occlusion | (d) Overflow | (e) Low Color Contrast |

Figure 4: Examples of five primary types of **design defects** (Please zoom in for details).

precise geometric analysis, which vision models often struggle with due to limited fine-grained spatial reasoning. We therefore developed a rule-based algorithm. The page is first segmented into visual modules by combining CSS-based structural analysis (for explicitly defined components such as cards or panels) with *heuristic analysis* that identifies distinct visual modules via unique styling cues (e.g., backgrounds, borders, shadows). Within each module, the algorithm checks whether any child element's geometry extends beyond its parent's bounds. To exclude deliberate designs, three cases are filtered out: 1) invisible elements, 2) decorative elements (e.g., with aria-hidden), and 3) intentional overflows within scrollable containers. This ensures that only genuine, unhandled overflows are flagged, yielding highly accurate detection.

**Low Color Contrast** occurs when text or chart elements are insufficiently distinguishable from their background (Fig 4(e)). Accurate assessment demands precise, quantitative color analysis, a task prone to LLM hallucinations. We therefore adopt a rule-based method grounded in the Web Content Accessibility Guidelines (WCAG 2.1). The method first determines effective foreground and background colors. *For text*, the background color is computed via recursive alpha-blending of parent colors up the DOM tree until a fully opaque value is obtained or the root node is reached, correctly handling layered and semi-transparent containers. For gradients, the algorithm derives a representative average from extracted color stops. *For charts*, a hybrid approach is used: 1) retrieving color configurations from charting library APIs (e.g., ECharts, Chart.js) when available; 2) sampling pixels from the borders of localized chart screenshots to infer an equivalent background color, robust to non-uniform patterns. Once the colors are identified, we calculate the contrast ratio (CR) following the WCAG standard, which involves linearizing sRGB values (Eq. 1):

$$C_{\text{linear}} = \begin{cases} \frac{C_{\text{sRGB}}}{12.92} & \text{if } C_{\text{sRGB}} \leq 0.03928 \\ \left(\frac{C_{\text{sRGB}}+0.055}{1.055}\right)^{2.4} & \text{if } C_{\text{sRGB}} > 0.03928, \end{cases} \tag{1}$$

computing relative luminance (Eq. 2):

$$L = 0.2126 \cdot R_{\text{linear}} + 0.7152 \cdot G_{\text{linear}} + 0.0722 \cdot B_{\text{linear}}, \tag{2}$$

and finding the final ratio of the lighter color ($L_1$) to the darker ($L_2$) via $CR = \frac{(L_1+0.05)}{(L_2+0.05)}$. However, standard WCAG thresholds proved too strict for our setting. To better align with human perception, we manually annotated 2,000 samples and statistically derived thresholds from issue cases: $\theta_{\text{text}} = 1.5$ and $\theta_{\text{chart}} = 1.05$. A defect is flagged when $CR < \theta$.

### 4.3 CONTENT QUALITY EVALUATION

We design a multi-dimensional automated framework to jointly assess the quality of textual content and the information conveyed by visualizations, as well as their alignment. This addresses the limitation of prior approaches that focused solely on text. We first parse the generated HTML with BeautifulSoup to isolate core content by removing irrelevant elements (e.g., CSS, JavaScript), thereby focusing the LLM evaluator on substantive information. Then, we employ Gemini-2.5-pro under the "LLM-as-a-Judge" paradigm to assess the parsed content across three dimensions. For *Faithfulness*, we further apply a "claim extraction and verification" strategy, in which key claims are first extracted and then evaluated individually to improve accuracy and granularity.

**Reasonableness** evaluates whether a response contains factual errors, logical fallacies, context inconsistency, or statements that contradict common sense. We define seven specific error types across

| | Design Quality | | | | Content Quality | | |
|---|---|---|---|---|---|---|---|
| Dimensions | Missing | Occlusion | Overflow | Contrast | Reasonableness | Comprehensiveness | Faithfulness |
| Results | 97.5% | 94.0% | 94.5% | 95.0% | 93.0% | 0.912 / 0.892 | 94.0% |

Table 2: **Agreement** between our automated multi-modal evaluation and human assessments.

two categories. The first category covers *general content errors*: 1) Factual and Commonsense Inaccuracy. Statements that contradict established facts or widely accepted common sense (e.g., "humans can survive without oxygen"). 2) Context Inconsistency. Internal contradictions within the response, especially misalignment between text and charts. 3) Temporal Inconsistency. Mismatches in time-related statements, or failure to align them with a benchmark date (set to the response generation time), including confusing past events with future predictions. The second category covers *visualization-related errors*: 4) Numerical Implausibility. Chart data that violates basic numerical logic (e.g., percentages in a pie chart not summing to 100%). 5) Unit Inconsistency. Comparing values with different units on the same axis without a clear indication. 6) Missing Essential Information. Omission of key elements such as titles, legends, or axis labels, which hinders unambiguous interpretation. 7) Inappropriate Chart Type. Selecting a visualization form that does not suit the analytical purpose (e.g., using a pie chart for time-series trends). These error types are encoded into a structured prompt for accurate automated evaluation. Because any such error can misinform the user, we apply strict binary scoring: 100 if entirely reasonable, 0 if any error is found.

**Comprehensiveness** assesses whether a response covering all aspects explicitly requested in the query. Following WildBench (Lin et al., 2025), we first generate a checklist for each query to evaluate coverage and adherence of its corresponding response. Using WildBench's scoring rubrics and prompt, an LLM evaluator then scores the response's comprehensiveness based on this checklist. As WildBench adopts a 1–10 scale, we multiply the obtained score by 10 to match other dimensions.

**Faithfulness** evaluates whether the generated content strictly adheres to the provided references, thereby detecting hallucinations. To achieve a fine-grained evaluation, we employ a "claim extraction and verification" strategy. First, Gemini-2.5-pro extracts factual statements (i.e., claims) from the response. For text, it targets assertions relevant to the user's query. For charts, it converts structured data into self-contained facts (e.g., transforming a data point on weight into "The iPhone 16 Pro Max weighs 225 grams" rather than simply describing that "The chart shows a weight of 225 grams."). Each claim is then cross-checked against the references and labeled as *Entailment*, *Contradiction*, and *Neutral*. A claim is considered faithful if labeled *Entailment*. The score of a response equals the percentage of its claims judged as faithful (e.g., 95% yields a score of 95).

### 4.4 AGREEMENT WITH HUMAN EVALUATION

To validate the effectiveness of our automated evaluation system, we analyzed its agreement with human expert judgments. For each design quality dimension (excluding oversized element due to its unambiguous definition), three experts in data visualization and HTML coding independently annotated 200 randomly selected samples, comprising 100 labeled "correct" and 100 "incorrect" by our system. For content quality, we followed WildBench to correlate *Comprehensiveness* scores with Chatbot Arena Elo ratings. For the other three dimensions, we also randomly collected 100 samples each for human annotation. Specifically, we randomly sampled 10 responses per model for *Reasonableness*, and performed stratified sampling for *Faithfulness* with 40 entailment, 30 neutral, and 30 contradiction. These were annotated by three native speakers capable of reading HTML code. In all human annotation tasks, the majority vote was used as the gold standard. Results are shown in Tab. 2 (SRCC/Pearson for Comprehensiveness, Accuracy for others). Further details on inter-annotator agreement, Precision, and Recall can be found in Appendix M.2. An analysis of potential self-preference biases in the LLM evaluator is provided in Appendix N and Appendix O.

## 5 EXPERIMENTS

In this section, we evaluate ten leading commercial and open-source LLMs on RAViG-Bench. The results reveal specific weaknesses in current LLMs and provide guidance for future improvements.

### 5.1 SETUP

**System Prompt.** For a fair and comprehensive evaluation, we developed a structured system prompt to maximize model potential on this complex task. Inspired by the core principles of Chain-of-Thought (Wei et al., 2022), the task is divided into six sub tasks: 1) Task understanding: ensure the final output is complete HTML code based on the references. 2) Content parsing: extract the topic

| Metric | GPT5* | Gemini2.5* | GPT4O* | Qwen3 | DeepSeekv3 | Doubao1.5* | Claude4* | Llama3 | Llama4 | Mistral |
|---|---|---|---|---|---|---|---|---|---|---|
| FSR↑ | **0.997** | 0.994 | 0.973 | 0.951 | 0.946 | 0.968 | **0.997** | **0.997** | 0.888 | 0.994 |
| DSR↑ | 0.819 | 0.685 | 0.878 | 0.763 | 0.814 | 0.724 | 0.652 | **0.882** | 0.762 | 0.274 |
| Comprehensiveness↑ | **93.19** | 86.18 | 72.18 | 78.16 | 74.71 | 72.25 | 80.72 | 63.70 | 61.90 | 66.45 |
| Reasonableness↑ | **95.18** | 74.43 | 54.97 | 43.37 | 34.43 | 68.74 | 41.47 | 51.14 | 49.71 | 40.97 |
| Faithfulness↑ | **98.14** | 92.98 | 84.04 | 88.55 | 84.62 | 88.81 | 86.26 | 78.08 | 82.67 | 84.01 |
| Faithfulness (Case)↑ | **85.34** | 51.54 | 38.00 | 31.41 | 24.66 | 57.45 | 21.25 | 39.22 | 42.41 | 35.50 |
| ECQ↑ | **95.50** | 84.53 | 70.40 | 70.03 | 64.59 | 76.60 | 69.48 | 64.31 | 64.76 | 63.81 |
| HPS↑ | **78.24** | 57.92 | 61.82 | 53.44 | 52.59 | 55.47 | 45.29 | 56.72 | 49.34 | 17.50 |
| $VC_{\text{score}}$↑ | **1.482** | 0.503 | -0.377 | 0.557 | 0.321 | -0.263 | 0.416 | -1.127 | -0.984 | -0.528 |
| $HPS_{vcc}$↑ | **78.24** | 57.92 | 54.82 | 53.44 | 52.59 | 51.10 | 45.29 | 37.55 | 34.78 | 14.72 |

Table 3: **Comprehensive performance of all models.** For each metric (row), the **best** result is highlighted in bold and the worst result is underlined. *denotes commercial closed-source LLMs.

and data from the reference document. 3) Chart selection: choose a suitable visualization format. 4) Layout design: plan the head and body structure, including styles and visualization containers. 5) HTML coding: write correct HTML code containing the chosen visualizations. 6) Validation and refinement: check that the output follows the reference documents and renders correctly, and output only the HTML code. Due to confidentiality constraints, we do not provide the full prompt. However, this omission does not affect the validity of our results, as all models were tested under the same controlled conditions. Examples of visually-rich responses are provided in Appendix F.

**Complexity-based Score Correction.** To mitigate bias from overly simple designs, we apply a complexity-based correction factor, $\gamma_{\text{vc}}$. This factor is derived from a visual richness score ($VC_{\text{score}}$), calculated as follows:

$$VC_{\text{score}} = 0.7z_{\text{module}} + 0.2z_{\text{chart}} + 0.1z_{\text{table}}. \tag{3}$$

Here, each $z_m$ is the z-score for a specific element type $m$ (distinct visual modules, charts, or tables), defined as $z_m = (x_m - \mu_m)/\sigma_m$. In this formula, $x_m$ is the average element count of the model, and $\mu_m, \sigma_m$ are the cross-model mean and standard deviation. The weighting scheme emphasizes visual modules, which are strong indicators of design complexity. The final correction factor is then defined as:

$$\gamma_{\text{vc}} = \min(1.0, 1 + \alpha \cdot VC_{\text{score}}), \quad \text{with } \alpha = 0.3. \tag{4}$$

Outputs at or above the average complexity ($VC_{\text{score}} \geq 0$) receive no adjustment, as existing design defect detection sufficiently reflects their quality.

**Metrics.** We define five metrics to comprehensively and hierarchically evaluate model performance. "*Functional Success Rate (FSR)*" measures the proportion of all samples that pass the functionality validation. "*Design Success Rate (DSR)*" measures the proportion of functionality-passed samples that also pass the design quality assessment. "*Effective Content Quality (ECQ)*" measures the mean content quality score over effective samples that pass both prior checks, where the score is the average of three content dimensions. "*Holistic Performance Score (HPS)*" measures the mean content quality score over all samples, assigning 0 to those failing the first two checks. "$HPS_{vcc}$" is the HPS multiplied by the correction factor $\gamma_{vc}$. In addition, we report the per-dimension content quality scores of effective responses. For *Faithfulness*, we also report case-level scores, in which any response containing an issue claim is assigned a score of 0.

**Models.** We evaluate ten leading commercial and open-source LLMs, including Claude-Opus-4-20250514 (Claude4) (Anthropic, 2025), GPT-4o-2024-11-20 (GPT4O) (Hurst et al., 2024), GPT-5-2025-08-07 (GPT5) (OpenAI, 2025a), Gemini-2.5-pro (Gemini2.5) (Comanici et al., 2025), DeepSeek-V3-0324(DeepSeekv3) (Liu et al., 2024a), Qwen3-235B-A22B-Instruct-2507(Qwen3) (Team, 2025), Doubao-1.5-Vision-Pro (Doubao1.5) (Guo et al., 2025a), Llama-4-Scout-17B-16E-Instruct (Llama4) (Meta AI, 2025), Llama-3.3-70B-Instruct (Llama3) (Dubey et al., 2024), and Mistral-Small-24B-Instruct-2501 (Mistral) (Mistral AI, 2025). For brevity, we use these abbreviations in the tables, figures, and some analyses throughout this section. Specifically, we set the hyperparameter temperature to 0 (except for GPT-5, which only accepts the default temperature=1.0). For the reasoning models, we set the thinking budget to 1024. We further provide in Appendix P an analysis of model behavior under increased generation diversity (temperature=0.7) and additional evidence of the robustness of our evaluation framework.

## 5.2 MAIN RESULTS

The overall results in Tab. 3 show that while both commercial and open-source LLMs can almost perfectly produce syntactically correct and functional outputs, design quality remains the primary bottleneck. Across models, Design Success Rates are notably lower than Functional Success Rates, with values mainly ranging from 60% to 80%. Considering the content quality, GPT5 performs best on almost all metrics, followed by Gemini2.5 and Doubao1.5, whereas Mistral records the weakest performance. The holistic performance score (HPS) reflects overall ability without adjusting for design complexity, with GPT5 leading at 78.24, well ahead of the second-best model and indicating substantial room for improvement in this task. Applying the complexity-based correction (HPS$_{vcc}$) provides a more realistic assessment: it reveals that some models, such as Llama3, achieve inflated design scores through overly simplistic layouts, whereas GPT5 and Gemini2.5 maintain relatively strong performance after correction. These findings suggest that future progress in RAViG will depend less on basic renderability, and more on advancing the integration of rich, accurate, reliable content with high-quality visual designs. Some content-issue cases are provided in Appendix I. Additional experimental results and analysis are provided in Appendix Q.

## 5.3 IN-DEPTH ANALYSIS

**The root of LLM design failures lies in deficiencies in specific capabilities rather than an overall lack of design sense.** Fig. 5 shows the category-level counts of design issues for the five most error-prone models. *Low Color Contrast (Fig. 4 (e))* is particularly salient, suggesting that achieving adequate contrast requires understanding color semantics, the interaction between foreground and background, and the visual impact of color combinations, beyond syntax validity. *Oversized Elements (Fig. 4 (a))* in Gemini2.5 are mainly caused by faulty CSS dependencies for SVG icons. Although the system prompt provides a valid dependency, the model, possibly due to overconfidence or pre-training bias, still uses a specific outdated version, leading to rendering failures. Doubao1.5 frequently produces *Overflow (Fig. 4 (d))*, indicating a weak grasp of global spatial layout.

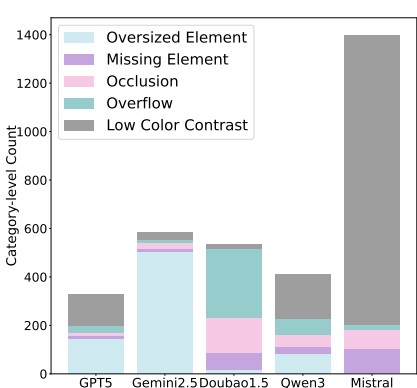

Figure 5: Distribution of design issues.

**Some LLMs struggle to maintain contextual consistency in visually-rich HTML generation.** An analysis of failures on the reasonableness dimension reveals that the primary issues are contextual inconsistency (Fig. 26), tense confusion (Fig. 28, left), and missing chart elements (Fig. 28, right). Contextual inconsistency is the most prevalent, and most often manifests as chart-text inconsistency (Fig. 25). These may stem from the inherent complexity of HTML generation: unlike linear plain-text generation, HTML output consists of non-contiguous segments such as text, tags, and script code, requiring the model to sustain long-range contextual memory and chart-text alignment across components, while the separated content structure further increases the difficulty of maintaining consistency.

**Most LLMs show vulnerability in faithfulness, as high accuracy on individual facts does not ensure a fully trustworthy response.** This gap emerges when comparing claim-level scores with the stricter case-level scores. Tab. 3 shows that most models exceed 80% faithfulness at the claim-level but drop sharply at the case-level; for instance, Claude4 falls from 86.3 to 21.3, indicating that nearly 80% of its responses contain factual errors (Fig. 30). Since a single mistake can undermine user trust, the strong case-level performance of models like GPT5 and Doubao1.5 signals a higher capacity for generating holistically reliable content, an essential step toward real-world applicability.

## 6 CONCLUSION

In conclusion, we introduce RAViG-Bench , the first benchmark with an automatic multi-modal evaluation framework for comprehensive assessment of LLM performance on RAViG. We adopt HTML as the representation, but the benchmark is not tied to HTML or specific visual elements. The dataset, evaluation framework, algorithms, and criteria are format agnostic; adapting to other structured formats only requires modifying the input and output processing modules. Experiments show that even state-of-the-art LLMs still have substantial room for improvement. While our work provides the first systematic capability to objectively detect readability-related defects and perform

basic visual complexity correction, it does not yet address subjective aesthetics or incorporate richer factors such as typography and layout. In addition, aesthetic judgments related to "over-design" are excluded, so visually richer yet high-quality outputs are considered evidence of model capability. Addressing these limitations offers promising directions for future work to incorporate user-centered aesthetic evaluation and nuanced visual refinement, extending the benchmark from assessing basic usability to evaluating outputs for visual appeal, accessibility, and overall user experience.

## ETHICS STATEMENT

All data used in this benchmark are either collected from the open-source Natural Questions (NQ) dataset or generated by large language models, and do not contain personally identifiable information or sensitive personal data. Licenses for all third-party datasets and open-source models used in this work are listed in Appendix C. For any human annotations involved, annotators were informed of the research purpose and consented to participate; the annotation process posed no physical, psychological, or legal risks. To mitigate bias, during the final human review for query selection and reference document annotation, we filtered out content with overt discrimination, ethical concerns, or harmful elements. We explicitly prohibit the use of this benchmark or any derived models in applications intended to discriminate against individuals or groups, or in any contexts that could cause harm. The full benchmark, along with associated code, will be released upon acceptance under a license that permits research use while prohibiting unlawful or malicious purposes. No part of this work violates privacy rights, confidentiality agreements, or applicable laws and regulations.

## REPRODUCIBILITY STATEMENT

We have taken multiple steps to ensure the reproducibility of our work. For dataset construction, we provide detailed selection criteria, categorization scheme, and the end-to-end pipeline for assembling both real-world and synthetic queries in Sec. 3.1 and Appendix E. For all LLM-based evaluation methods, we report in Sec. 4.2 and Sec. 4.3 the full set of dimensions and structural organization contained in the prompts. For our rule-based baselines, Sec. 4.2 include the complete algorithmic workflows and relevant hyperparameter settings. We commit to publicly releasing the full dataset, all prompts, and code upon acceptance. These resources, together with the descriptions in the main text and supplementary materials, will enable other researchers to fully reproduce our results.

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

## A   RELATED WORK

**Data Construction with Real-World Use Cases and LLMs.** To ensure alignment with real-world LLM usage scenarios, the majority of our benchmark data is sourced from a real-world dataset Natural Questions (Kwiatkowski et al., 2019). Works (Chiang et al., 2024; Li et al., 2023) like WildBench (Lin et al., 2025) also leverage real-world data, underscoring its importance in LLM evaluation. Additionally, several open-ended benchmarks rely on human experts or crowdsourcing workers for data creation (Srinivasan et al., 2021; Guo et al., 2025b). On the other hand, LLMs have been widely used to facilitate data synthesis, enhancing data diversity (Yuan et al., 2021; Yang et al., 2024b; Yoo et al., 2021; Chen et al., 2022; Luo et al., 2021b). For example, NL2VL (Ko et al., 2024) employs LLM-generated samples followed by human filtering, establishing a new paradigm that significantly improves data quality. Following this trend, we leverage LLMs to augment our dataset, addressing certain limitations of the filtered real-world data.

**Benchmarks for Retrieval-Augmented Generation (RAG).** Significant progress has been made in the evaluation of Retrieval-Augmented Generation (RAG) (Gan et al., 2025). Existing research predominantly focuses on the quality of text outputs, assessing aspects like accuracy, faithfulness, and hallucination (Sorodoc et al., 2025; Yang et al., 2024a; Saad-Falcon et al., 2024; Liu et al., 2023; Niu et al., 2024; Katsis et al., 2025). Evaluation paradigms often achieve in-depth analysis by independently assessing key components such as the retriever and generator (LLM), designing complex scenarios, or introducing novel metrics. For instance, RAGAS (ES et al., 2024) explores automated, ground-truth-free evaluation using LLMs, while RGB (Chen et al., 2024) tests model robustness by injecting noise. Another line of research extends the evaluation scope to multi-modal understanding (Zhan et al., 2025; Strich et al., 2025; Masry et al., 2022; Dong et al., 2025; Ma et al., 2024; Ding et al., 2024; Lim et al., 2025; Wasserman et al., 2025; Hu et al., 2025). Visual-RAG (Wu et al., 2025), for example, focuses on a model's ability to retrieve and comprehend image information. However, such work focuses on models' *understanding* of multimodal content; the capacity for LLMs to *generate* visually rich responses remains largely unexplored. Among these, MRAMG-Bench (Yu et al., 2025) is most similar to our setting, but despite involving text-image layout, it is essentially a text-only generation task, as its answers combine model-generated text with images retrieved from source material. In contrast, our benchmark requires models to autonomously design and produce visually-rich answers. Furthermore, some studies have assessed the code generation capabilities of RAG-enhanced models (Liu et al., 2024b; Wang et al., 2025). However, their emphasis is on understanding code context and generating correct outputs, with the code itself being the final object of evaluation. In summary, there is currently no benchmark that can perform a comprehensive, end-to-end evaluation of an LLM's ability to synthesize information from multiple documents and generate appropriate visually rich answers in response to open-ended analytical queries.

**Benchmarks for Natural Language to Visualization Generation.** Recent years have seen rapid progress in using LLMs for natural language to visualization (NL2VIS) tasks, leading to various benchmarks and datasets to systematically measure and advance model capabilities in this area ( (Cheng et al., 2023; Vázquez, 2024; Li et al., 2024; Luo et al., 2021a; Hu et al., 2019; Fu et al., 2020)). For example, nvBench 2.0 (Luo et al., 2025) provides support for complex and ambiguous queries, paired with disambiguation reasoning paths. VisEval (Chen et al., 2025) introduces a high-quality dataset and an automated evaluation framework, encompassing validity, legality, and readability of generated code and charts. To enable more fine-grained evaluation and feedback, EvaLLM (Podo et al., 2024) decomposes the evaluation into semantic and procedural components, and implements this through an evaluation platform that supports both automated and manual assessment. Vis-Shepherd (Pan et al., 2025) further introduces a multimodal LLM-based critic and a specialized critique dataset to enable automated feedback. In addition, dialogue-based datasets (Song et al., 2024), human-collected query corpora (Srinivasan et al., 2021), and domain-specific benchmarks (Joseph et al., 2025) further enrich the evaluation landscape. However, these benchmarks mainly focus on direct conversion from structured queries (typically including both the database and a data-specific query) into a single visual component, where the LLM's role is limited to generating a visualization based on given inputs. In contrast, our setting is more challenging and user-centric: given open-ended questions (e.g., "What is the trend of gold prices?"), the model must decide when and how to incorporate visualizations into a rich-text response. Existing benchmarks fail to evaluate the quality of visually-rich responses generated by LLMs.

| Key | Description |
|---|---|
| page id | A unique identifier for the document within a set of query results or references. |
| page url | The website or platform where the document was originally published. |
| page title | The main title of the document. |
| page synopsis | A brief overview or abstract of the document's content. |
| page content | The complete textual content of the document. |
| last modified time | The date and time when the document was published. |

Table 4: **Field descriptions** of the reference document schema.

**Deep Research Agents and Data Insight Benchmark.** Our work is also related to "deep research" agents Google (2025); Team et al. (2025); OpenAI (2025b) and data insight benchmarks like InsightBench Sahu et al. (2024). While these can also generate visual reports, RAViG-Bench is distinguished by its specific task focus and evaluation methodology. First, the RAViG task is defined for the real-time, realistic, and everyday context of conversational AI, contrasting with the time-intensive, in-depth, and analyst-style nature of typical "deep research" tasks. Second, and more importantly, our evaluation framework provides a novel, holistic assessment not found in existing benchmarks. For instance, InsightBench evaluates the correctness of data insights derived from structured data. In contrast, RAViG-Bench evaluates an LLM's ability to synthesize information from unstructured web documents and present it in a multi-component visual layout. Our evaluation is also broader, covering not only content quality (including chart-text consistency) but also functionality and visual design quality, aspects crucial for user-facing outputs but not addressed by prior benchmarks. Notably, our framework is versatile and can be used to evaluate the HTML output of any agent, including those designed for deep research.

## B    DETAILED FIELDS OF RETRIEVED REFERENCE DOCUMENTS

Document Fields are listed in the Tab. 4.

## C    LICENSES

- Natural Questions is released under the Creative Commons Share-Alike 3.0 license.
- DeepSeek-V3 is licensed under the MIT License.
- Llama-4-Scout-17B-16E-Instruct is licensed under the Llama 4 Community License.
- Llama-3.3-70B-Instruct is licensed under the Llama 3 Community License.
- Mistral-Small-3.1-24B-Instruct-2503 and Qwen3-235B-A22B-Instruct-2507 are licensed under Apache 2.0 License.

## D    THE USE OF LARGE LANGUAGE MODELS (LLMS)

In the preparation of this paper, we utilized the Google Gemini Pro model (version Gemini-2.5-Pro-0613 accessed via the Google AI Platform) as a writing assistant. The use of the LLM was strictly limited to the following assistive tasks:

**Language Enhancement**: The model was employed to improve grammar, refine phrasing, and enhance the clarity and readability of the text. All suggestions were critically reviewed by the authors, and only those that accurately reflected our intended meaning were incorporated.

**Prompt Optimization**: For the prompt-based components of our methodology, we used the LLM to brainstorm and iterate on prompt structures. The final prompts used in our experiments were developed and finalized by the authors.

Crucially, the LLM did not contribute to the core research ideas, experimental design, or the analysis of results. The intellectual contributions, including the formulation of the research problem, the development of the proposed method, and the interpretation of the findings, are entirely the work of

| Type | Intention | Typical Visualization |
|------|-----------|----------------------|
| **Trend** | Changes over time. | Line Chart, Area Chart |
| **Proportion** | Part-to-whole relationships. | Pie Chart, Donut Chart |
| **Comparison** | Comparing distinct items. | Bar Chart, Radar Chart |
| **Distribution** | How data is spread across a range. | Histogram, Box Plot |
| **Ranking** | Ordering items by a metric. | Sorted Bar, Leaderboard |
| **Details** | Retrieving raw/detailed data. | Card List, Enhanced Table |

Table 5: Six query types for visually rich answers generation.

| Type | Representative Query Examples |
|------|------------------------------|
| **Trend** | - Yearly average ocean pH levels since 1980
- Number of billionaires worldwide by year since 1990
- Annual number of 3D printers sold globally since 2007
- What is the monthly average temperature in antarctica?
- The trend in Netflix subscriber growth each quarter since 2013 |
| **Proportion** | - What is the ratio of gases in air?
- Composition of Earth's crust by major elements
- Of the total water on earth what percent is freshwater?
- What percent of american adults have a bachelor's degree?
- What percent of the us population is employed in agriculture? |
| **Comparison** | - Minimum wage vs cost of living by country
- Compare the speed of a cheetah versus a gazelle
- Who has sold more albums: Taylor Swift or Beyonce?
- How many times bigger is vy canis majoris than our sun?
- How does the temperature of venus compare to that of earth? |
| **Distribution** | - Distribution of scores on the 2024 LSAT exam
- Distribution of individual income in the United States
- What is the distribution of average sleep quality scores?
- Distribution of children's book word counts published in 2023
- Describe the scope of the foodservice industry in relation to its size |
| **Ranking** | - Top 10 box office movies in the world
- 10 most populated capital cities in the world
- Three largest cities in the world by population
- 10 wealthiest countries in the world by income
- What are the eight most common elements in the earth's crust? |
| **Details** | - GDP per capita by state in the us
- What are the depths of the five great lakes?
- Population of the 5 boroughs of new york city
- Details of all Nobel Prize winners in Literature since 2010
- List major volcanic eruptions since 2000 with VEI and fatalities |

Table 6: Examples of dataset queries

the human authors. All final content, data, and conclusions presented in this paper were personally verified and approved by the authors to ensure scientific accuracy and integrity.

# E    DETAILED DESCRIPTION OF QUERY CATEGORY AND EXAMPLES

As introduced in Section 3.2, the queries in RAViG-Bench are classified into six distinct categories to cover a breadth of real-world analytical tasks. These categories form the foundation for evaluating a model's ability to generate appropriate visually-rich responses. Tab.5 summarizes these categories, detailing the core intention for each type and the typical visualization format suited to address it.

Following the definitions above, Tab. 6 showcases representative queries for each category to ground them in real-world examples.

# F  EXAMPLES OF VISUALLY RICH ANSWERS GENERATED BY LLMS

This section presents some examples of LLM-generated visually-rich responses. It also shows the effects of our full webpage screenshots and module screenshots.

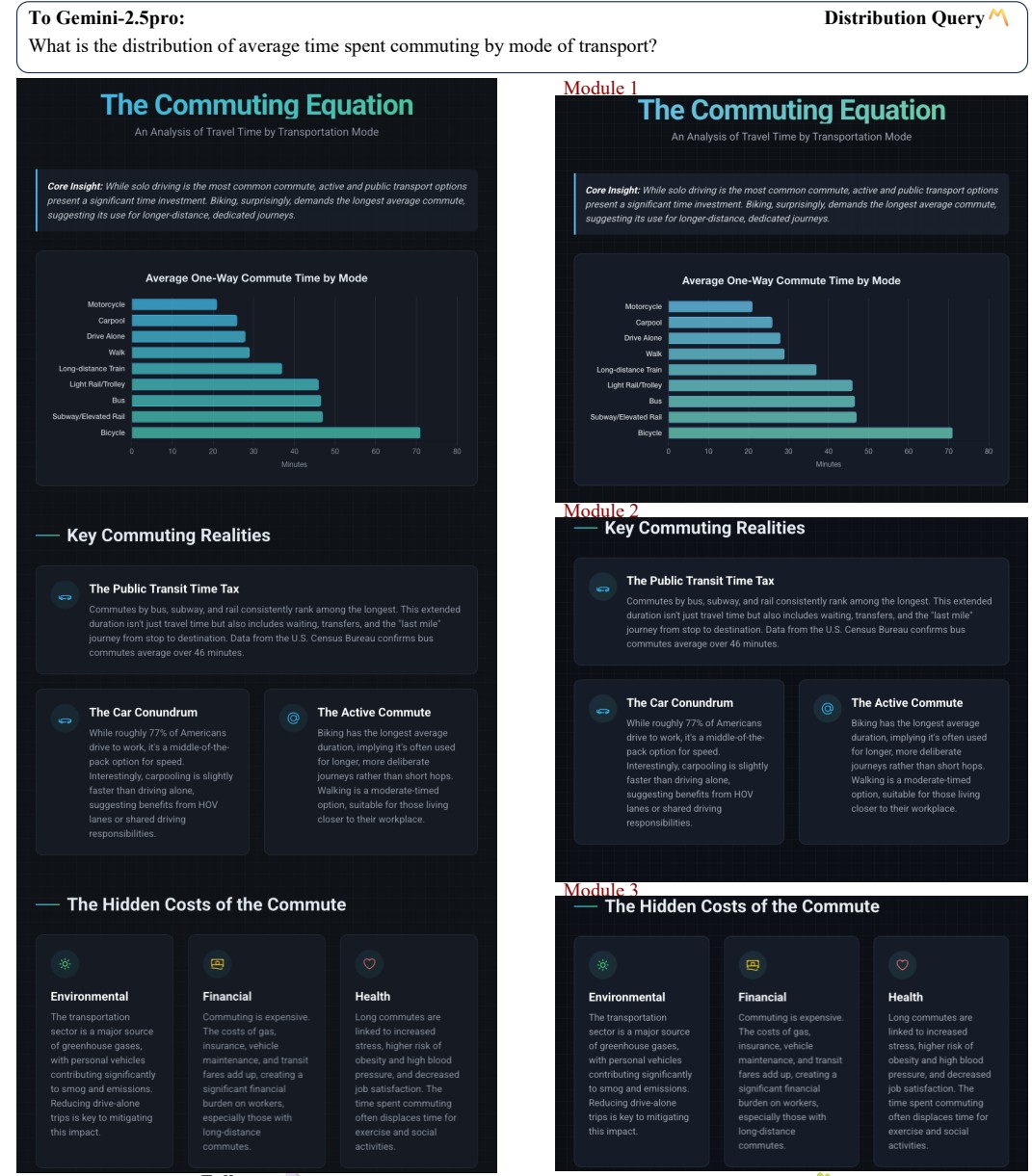

Figure 6: **Visually rich answers generated by Gemini-2.5-pro.**

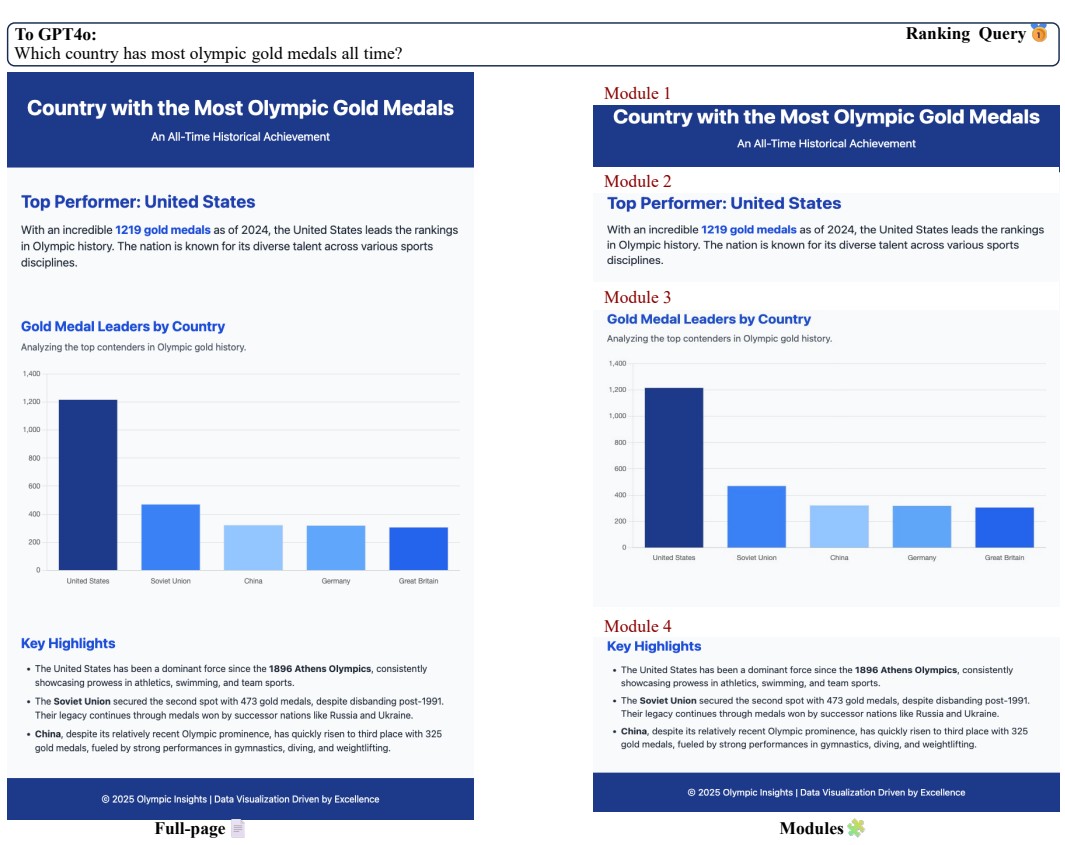

Figure 7: **Visually rich answers generated by GPT-4o.**

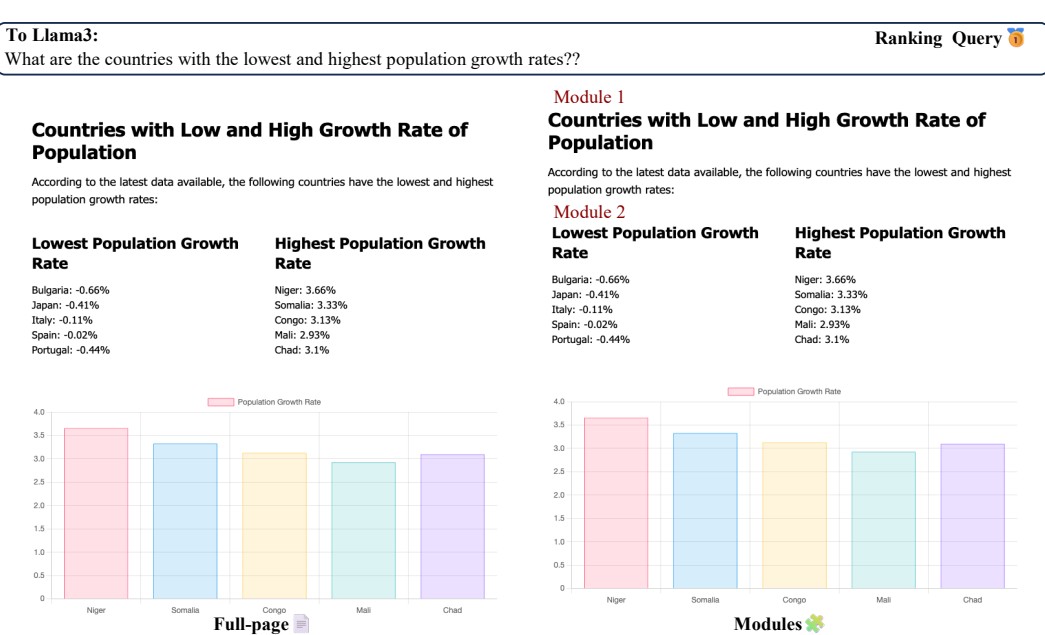

Figure 8: **Visually rich answers generated by Llama3.**

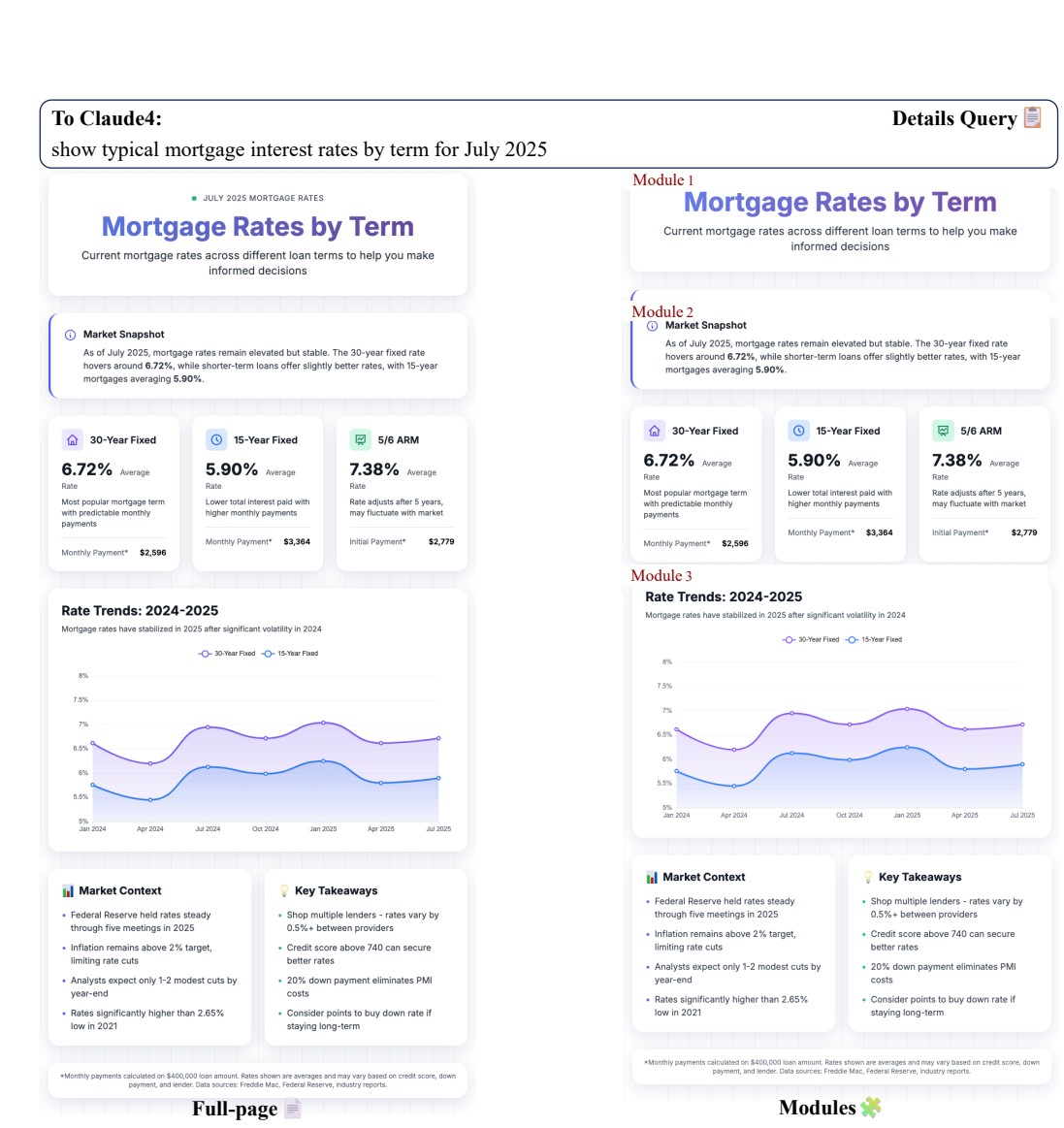

Figure 9: **Visually rich answers generated by Claude-Ops-4.**

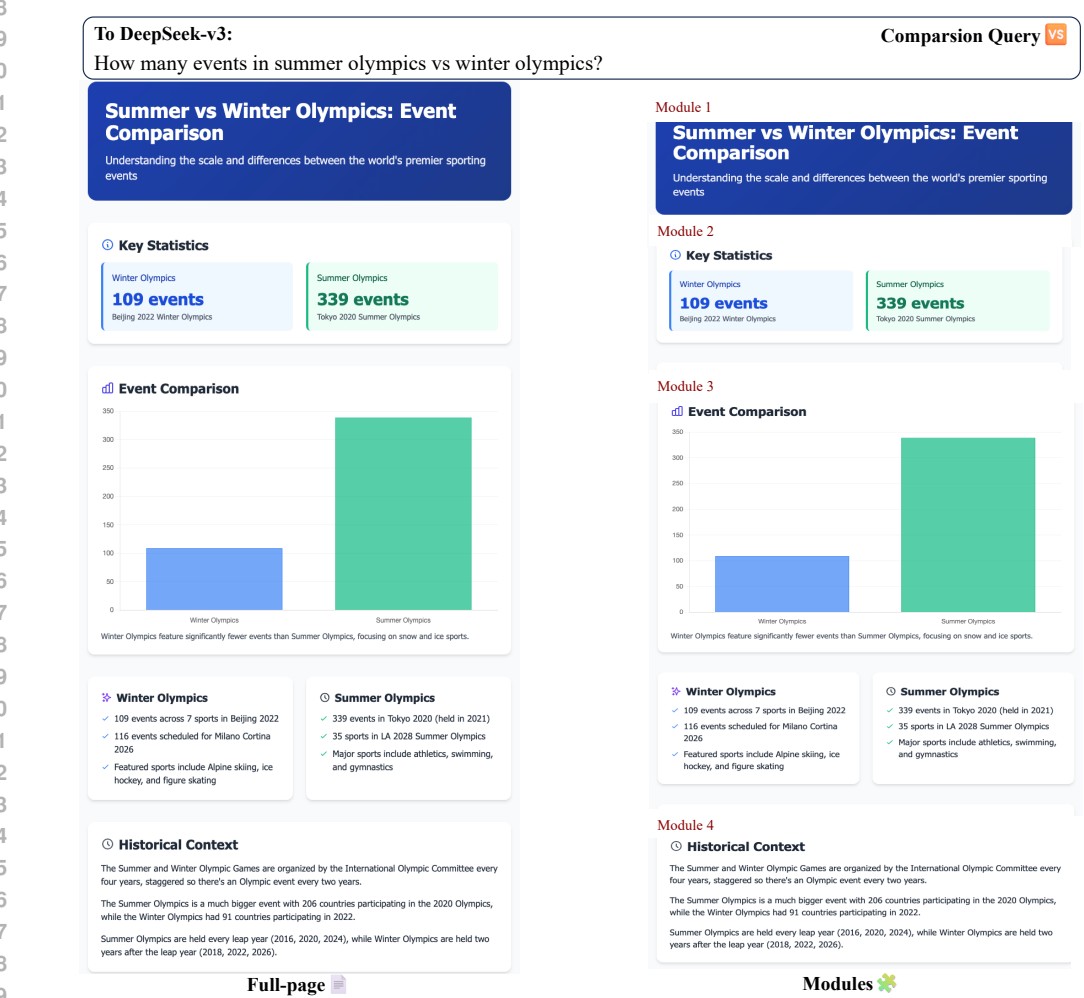

Figure 10: **Visually rich answers generated by DeepSeek-v3.**

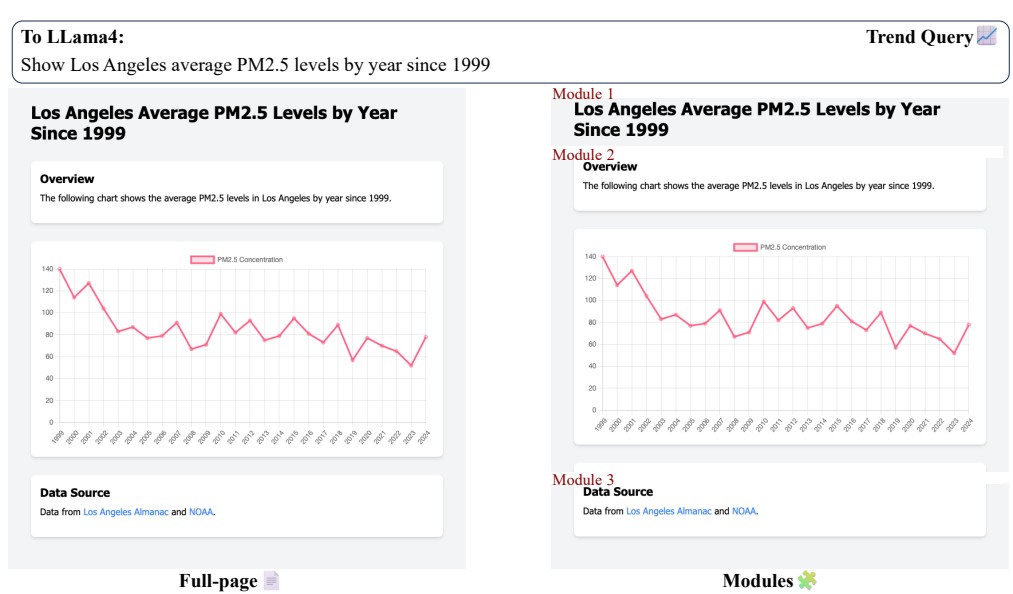

Figure 11: **Visually rich answers generated by Llama4.**

# G    EXAMPLES OF VISUALLY RICH ANSWERS FROM DIFFERENT LLMS FOR THE SAME QUERY

This section presents responses from different LLMs to the same query "How many jimmy johns are there in the us" (for lengthy responses, the output is split into two parts for display from top to bottom). For such open-ended task, traditional evaluation based on a ground truth is not applicable. Consequently, we constructed a GT-free framework to assess the quality of the generated outputs.

**Query: How many jimmy johns are there in the us**

**Model: Claude**

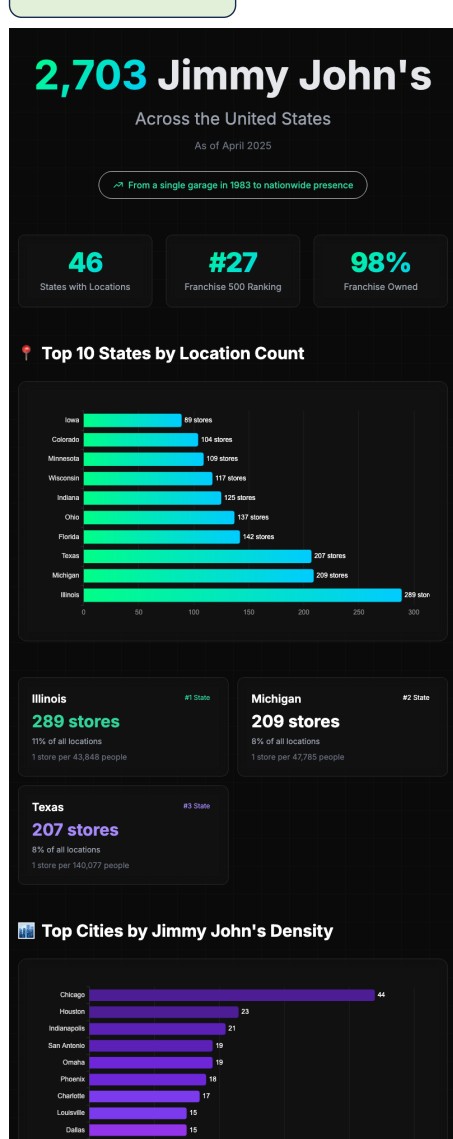
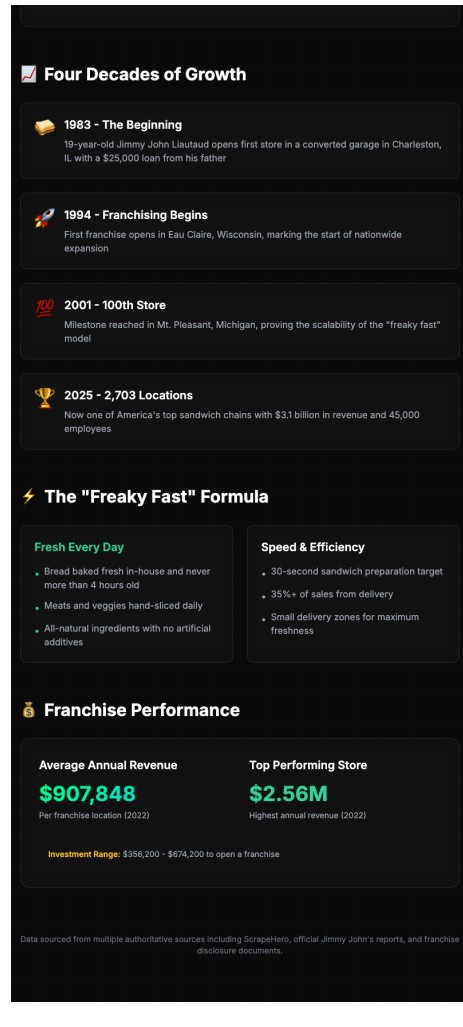

Figure 12: **Query and visually rich answers generated by Claude-Opus-4.**

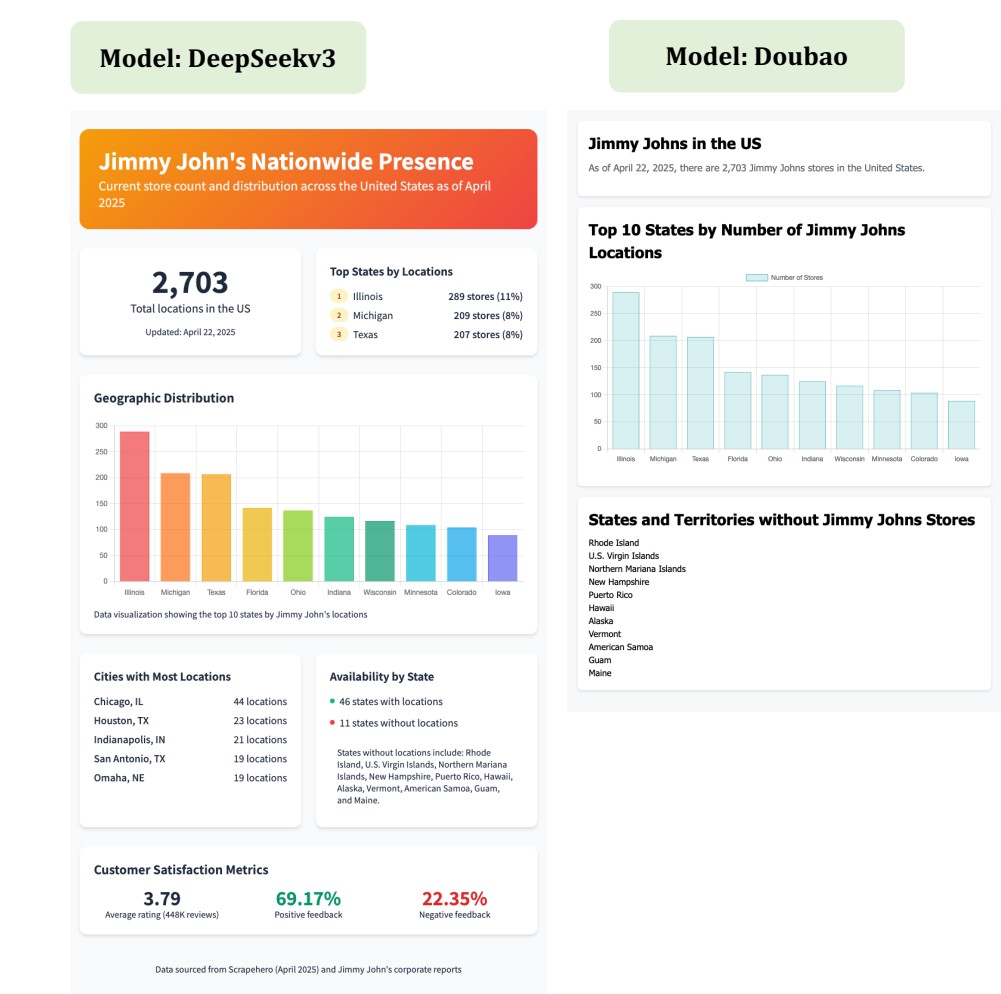

Figure 13: **Query and visually rich answers generated by DeepSeek-v3 and Doubao-1.5-V-pro.**

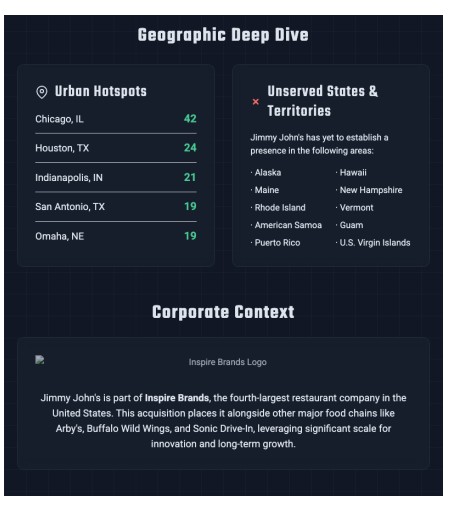

Figure 14: **Query and visually rich answers generated by Gemini-2.5-pro.**

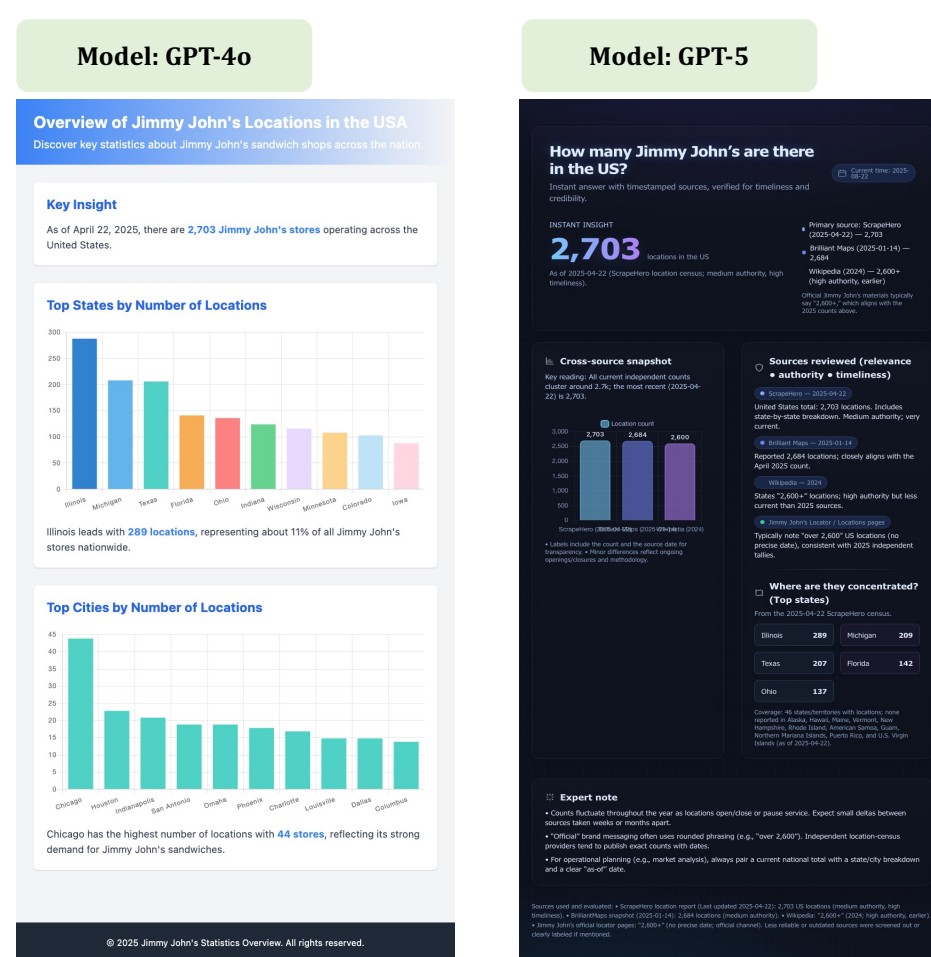

Figure 15: **Query and visually rich answers generated by GPT-4o and GPT-5.**

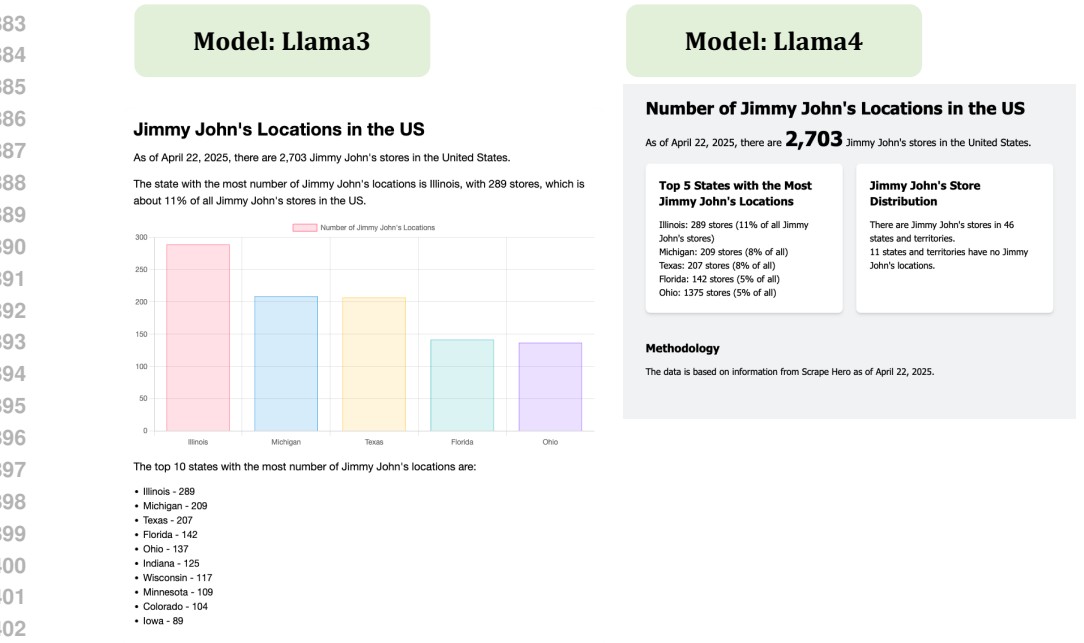

Figure 16: **Query and visually rich answers generated by Llama3 and Llama4.**

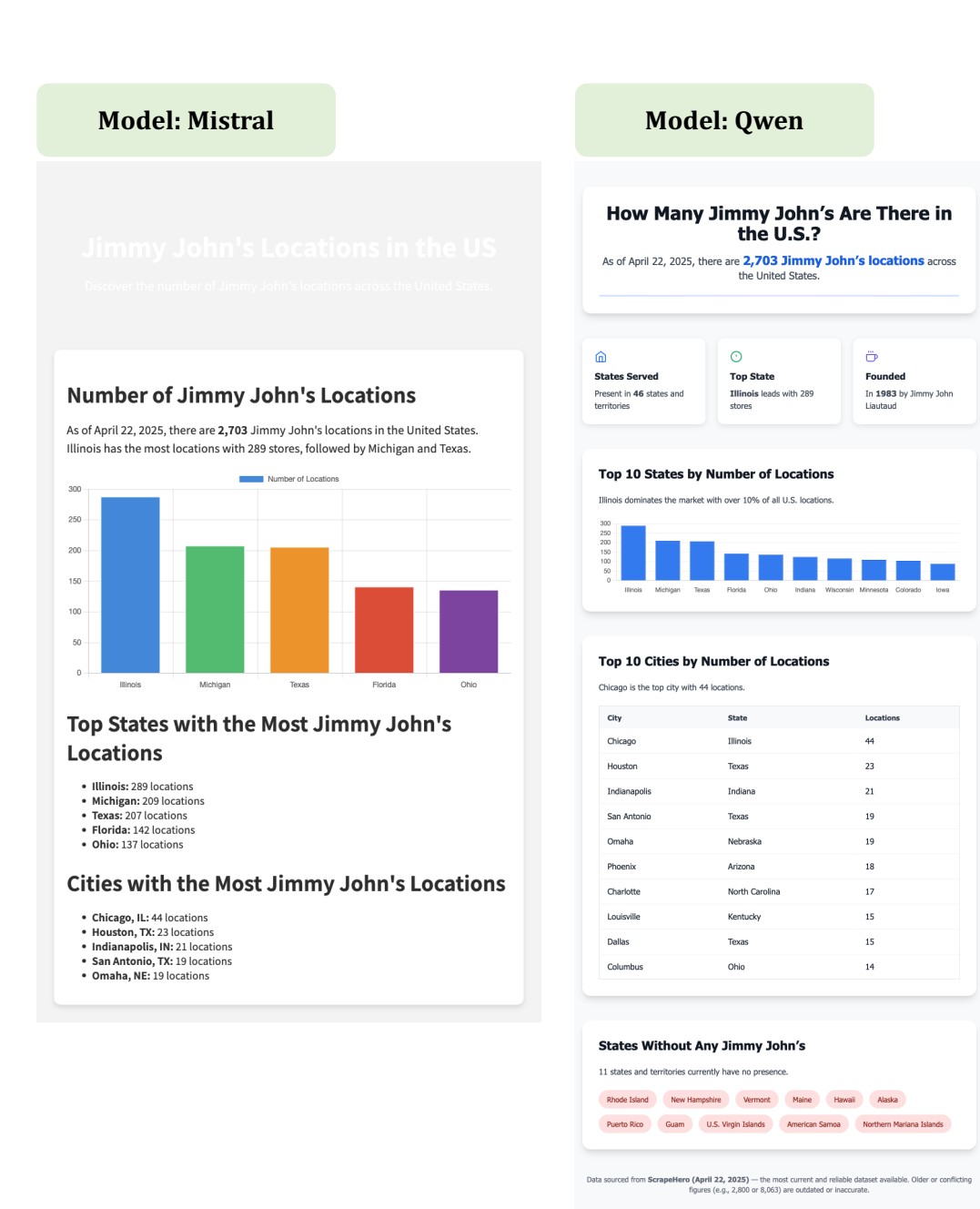

Figure 17: **Query and visually rich answers generated by Mistral and Qwen3.**

## H  EXAMPLES OF HUMAN-MACHINE INCONSISTENT CASES IN DESIGN AND CONTENT DIMENSIONS

In this section, we present an analysis of the inconsistencies between our automated judges (Gemini 2.5-Pro, GPT-4o, and GPT-5) and the majority vote of three human experts. During this analysis, we made a surprising discovery: a subset of the disagreements stemmed not from model error, but from subtle defects that human annotators had initially overlooked. These cases, often requiring meticulous cross-modal inspection (e.g., comparing fine-grained details between rendered output and source code), highlight a limitation of manual evaluation and underscore the advantage of using LLMs-as-Judges. In the following, we show several examples of human-machine inconsistencies, presenting the evaluation results from different models alongside the original human judgments.

### H.1  DESIGN DIMENSIONS

Fig. 18 and 19 show representative examples of disagreement for the Missing Element dimension. Correct assessments are marked in green, and the rationale from models that made a correct judgment is included below each example. As these examples demonstrate, human evaluators can also make mistakes, particularly when identifying subtle issues such as the omission of fine-grained details or defects that require meticulous comparison with the source code.

Figures 20 and 21 present representative examples of disagreement on the Occlusion dimension. Similar to the previous examples, correct judgments are highlighted in green. For cases where a model experienced a hallucination (i.e., incorrectly identified an occlusion), we provide the model's flawed reasoning to illustrate the nature of the error. The occlusions missed by LLMs (false negatives) are typically very subtle. And their hallucinations (false positives) are often triggered by low color contrast or small font sizes.

| Gemini-2.5: Issue | GPT-4o: No Issue |
|---|---|
| GPT-5: No Issue | Human : No Issue |

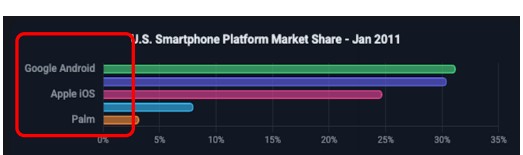

**Gemini2.5 Reason**: In the 'A Market Tipping Point' section, the bar chart with ID 'osMarketShareChart' is not fully rendered, resulting in missing information. The HTML code defines five data categories for the chart ('Google Android', 'RIM (BlackBerry)', 'Apple iOS', 'Microsoft', 'Palm'), but in the rendered screenshot, the chart only shows three of these categories, missing 'RIM (BlackBerry)' and 'Microsoft' and their corresponding data bars. This is a rendering anomaly, preventing users from seeing complete market share comparison information.

| Gemini-2.5: Issue | GPT-4o: No Issue |
|---|---|
| GPT-5: No Issue | Human: No Issue |

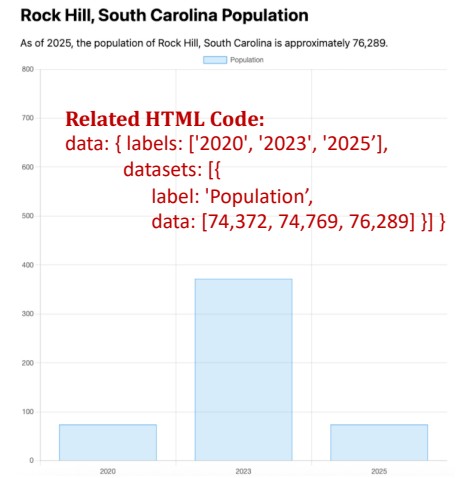

**Gemini2.5 Reason**: Although the chart component (population-chart) is rendered, the data it displays is significantly different from the data intended to be expressed in the code, resulting in errors and missing information. The specific reason is that in the JavaScript code that initializes the chart, the values in the data array use commas as thousands separators, for example, `[74,372, 74,769]`.

**Figure 18: Examples of inconsistencies on Missing Element dimension (1)**, correct assessments are marked in green.

| Gemini-2.5: Issue GPT-4o: Issue | Gemini2.5: No Issue GPT-4o: No Issue |
|---|---|
| **GPT-5: No Issue** **Human : No Issue** | **GPT-5: Issue** **Human: Issue** |

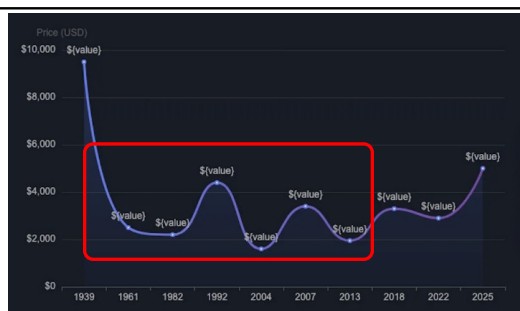

**Gemini2.5 Reason**: In the "Foreign Holders of U.S. Debt" bar chart, the legend is displayed as "Undefined," indicating missing information. This is because the `datasets` object in the JavaScript code used to initialize the chart lacks a 'label' property. This prevents the chart library from generating meaningful legend labels, leaving users unable to understand the meaning of the chart data.

**GPT-4o Reason**: The second chart component (Foreign Holders of U.S. Debt) was not rendered correctly, resulting in missing information. The specific reasons are as follows: 1. There is a problem with the chart data: the `label` value is not set, causing it to display as `undefined`, thus affecting the rendering of the chart.

**GPT-5 Reason**: The values in the "Price Evolution Timeline" line chart are not displayed correctly: the y-axis ticks and data point labels are displayed as the literal string "$value" without showing the actual numbers. This results in the chart lacking crucial numerical information. The specific reason is an error in the ECharts formatter configuration: the code uses `axisLabel.formatter: '${value}'` and `series.label.formatter: '${value}'`. In ECharts, '{value}' (axis ticks) or '{c}'/function (data values) should be used instead of placeholders with dollar signs, thus it is currently being rendered as plain text. Other content (such as the "Global Smart TV Market Share 2022" pie chart, text cards, and statistics) displays correctly in the screenshot, and no external resources or images failed to load.

Figure 19: **Examples of inconsistencies on Missing Element dimension (2),** correct assessments are marked in green.

| Gemini-2.5: Issue GPT-4o: Issue | Gemini2.5: No Issue GPT-4o: Issue |
|---|---|
| **GPT-5: No Issue** **Human : Issue** | **GPT-5: No Issue** **Human: Issue** |

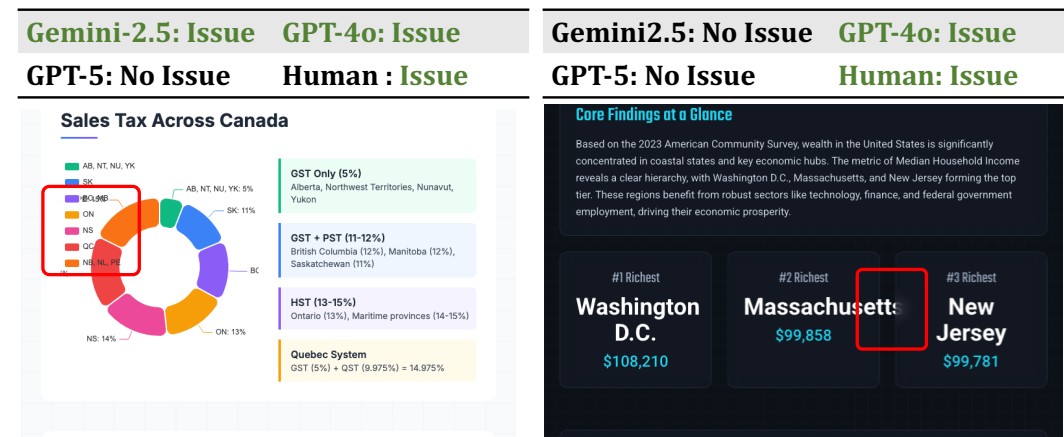

Figure 20: **Examples of inconsistencies on Occlusion dimension (1),** correct assessments are marked in green.

| Gemini2.5: N-Issue | GPT-4o: N-Issue | Gemini2.5: N-Issue | GPT-4o: Issue |
|---|---|---|---|
| GPT-5: Issue | Human : N-Issue | GPT-5: N-Issue | Human: N-Issue |

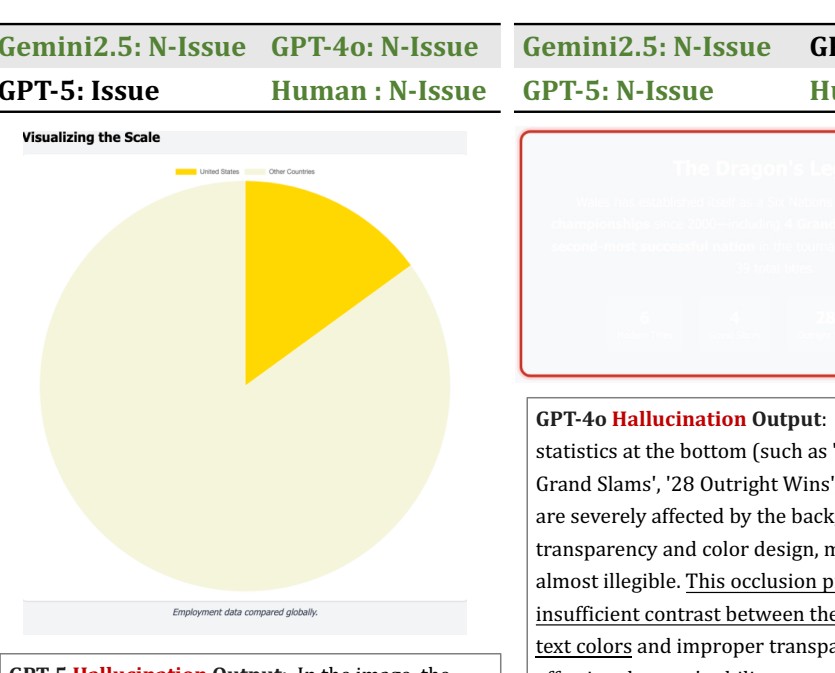

**GPT-5 Hallucination Output**: In the image, the large pie chart overflows its container boundary, and the bottom pie chart area covers and overlaps with the area containing the caption "Employment data compared globally." This overlap and conflict between the description and the chart, interfering with readability. This is a problem of element overlap and occlusion caused by chart overflow, affecting the user's retrieval of chart information.

**GPT-4o Hallucination Output**: In the image, the statistics at the bottom (such as '6 Modern Titles', '4 Grand Slams', '28 Outright Wins', '39 Total Titles') are severely affected by the background transparency and color design, making the text almost illegible. This occlusion problem is caused by insufficient contrast between the background and text colors and improper transparency settings, affecting the user's ability to read and understand key information.

Figure 21: **Examples of inconsistencies on Occlusion dimension (2)**, correct assessments are marked in green.

## H.2 Content Dimensions

Fig. 22 and 23 show representative examples of disagreement for the reasonableness dimension. Correct assessments are marked in green, and the rationale from models that made a correct judgment is included below each example. Assessing reasonableness requires LLMs to possess strong reasoning capabilities as well as a solid understanding of HTML code. Consequently, non-reasoning LLMs typically exhibit poor consistency on this dimension. Even high-performing models such as Gemini-2.5-Pro and GPT-5 can make erroneous judgments due to excessive reasoning (Fig. 23) and knowledge hallucination (Fig. 22). Meanwhile, humans also have gaps in certain areas of common sense, often struggling to identify relatively obscure factual errors or being misled by similar entities.

Fig. 24 show representative examples of disagreement for the faithfulness dimension. Similar to the previous examples, correct judgments are highlighted in green. In terms of faithfulness evaluation, LLMs demonstrate high agreement with human; Gemini-2.5-pro and GPT-5 achieve consistency rates of 94% and 95%, respectively. However, in scenarios involving long texts, noisy inputs, and required reasoning, both human experts and LLMs still exhibit some degree of hallucination.

| Gemini-2.5: Issue | GPT-4o: No Issue | | Gemini-2.5: No Issue | GPT-4o: No Issue |
|---|---|---|---|---|
| **GPT-5: No Issue** | **Human : No Issue** | | **GPT-5: Issue** | **Human: No Issue** |

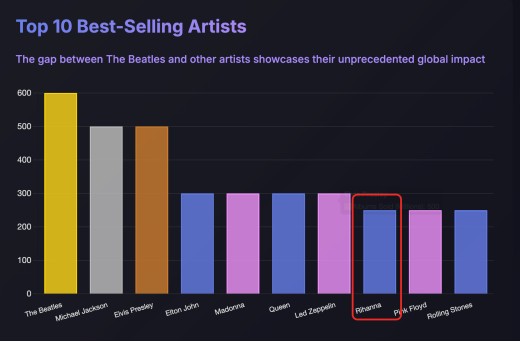

**Gemini-2.5 Reason**: According to official NBA statistics (e.g., from Basketball-Reference), the leaders for the 2017-18 season were Karl-Anthony Towns (68) and Andre Drummond (68). The response lists Towns with 68 but Drummond with 62. It also includes other players with incorrect numbers (e.g., Westbrook had 57, not 59). This is a clear factual error regarding static historical data. The Timeliness Exemption does not apply here. This is a violation.

GPT-5 **Hallucination** Output: The players listed (Karl-Anthony Towns, Andre Drummond, Russell Westbrook, Dwight Howard, LeBron James) and their team associations for the 2017-18 season are correct, and the double-double counts are plausible for that season.

**GPT-5 Reason**: the response repeatedly and explicitly frames figures as `albums sold` and assigns numbers that are widely known to be claims for total `records` (albums + singles), not albums. Examples: Michael Jackson at 500M albums and Elvis Presley at 500M albums are not established album-only figures; **Rihanna at 250M albums is especially implausible, as her widely cited 250M figure refers to total records with album sales far lower.**

GPT-4o **Hallucination** Output: I reviewed the text content for factual and common sense accuracy. The claim that The Beatles have sold 600 million albums globally aligns with widely accepted historical data, and no static factual errors were identified.

Figure 22: **Examples of inconsistencies on Reasonableness dimension (1)**, correct assessments are marked in green.

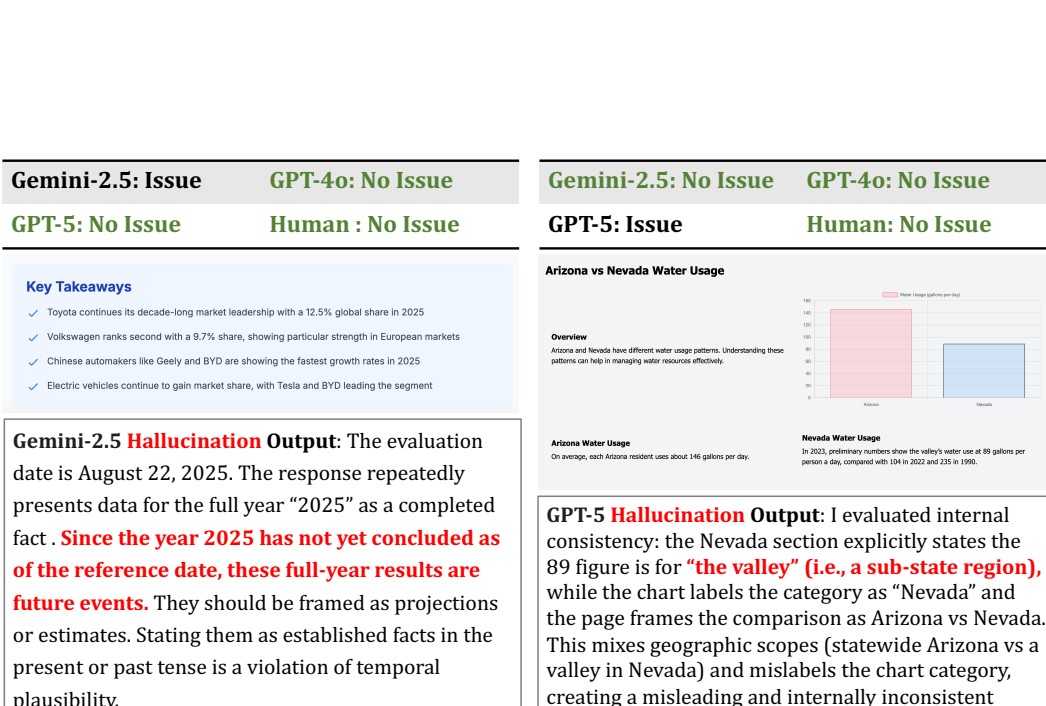

**Gemini-2.5: Issue**     **GPT-4o: No Issue**

**GPT-5: No Issue**     **Human : No Issue**

**Key Takeaways**

✓ Toyota continues its decade-long market leadership with a 12.5% global share in 2025

✓ Volkswagen ranks second with a 9.7% share, showing particular strength in European markets

✓ Chinese automakers like Geely and BYD are showing the fastest growth rates in 2025

✓ Electric vehicles continue to gain market share, with Tesla and BYD leading the segment

**Gemini-2.5 Hallucination Output**: The evaluation date is August 22, 2025. The response repeatedly presents data for the full year "2025" as a completed fact . **Since the year 2025 has not yet concluded as of the reference date, these full-year results are future events.** They should be framed as projections or estimates. Stating them as established facts in the present or past tense is a violation of temporal plausibility.

**Gemini-2.5: No Issue**     **GPT-4o: No Issue**

**GPT-5: Issue**     **Human: No Issue**

**Arizona vs Nevada Water Usage**

**Overview**
Arizona and Nevada have different water usage patterns. Understanding these patterns can help in managing water resources effectively.

**Arizona Water Usage**
On average, each Arizona resident uses about 146 gallons per day.

**Nevada Water Usage**
In 2023, preliminary numbers show the valley's water use at 89 gallons per person a day, compared with 104 in 2022 and 235 in 1990.

**GPT-5 Hallucination Output**: I evaluated internal consistency: the Nevada section explicitly states the 89 figure is for **"the valley" (i.e., a sub-state region)**, while the chart labels the category as "Nevada" and the page frames the comparison as Arizona vs Nevada. This mixes geographic scopes (statewide Arizona vs a valley in Nevada) and mislabels the chart category, creating a misleading and internally inconsistent comparison.

Figure 23: **Examples of inconsistencies on Reasonableness dimension (2)**, correct assessments are marked in green.

| Gemini-2.5: No Issue | GPT-4o: Issue | Gemini-2.5: Issue | GPT-4o: No Issue |
| --- | --- | --- | --- |
| GPT-5: No Issue | Human : No Issue | GPT-5: No Issue | Human: No Issue |

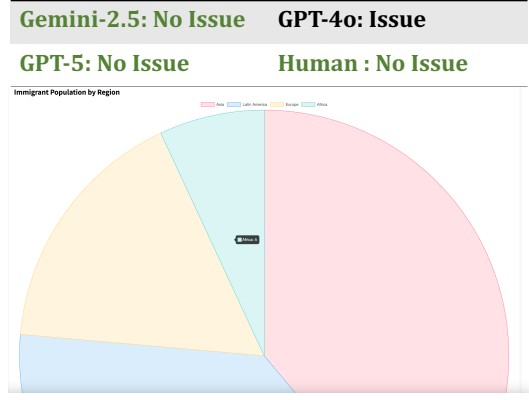

**Claim:** 5% of the U.S. immigrant population is from Africa.

**Docs: …**- Sub-Saharan Africa 5 - Middle East and North Africa 4…

**Label: Neutral**

**Gemini-2.5 Reason**: Doc 1 provides a 2022 regional breakdown including `Sub-Saharan Africa 5%` and `Middle East and North Africa 4%`. **We cannot sum `Sub-Saharan Africa (5%)` with `Middle East and North Africa (4%)` to get an Africa total because the 4% includes non-African Middle Eastern origins**. Therefore, the references neither confirm nor refute that 5% of the U.S. immigrant population is from Africa. [Final Conclusion] The references lack the precise, matching metric needed to verify the claim.

**GPT-4o Hallucination Output**: The references explicitly state that immigrants from sub-Saharan Africa make up 5% of the U.S. immigrant population (Document 1) and that 5% of new arrivals between 2021 and 2023 were from sub-Saharan Africa (Document 2). This directly supports the claim that 5% of the U.S. immigrant population is from Africa.

**Claim:** Connected TV (CTV) ad spend is growing at 10.9%, while traditional TV ad spend is declining by 1.8%.

**Docs: …**Connected television ad spend is now forecast to grow by 10.9 whilst total television spend including broadcast television is expected to decline by 1.8…

**Label: Contradiction**

**Gemini-2.5 Reason**: "Total" spend is the sum of its parts. Since one part (Connected TV) is growing significantly at 10.9%, **the other part (traditional TV) must be declining by a rate greater than 1.8% for the overall total to decline by 1.8%**. Therefore, the claim incorrectly applies the decline figure for the *total* market to just the *traditional* TV market segment.

**GPT-5 Hallucination Output**: Doc 1 provides specific ratios (CTV +10.9%, total TV -1.8%), but lacks clear evidence supporting the -1.8% figure for traditional TV; therefore, this reference is insufficient to adequately substantiate its claim.

Figure 24: **Examples of inconsistencies on Faithfulness dimension**, correct assessments are marked in green.

## I EXAMPLES OF CASES WITH CONTENT ISSUES

This section shows prevalent content quality deficiencies in leading LLMs, illustrated through specific instances of contextual inconsistency, irrelevant response and factual hallucination. Fig. 25, 26, 27, and 28 illustrate issues with Reasonableness, Fig. 29 presents failures in Comprehensiveness, and Fig. 30 provides examples of Faithfulness errors. The problematic areas and their corresponding reasons are annotated in the figure. In each example, the erroneous area is highlighted, and an annotation explains the nature of the error.

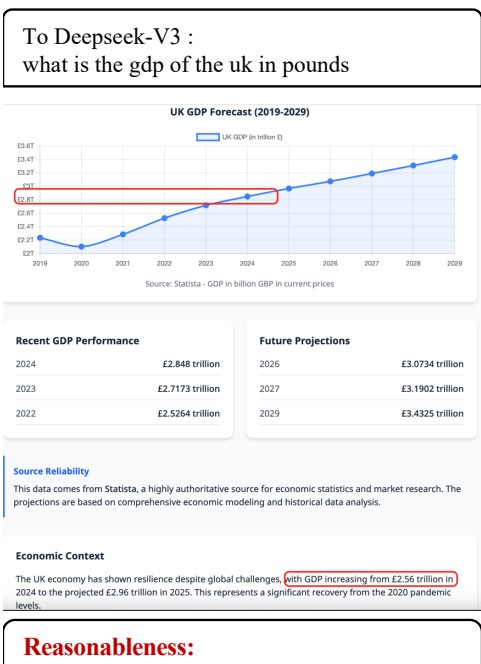

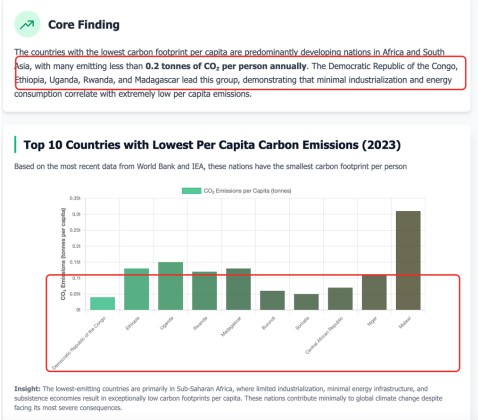

**Reasonableness:**
Issues Type: **Chart-text inconsistent**
Reason：The chart's underlying data for the year 2024 is 2.848 (trillion). However, the "Economic Context" section at the bottom states: "...GDP increasing from £2.56 trillion in 2024 to the projected £2.96 trillion in 2025. This is a direct contradiction.

**Reasonableness:**
Issues Type: **Chart-text inconsistent**
Reason：The "Core Finding" text states that "Democratic Republic of the Congo, Ethiopia, Uganda, Rwanda, and Madagascar lead this group". However, the chart data shows the top 5 lowest emitters are DRC (0.04), Somalia (0.05), Burundi (0.06), Central African Republic (0.07), and Niger (0.11). The countries mentioned in the text (other than DRC) are actually ranked 6th (Rwanda), 7th (Ethiopia), 8th (Madagascar), and 9th (Uganda). The summary text misrepresents the ranking presented in the chart, creating an internal contradiction.

Figure 25: **Examples of Reasonableness failures manifesting as chart-text inconsistencies.** The problematic areas along with the rationale are annotated.

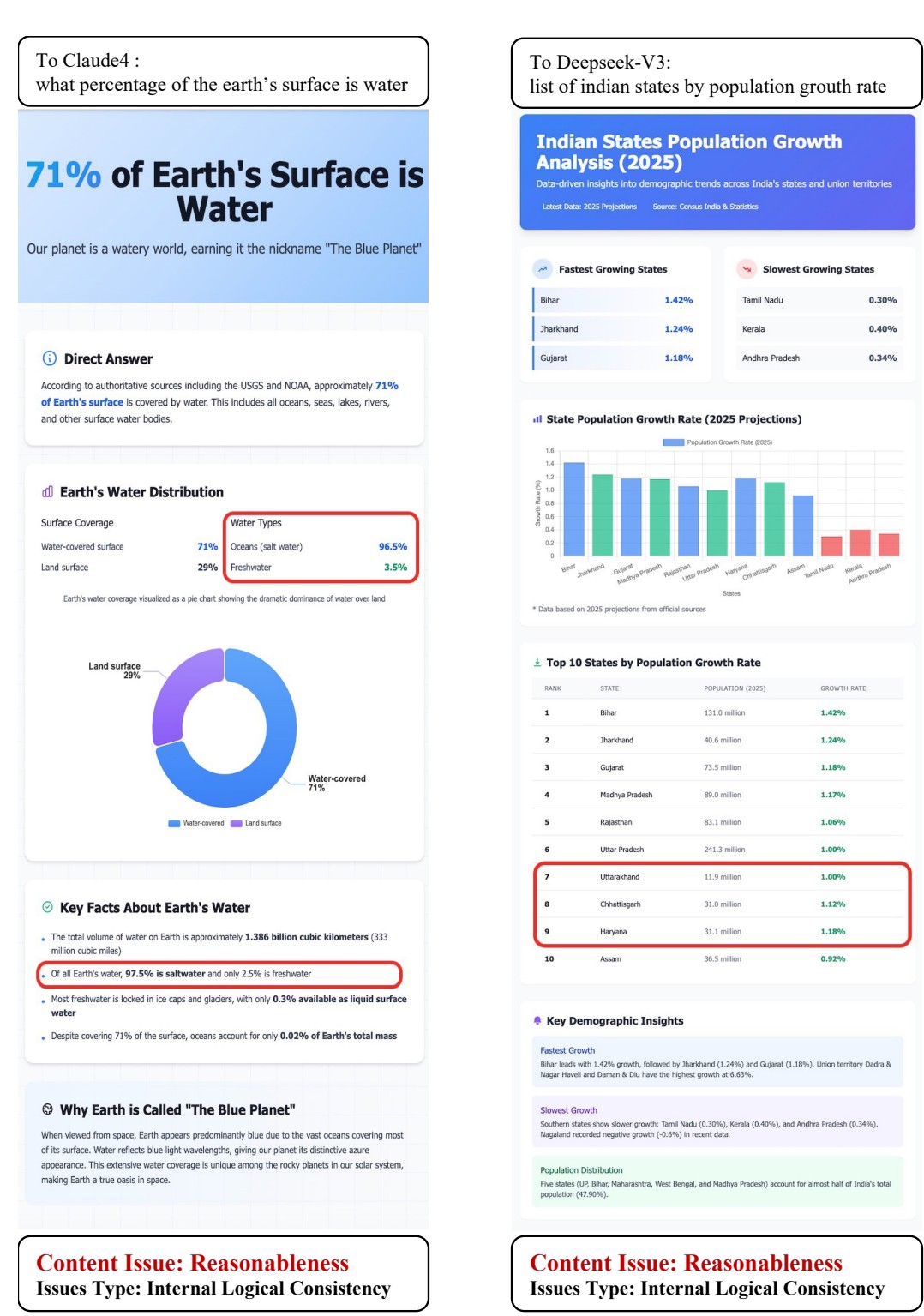

Figure 26: **Examples of Reasonableness failures manifesting as internal logical inconsistencies.** The contradictory parts of the text are highlighted with red boxes.

To Qwen3-235b-a22b-instruct-2507 :
what is the population of fayetteville north carolina

To Gemini-2.5-pro:
which country is experiencing the most rapid rate of population aging in the world

**Content Issue: Reasonableness**
**Issues Type: Chart- Numerical Plausibility**
**Reason:** The sum of percentages in a pie chart exceeds 100%

**Content Issue: Reasonableness**
**Issues Type: Internal Logical Consistency**
**Reason:** The response contradicts itself by stating that Japan's elderly population is both "over a third" (which is >33.3%) and specifically "28.2%"

Figure 27: **Examples of Reasonableness failures manifesting as numerical errors and internal logical inconsistencies.** The problematic areas along with the rationale are annotated.

To Deepseek-V3 :
where does the UK get its electricity from

**Energy Transition Timeline**

**1990s**
"Dash for Gas" transition from coal to natural gas generation

**2017**
First full day without coal power since the Industrial Revolution

**2019**
First coal-free week in over a century

**2024**
Final coal-fired power station (Ratcliffe-on-Soar) closed

**Regional Energy Strategies**

| Scotland | UK National |
|---|---|
| Targets 80% renewable electricity by 2020, with emphasis on wind and hydro power. Cut greenhouse gas emissions by 46% between 1990-2014. | Committed to reducing CO2 emissions through the Low Carbon Transition Plan, aiming for 30% renewable electricity by 2020 (exceeded with 38.9% in Q3 2019). |

**Reasonableness:**
**Issues Type: Tense Confusion**
**Reason: Using the present tense "Targets" for a goal with a deadline in the past (2020) is temporally illogical. It should be phrased in the past tense (e.g., "targeted" or "had a target of")**

To Llama3 :
where is the most oil produced in the US

**Oil Production in the US**
The United States is one of the top crude oil producers in the world. In 2022, the top five crude oil-producing states in the US were Texas, New Mexico, North Dakota, Colorado, and Alaska.

**Reasonableness:**
**Issues Type: Missing Chart Elements**
**Reason: The y-axis is missing its unit label.**

Figure 28: **Examples of Reasonableness failures manifesting as tense confusion and missing chart elements.** The problematic areas along with the rationale are annotated.

To Llama-3.3-70B-Instruct :
distribution of free-throw percentages among WNBA players 2023 season

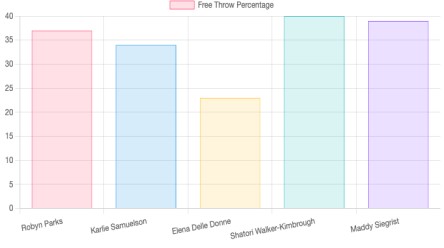

WNBA 2023 Free Throw Percentage Distribution

The following chart shows the distribution of free throw percentages among WNBA players in the 2023 season.

The data shows that the top 5 players with the highest free throw percentages in the 2023 WNBA season are Robyn Parks, Karlie Samuelson, Elena Delle Donne, Shatori Walker-Kimbrough, and Maddy Siegrist.

**Comprehensiveness**:
**Reason:** In The response fundamentally misunderstands the user's request for a "distribution" and provides wildly inaccurate data. It is misleading and unhelpful.

To Llama-4-Scout-17B-16E-Instruct:
Details of all F1 Grand Prix races for the 2025 season

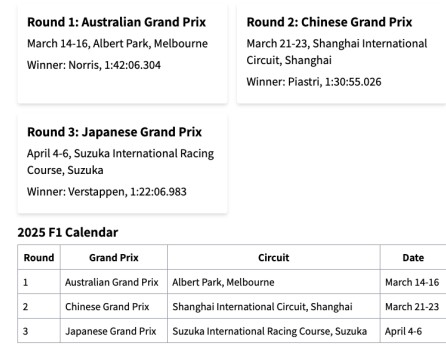

**2025 F1 Grand Prix Races**

The 2025 FIA Formula One World Championship is an ongoing motor racing championship for Formula One cars and the 76th running of the Formula One World Championship.

**Round 1: Australian Grand Prix**
March 14-16, Albert Park, Melbourne
Winner: Norris, 1:42:06.304

**Round 2: Chinese Grand Prix**
March 21-23, Shanghai International Circuit, Shanghai
Winner: Piastri, 1:30:55.026

**Round 3: Japanese Grand Prix**
April 4-6, Suzuka International Racing Course, Suzuka
Winner: Verstappen, 1:22:06.983

**2025 F1 Calendar**

| Round | Grand Prix | Circuit | Date |
|---|---|---|---|
| 1 | Australian Grand Prix | Albert Park, Melbourne | March 14-16 |
| 2 | Chinese Grand Prix | Shanghai International Circuit, Shanghai | March 21-23 |
| 3 | Japanese Grand Prix | Suzuka International Racing Course, Suzuka | April 4-6 |

**Comprehensiveness**:
**Reason:** The user asked for details of all F1 races for the 2025 season, but the response only lists 3 out of the 24 scheduled races.

Figure 29: **Examples of cases with Comprehensiveness Issues.**

To GPT-4o :
canada's top 5 trading partners imports and exports

**Canada's Top 5 Trading Partners**

**Overview**
Canada's top 5 trading partners are the United States, China, United Kingdom, Japan, and Mexico. These countries account for the majority of Canada's exports and imports.

**Exports**
Canada's top exports include mineral fuels, vehicles, machinery, gems, and precious metals. The United States is Canada's largest export partner, accounting for 76.88% of total exports.

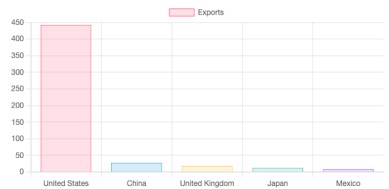

**Imports**
Canada's top imports include vehicles, machinery, electrical equipment, and plastics. The United States is also Canada's largest import partner, accounting for 49% of total imports.

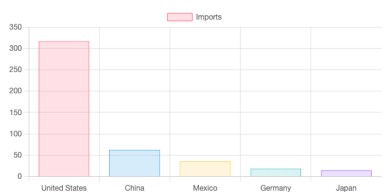

**Trade Balance**
Canada had a trade surplus of $6.6 billion in 2021, with exports totaling $636 billion and imports totaling $631 billion.

**Content Issue: Faithfulness**
**Claim:** In 2024, Canada's imports from Japan were valued at $13.14 billion, representing 2.30% of total imports.
**Label: Contradiction**
**Docs**: <page id=”3" … **2022** Canada top 5 Export and Import partners … **Japan 138222.32** </page>
<page id="9" … Canada Imports By Country Value Year … **Japan 15.54B 2024** </page>

To Llama-4-Scout-17B-16E-Instruct:
list of densely populated countries in the world

**List of Densely Populated Countries**

The following is a list of countries and territories by population density, based on the number of people per square kilometer.

| Rank | Country | Population Density (people/km²) | Population | Area (km²) |
|------|---------|-------------------------------|------------|------------|
| 1 | Macau | 21,946 | 683,972 | 32.9 |
| 2 | Monaco | 19,171 | 39,241 | 2.02 |
| 3 | Singapore | 8,177 | 5,690,000 | 720.5 |
| 4 | Hong Kong | 7,044 | 7,519,000 | 1,106 |
| 5 | Gibraltar | 5,901 | 33,701 | 6.8 |
| 6 | Bahrain | 2,093 | 1,463,330 | 778 |
| 7 | Maldives | 1,766 | 436,000 | 300 |
| 8 | Malta | 1,704 | 475,701 | 315 |
| 9 | Bangladesh | 1,350 | 163,046,161 | 147,570 |
| 10 | Sint Maarten | 1,293 | 41,220 | 34 |

Visualization of Population Density

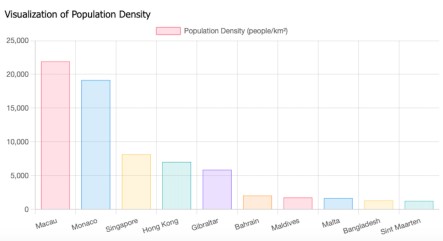

**Content Issue: Faithfulness**
**Claim:** Ranked 1st in population density, Macau has 21,946 people per square kilometer, a total population of 683,972.
**Label: Contradiction**
**Docs**: <page id="4"…> …
<synopsis>Highest population density by country 2024 **Monaco led the ranking for countries with the highest population density in 2024**, with nearly 26,000 ...</synopsis> … </page>

Figure 30: **Examples of Faithfulness issues** where the generated answers contain contradicts information from the reference document.

## J EXAMPLES OF RESPONSES FROM DEEP RESEARCH AGENTS AND RAVIG

In this section, we compare the responses generated by Gemini Deep Research Google (2025), Tongyi Deep Research Team et al. (2025), and our Gemini-2.5-Pro Comanici et al. (2025) with RAViG for the same query. As shown in Fig. 31 and 33, for everyday conversational queries, our end-to-end retrieval-augmented visually-rich generation (RAViG) is comparable to leading Deep Research products in both content quality and page design, while its response time ($\approx$ 30s) is 10-20x faster than Deep Research ($\approx$ 10m). On the other hand, the extremely long responses produced by Tongyi Deep Research (Fig. 32 and 34) are advantageous for in-depth analytical tasks, but may be less suitable for casual, chat-style queries. This contrast highlights the different application scenarios and positioning of these two types of systems.

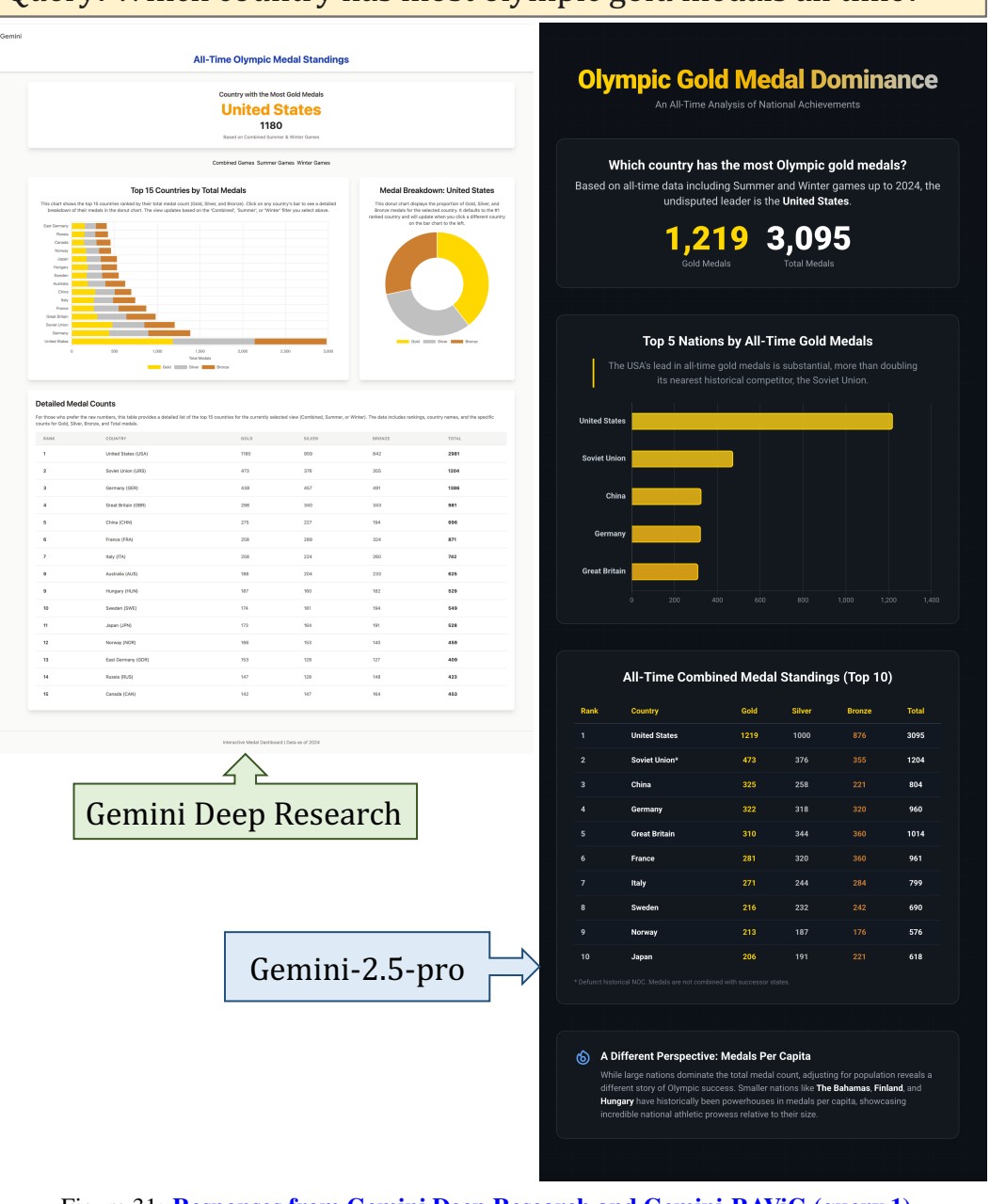

Figure 31: **Responses from Gemini Deep Research and Gemini-RAViG (query 1).**

Figure 32: **Responses from Tongyi Deep Research (query 1).**

Query: Details of all major stock exchanges globally and their operating hours?

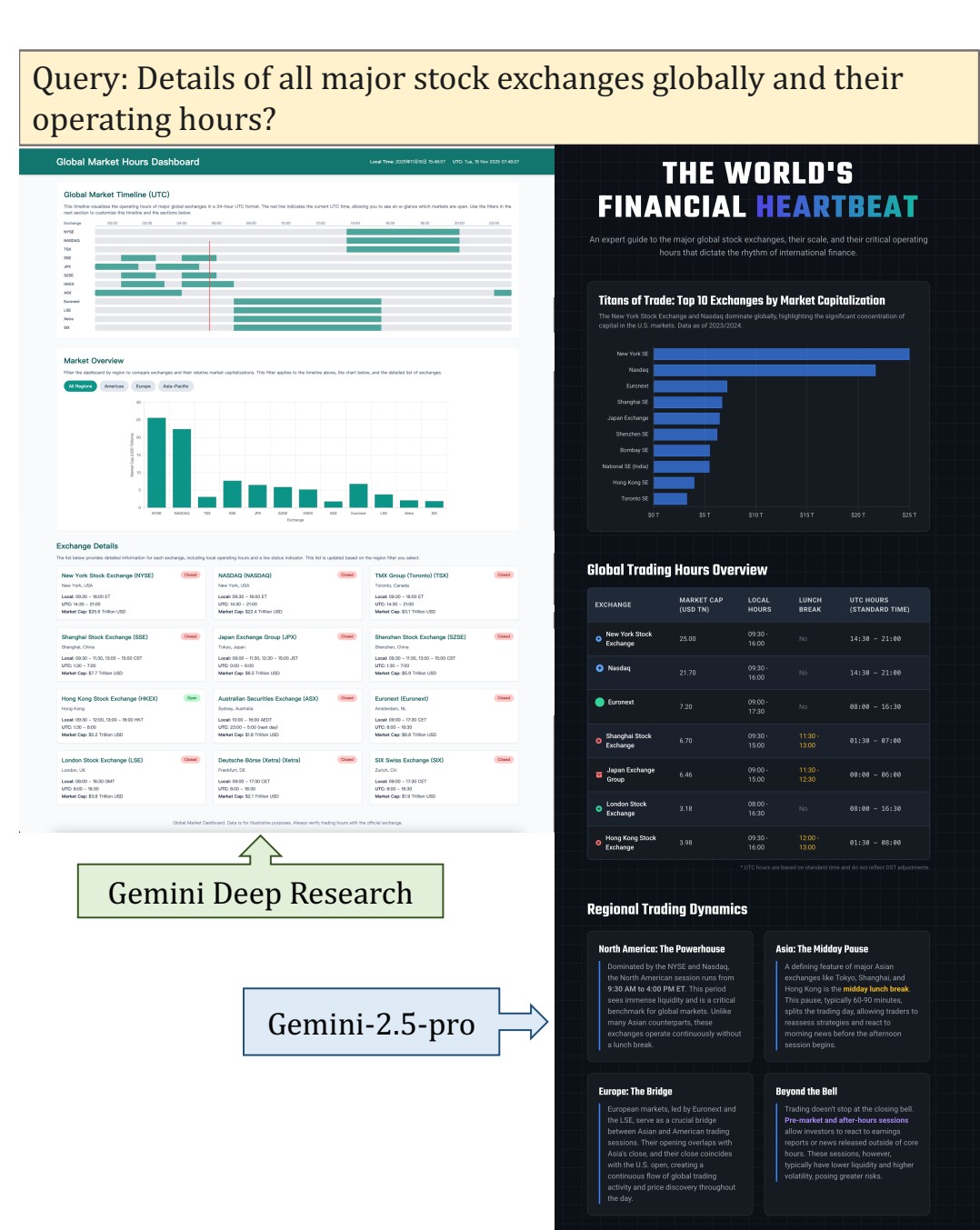

Figure 33: **Responses from Gemini Deep Research and Gemini-RAViG (query 2).**

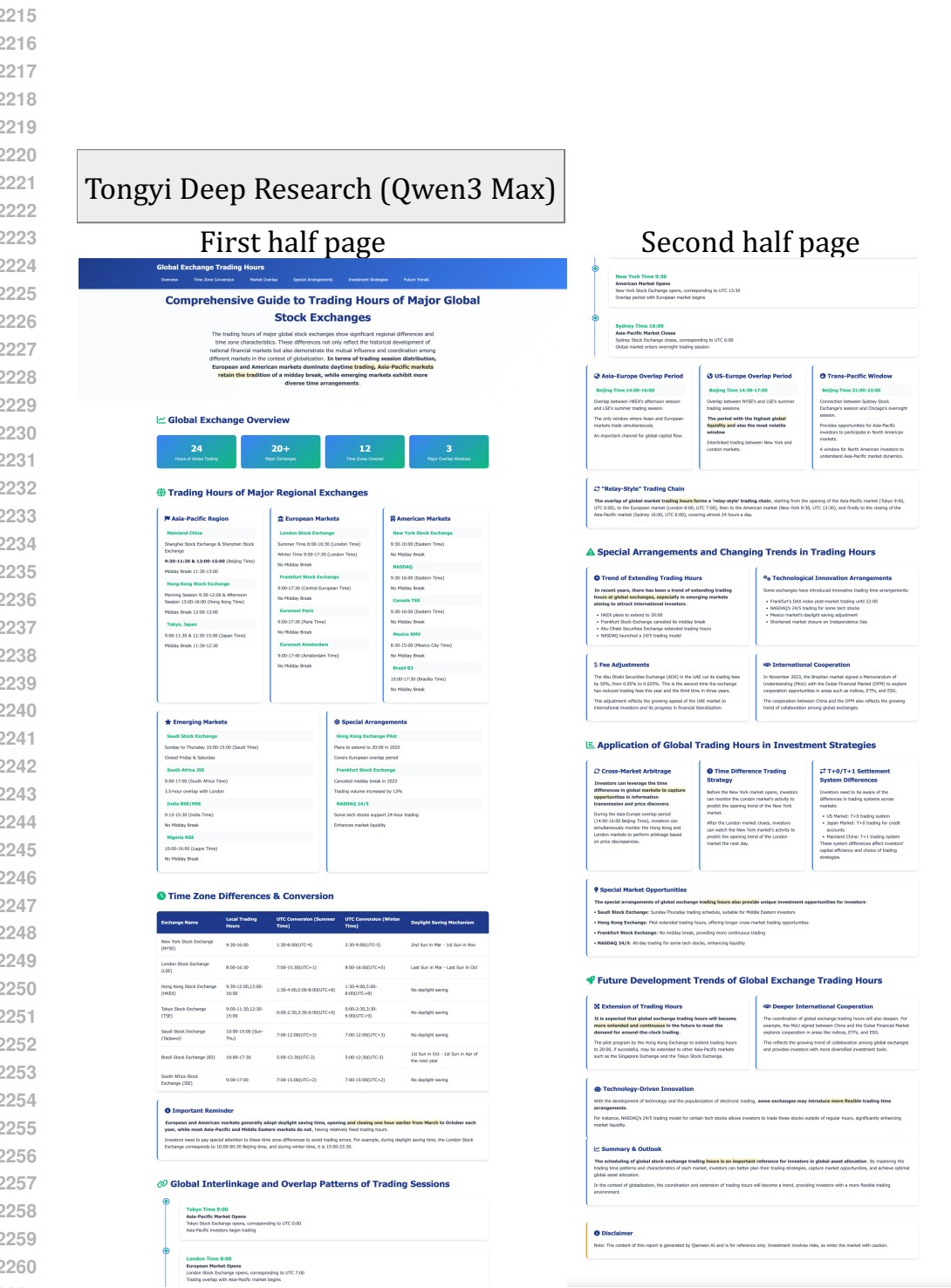

Figure 34: **Responses from Tongyi Deep Research (query 2).**

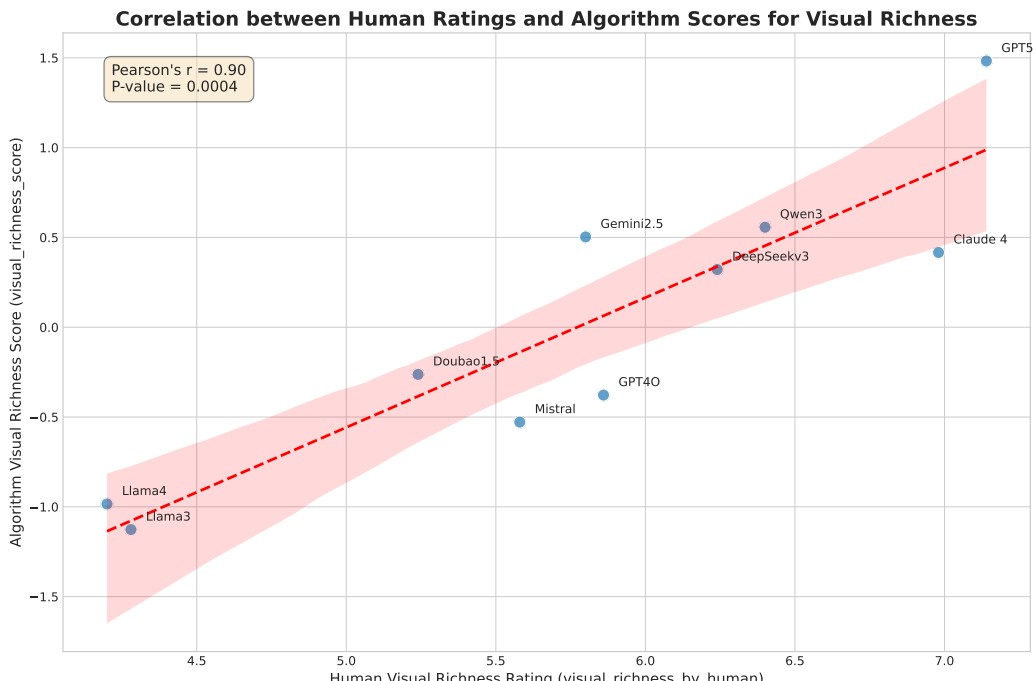

Figure 35: **Correlation bewteen human ratings and our $VC_{\text{score}}$.**

## K   HUMAN ANNOTATION OF VISUAL RICHNESS

We randomly selected 100 queries. The corresponding responses from each model were then rated by three experts on a 5-point scale (2, 4, 6, 8, 10) based on their subjective assessment of the visual richness of each answer. The final score for each response is the average of the three expert ratings. As shown in Fig. 35, A Pearson correlation analysis between our visual richness score $VC_{\text{score}}$ and the averaged human ratings shows a strong, positive, and statistically significant correlation (r=0.90, p=0.0004). This indicates that our metric effectively captures the structural aspects of visual richness as perceived by humans.

## L   SENSITIVITY ANALYSIS OF THE SCORE CORRECTION MECHANISM

We analyzed how model rankings were affected by changes to key hyperparameters: $\alpha$ and the internal weights of $VC_{score}$. Firstly, we analyzed the effect of varying $\alpha$ from 0.1 to 1.0. As expected, model rankings showed moderate sensitivity to the penalty's strength (Fig. 36). Crucially, this analysis revealed that our chosen value of $\alpha = 0.3$ falls within an optimal range (0.1–0.3) where the resulting rankings best align with human preferences. Secondly, we fixed the weight of table $t_w$ and varied $m_w$ from 0 to 0.9. We found that model rankings were highly robust to $m_w$, with Kendall's $\tau > 0.95$ for the vast majority of its tested range (Fig. 37) This demonstrates that the overall signal of visual richness is strong enough that the final outcome is not dependent on the specific fixed weights.

## M   MORE ABOUT HUMAN-MACHINE AGREEMENT

### M.1   GLOSSARY FOR HUMAN-MACHINE AGREEMENT EVALUATION

#### M.1.1   FLEISS' KAPPA

Fleiss' Kappa ($\kappa$) is a statistical measure that assesses the agreement reliability between three or more raters on categorical ratings (e.g., "correct" vs. "incorrect"). It quantifies the degree of agree-

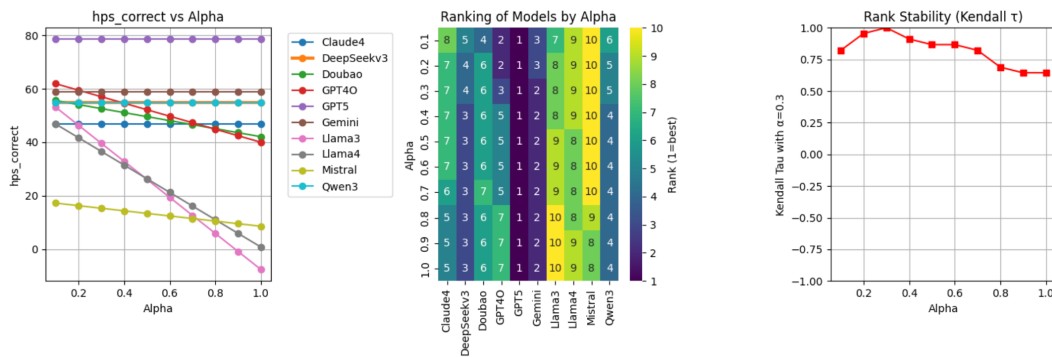

Figure 36: **Changes in model scores and rankings under different $\alpha$ values.**

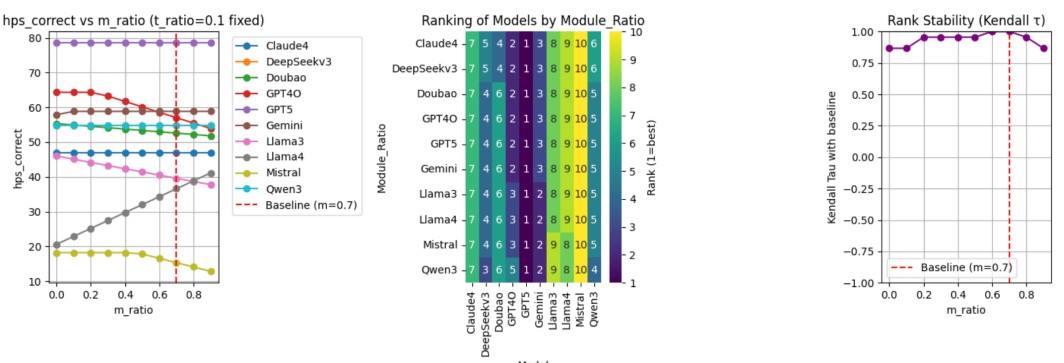

Figure 37: **Changes in model scores and rankings under different $m_{\mathbf{w}}$ values.**

ment that is above what would be expected by chance. The value of $\kappa$ ranges from -1 to 1, where higher values indicate stronger agreement. We interpret the scores using established benchmarks:

- $\kappa \leq 0$: Poor agreement.
- $0.01 \leq \kappa \leq 0.20$: Slight agreement.
- $0.21 \leq \kappa \leq 0.40$: Fair agreement.
- $0.41 \leq \kappa \leq 0.60$: Moderate agreement.
- $0.61 \leq \kappa \leq 0.80$: Substantial agreement.
- $0.81 \leq \kappa \leq 1.00$: Almost perfect agreement.

### M.1.2 INTRACLASS CORRELATION COEFFICIENT (ICC)

The Intraclass Correlation Coefficient (ICC) is a descriptive statistic used to measure the consistency of measurements made by multiple observers on the same set of subjects. It is particularly well-suited for continuous or ordinal data (e.g., scores from 1 to 10). The ICC value typically falls within the range of 0 to 1. Higher values signify greater reliability and consistency among raters.

- ICC $< 0.5$: Poor reliability.
- $0.5 \leq$ ICC $< 0.75$: Moderate reliability.
- $0.75 \leq$ ICC $< 0.9$: Good reliability.
- ICC $\geq 0.9$: Excellent reliability.

### M.1.3 SPEARMAN'S RANK CORRELATION COEFFICIENT (SRCC)

Spearman's Rank Correlation Coefficient ($\rho$) is a non-parametric measure of the monotonic relationship between two variables. It assesses how well the relationship between two variables can be

described using a monotonic function, making it ideal for ordinal data. The value of $\rho$ is constrained to the interval $[-1, 1]$.

- $\rho = +1$: Indicates a perfect positive monotonic correlation.

- $\rho = -1$: Indicates a perfect negative monotonic correlation.

- $\rho = 0$: Indicates the absence of a monotonic correlation.

- The magnitude $|\rho|$ indicates the strength of the monotonic relationship.

### M.1.4 CLASSIFICATION PERFORMANCE METRICS

We evaluate classification performance using the standard metrics of Accuracy, Precision, and Recall, which are based on the counts of true positives (TP), false positives (FP), true negatives (TN), and false negatives (FN).

- **Accuracy** measures the proportion of all correct classifications.

- **Precision** measures the proportion of true positives among all instances predicted as positive.

- **Recall** measures the proportion of true positives among all actual positive instances.

### M.2 ADDITIONAL RESULTS AND ANALYSIS ON HUMAN-MACHINE AGREEMENT

This part provides more detailed results of the agreement between our automated evaluation framework and human expert judgments in Tab. 7.

| Metric | Design Quality | | | | Content Quality | | |
|---|---|---|---|---|---|---|---|
| | Missing | Occlusion | Color | Overflow | Reasonable | Faithfulness | Comprehensiveness |
| Fleiss' Kappa | 0.980 | 0.966 | 0.853 | 0.927 | 0.852 | 0.833 | - |
| Accuracy | 0.975 | 0.940 | 0.945 | 0.950 | 0.930 | 0.940 | - |
| Precision | 0.990 | 0.980 | 0.990 | 0.960 | 0.925 | 1.000 | - |
| Recall | 0.961 | 0.907 | 0.908 | 0.941 | 0.942 | 0.945 | - |
| ICC(3,k) | - | - | - | - | - | - | 0.8883 |
| SRCC | - | - | - | - | - | - | 0.7770 |

Table 7: Detailed human-machine agreement metrics across dimensions

**Analysis of the Results** The high Fleiss' kappa (all $\geq 0.8$) and ICC scores (=0.8883) demonstrate strong agreement among human experts and the reliability of the human evaluation results. The corresponding Accuracy, Precision, and Recall metrics reflect the effectiveness of our automatic evaluation framework.

## N ANALYSIS OF JUDGE MODEL SELF-PREFERENCE

To validate the objectivity of our LLM-as-a-Judge evaluation framework, we conducted an analysis to investigate potential self-preference in our primary judge models on comprehensiveness scores: GPT-5 and Gemini-2.5-Pro. We chose comprehensiveness because it is the scoring dimension most susceptible to subjectivity. In this dimension, models are required to assign a score from 1 to 10 based on the perceived coverage and completeness of the answer. In contrast, the other dimensions are binary (0 or 1) and simply judge correctness, for which we have already validated accuracy. Self-preference occurs if a judge model unfairly favors outputs generated by itself or models from its own family (for example, GPT-5 favors GPT series).

Our methodology involved comparing the scores assigned by GPT-5 and Gemini-2.5-Pro to the outputs from a diverse set of ten models. A low correlation or systematic deviation in scores could indicate bias. Conversely, a strong linear correlation would suggest that both judges evaluate outputs based on similar, objective criteria, thus minimizing the impact of self-preference.

The evaluation scores assigned by the two judge models are detailed in Table 8.

| Model Under Evaluation | GPT-5 Score | Gemini-2.5-Pro Score |
| --- | --- | --- |
| Claude4 | 63.83 | 81.22 |
| doubao1.5 | 59.73 | 72.20 |
| DeepSeek-v3 | 61.60 | 74.79 |
| Gemini2.5 | 69.28 | 86.17 |
| GPT4O | 61.98 | 71.97 |
| GPT5 | 80.21 | 93.30 |
| Llama3 | 56.86 | 63.54 |
| Llama4 | 56.69 | 61.58 |
| Mistral | 58.55 | 68.44 |
| Qwen3 | 62.49 | 75.49 |

Table 8: Comprehensiveness scores assigned by judge models GPT-5 and Gemini-2.5-Pro.

To assess the relationship visually and quantitatively, we plotted the average query-level scores from each judge against one another. The resulting scatter plot illustrates the degree of correlation between their evaluations, as shown in Figure 38.

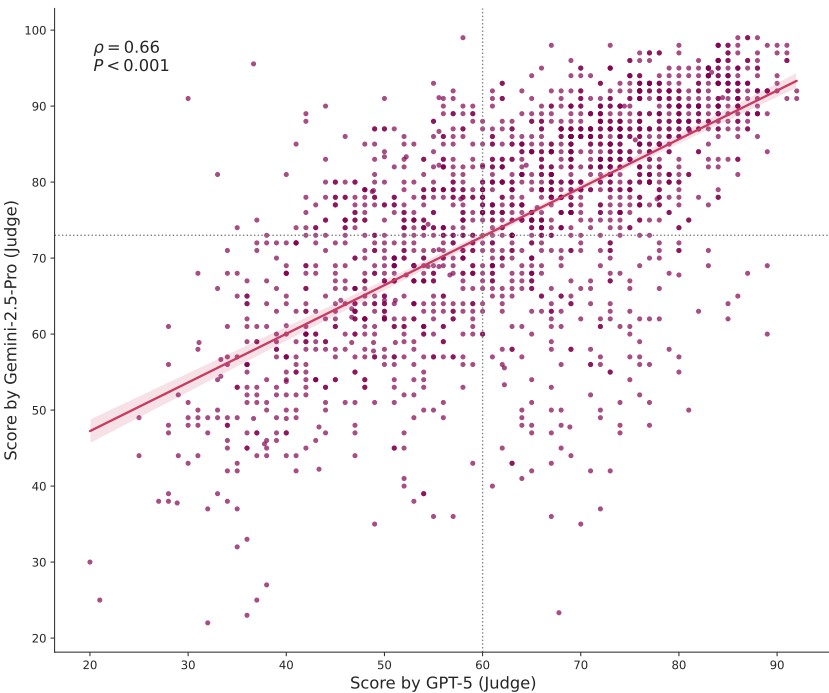

Figure 38: Correlation of comprehensiveness scores assigned by GPT-5 and Gemini-2.5-Pro. The solid line represents the linear regression fit the strong positive correlation (Pearson's $r = 0.66$) suggests a high degree of agreement between the two judge models.

As illustrated in Figure 38, the comprehensiveness scores assigned by GPT-5 and Gemini-2.5-Pro exhibit a **strong and significant positive linear correlation**. The data points cluster tightly around the linear regression line, indicating a high degree of agreement in their relative ranking of the models. Quantitatively, the **Pearson correlation coefficient ($r$) is 0.66** ($p < 0.001$), which confirms the visual observation. This high level of inter-judge agreement suggests that both models are leveraging similar underlying criteria to assess the quality of the outputs.

While we cannot entirely rule out subtle biases, the strong cross-model consistency indicates that **systematic self-preference is not a dominant factor** in our evaluation. For example, GPT-5 assigns the highest score to its own output, but Gemini-2.5-Pro concurs, also giving GPT-5 the top score. This consistency across the board bolsters the credibility and objectivity of our LLM-as-a-Judge results, supporting its use as a reliable evaluation method for this task.

| Judge Configuration | HPS(paper, GPT4o+Gemini2.5) | HPS(Gemini2.5+Gemini2.5) | HPS(GPT4o+GPT5) |
|---|---|---|---|
| GPT5 | 73.54 | 67.71 | 71.67 |
| Gemini2.5 | 51.56 | 47.61 | 51.91 |
| GPT4o | 64.77 | 63.08 | 67.81 |
| Qwen3 | 51.12 | 40.86 | 52.61 |
| DeepSeekv3 | 51.03 | 48.51 | 57.17 |
| Doubao1.5 | 56.68 | 51.94 | 55.38 |
| Claude 4 | 45.86 | 39.71 | 46.01 |
| Llama3 | 58.52 | 49.61 | 60.38 |
| Mistral | 16.19 | 15.00 | 17.13 |
| Llama4 | 46.58 | 41.64 | 51.56 |

Table 9: Robustness Analysis Across Different Judge Configurations.

## O  ROBUSTNESS AND BIAS ANALYSIS OF THE LLM-AS-A-JUDGE PROTOCOL

Our framework is designed to minimize subjectivity. As detailed in the paper, all LLM-judged dimensions (except for Comprehensiveness) **rely on a binary "Yes/No" judgment based on strict criteria, not open-ended scoring**. This design inherently limits the potential for judge-specific bias. Appendix N reports a self-preference analysis for Comprehensiveness, showing no systematic self-preference bias. The LLM-as-a-judge paradigm itself is also well-established in prior work Zheng et al. (2023); Chen et al. (2025).

To demonstrate the robustness of our evaluation framework, we re-evaluated the responses from ten models to 100 randomly sampled queries (1,000 responses in total) *using a cross-family LLM-judge configuration*. As shown in Tab. 9 ("Model A + Model B" denotes Model A for design, Model B for content):

1) *High Ranking Stability*: The relative model rankings are highly stable across all judge configurations.

2) *No Self-Preference Bias*: When Gemini-2.5 acts as judge, it still scores its own performance significantly below GPT-5. Similarly, GPT-based judges do not artificially inflate the scores of other GPT models.

These results suggest that **our evaluation is not dominated by judge-specific biases, and the comparative ranking of models is robust**.

## P  ROBUSTNESS BETWEEN DETERMINISTIC AND MILDLY STOCHASTIC DECODING SETTINGS

We conducted an additional experiment to compare model performance under deterministic decoding (in our paper) versus a mildly stochastic decoding setting (temperature=0.7, top_p=0.9) on random sample of 100 queries. As shown in Tab. 10:

1) *Overall scores and rankings are stable*: The model rankings demonstrate high stability across both settings. This confirms the robustness and reliability of our evaluation framework.

2) *Larger fluctuations are driven by generation-side variability*: For instance, Gemini-2.5-pro's DSR improved notably. Our analysis traces this to a large reduction in "oversized elements" issues (from 25% to 13%), which comes from stochastic decoding helping the model avoid a deprecated CSS link that it consistently defaults to when temperature=0.

## Q  ADDITIONAL EXPERIMENTAL RESULTS AND ANALYSIS

In this section, we provide the detailed result for our experiment.

| Metric | Paper Setting | | | Temp=0.7 | | |
|--------|-----|------|------|-----|------|------|
| | DSR | ECQ | HPS | DSR | ECQ | HPS |
| GPT5 | 0.77 | 95.5 | 73.5 | 0.76 | 94.1 | 71.5 |
| Gemini2.5 | 0.61 | 84.5 | 51.6 | 0.77 | 81.2 | 62.5 |
| GPT4o | 0.92 | 70.4 | 64.8 | 0.94 | 68.6 | 64.5 |
| Qwen3 | 0.73 | 70.0 | 51.1 | 0.66 | 73.9 | 48.8 |
| DeepSeekv3 | 0.79 | 64.6 | 51.0 | 0.84 | 71.1 | 59.8 |
| Doubao1.5 | 0.74 | 76.6 | 56.7 | 0.72 | 74.4 | 53.5 |
| Claude 4 | 0.66 | 69.5 | 45.9 | 0.59 | 73.7 | 43.5 |
| Llama3 | 0.91 | 64.3 | 58.5 | 0.90 | 63.4 | 57.1 |
| Mistral | 0.25 | 64.8 | 16.2 | 0.29 | 55.4 | 16.1 |
| Llama4 | 0.73 | 63.8 | 46.6 | 0.73 | 64.0 | 46.7 |

Table 10: Comparison of Model Performance under Deterministic (Paper Setting) and Stochastic (Temperature=0.7) Decoding.

## Q.1 DETAILED RESULT FOR DESIGN QUALITY TEST

Tab. 11 reports the number of issue cases for five types of design flaws across all models. The count of design quality failures is not equal to the sum of issue cases across dimensions because a single failure may involve multiple types of design flaws.

| Error Type | GPT5* | Gemini2.5* | GPT4O* | Qwen3 | DeepSeekv3 | Doubao1.5* | Claude4* | Llama3 | Llama4 | Mistral |
|------------|-------|-----------|--------|-------|-----------|-----------|----------|--------|--------|---------|
| Oversized Element | 145 | 505 | 60 | 80 | 58 | 18 | 233 | 0 | 1 | 2 |
| Missing Element | 11 | 13 | 36 | 33 | 34 | 68 | 39 | 151 | 187 | 103 |
| Occlusion | 13 | 22 | 21 | 48 | 59 | 146 | 25 | 45 | 21 | 77 |
| Overflow | 28 | 15 | 21 | 69 | 98 | 283 | 16 | 4 | 21 | 22 |
| Low Color Contrast | 131 | 30 | 53 | 182 | 65 | 20 | 386 | 12 | 6 | 1196 |
| **Design Quality Failures** | 320 | 555 | 171 | 337 | 237 | 439 | 620 | 207 | 227 | 1295 |

Table 11: Detailed of five design error types across all models

## Q.2 DETAILED RESULT FOR CONTENT QUALITY TEST

This part presents a detailed visualization of the content quality evaluation results, further exploring the relationships between different performance metrics and model characteristics like output length. The figures below provide a granular analysis of model capabilities.

**Analysis of Figure 39(a)** Figure 39(a) reveals that a longer output does not necessarily correlate with higher quality. While Comprehensiveness shows a weak positive correlation with render length, this trend is not absolute. For example, the increase in output length from Llama4 to DeepSeekv3 resulted in little improvement in the comprehensiveness score, and even showed a drop in the middle (Mistral). Furthermore, Reasonableness and the overall Effective Content Quality (ECQ) show little to no correlation with output length. Models such as GPT5 maintain high scores with long outputs, while others like Claude4 exhibit poor performance despite similar lengths. This demonstrates that our evaluation metrics effectively resist length bias and prioritize the quality of the content.

**Analysis of Figure 39(b)** Figure 39(b) provides a more granular view of Faithfulness scores. As highlighted in the main text, a significant drop is observed from the claim-level scores to the more stringent case-level scores for most models. This underscores a critical challenge: even a high accuracy on individual facts (claim-level) does not guarantee a fully trustworthy response (case-level), as a single error can undermine the entire output's reliability.

**Analysis of Figure 39(c)** Figure 39(c) illustrates the performance of leading LLMs on the Effective Content Quality (ECQ) score, across a spectrum of reference text lengths, ranging from under 10k to over 40k tokens. The results reveal a clear performance hierarchy, with GPT5 establishing a commanding lead. It consistently achieves an ECQ score above 90 across all token ranges, significantly outperforming its competitors. In contrast, some models exhibit two distinct patterns: a lower performance ceiling and significant performance volatility. For instance, Llama4 and Qwen3 experience a dramatic drop-off, with Llama4's score plummeting to 39.74 in the longest context window. This steep degradation suggests a failure to manage information effectively over extended

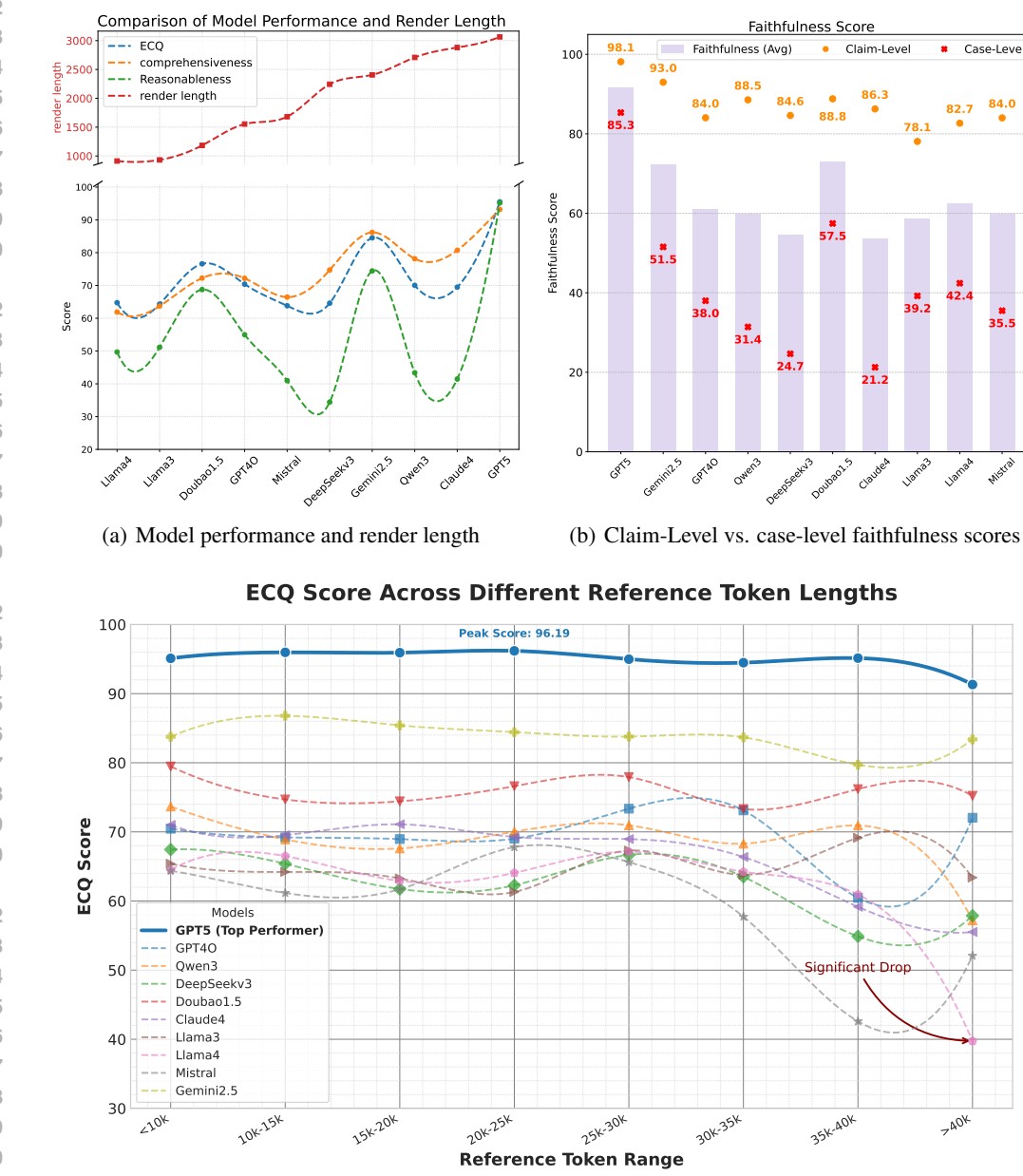

(a) Model performance and render length

(b) Claim-Level vs. case-level faithfulness scores

(c) Model performance on ECQ score And Reference Tokens Range

Figure 39: Detailed visualization of content quality evaluation.

sequences, a common challenge known as "long context loss". Interestingly, this failure is not universal: models like Gemini2.5 and Doubao1.5 exhibit a performance rebound in this same range, pointing to diverse and variably effective strategies for handling extreme context lengths.

