# OpenReview forum: "RAViG-Bench: A Benchmark for Retrieval-Augmented Visually-rich Generation with Multi-modal Automated Evaluation"
_ICLR.cc/2026/Conference — Submitted to ICLR 2026_

### Official Review · Reviewer_QJiz · 2025-10-28

**Soundness:** 3
**Presentation:** 3
**Contribution:** 2
**Rating:** 2
**Confidence:** 3

**Summary:**

This paper introduces a new task named Retrieval-Augmented Visually-rich Generation (RAViG), which involves automatically generating HTML-style reports that integrate multiple visualizations (e.g., charts, tables) with explanatory text, based on retrieved web documents.

The authors highlight the gap in existing benchmarks: 1) RAG benchmarks primarily evaluate text-only outputs, and 2) NL2VIS (Natural Language to Visualization) benchmarks assume the underlying data is already provided, consequently, they fail to cover the unique failure modes of this task, such as HTML rendering errors, design flaws, text-visual misalignment, and data hallucinations.
To address this problem, the authors constructed the RAViG-Bench dataset and proposed a comprehensive, automated evaluation framework to assess LLM performance on this task.

**Strengths:**

1. Comprehensive Benchmark：As a benchmark paper, the authors establishes a complete framework for this new RAViG task, comprising a dataset, detailed task definitions, and an end-to-end evaluation pipeline.

2. Multi-faceted Evaluation Framework:
The proposed evaluation framework is comprehensive, employing a assessment from various aspects like Functionality, Design, and Content, that progressively filters and evaluates model outputs.

**Weaknesses:**

1. Limited Task Novelty: Even the authors discuss a lot about related works in RAG and NL2Vis, actually RAViG task closely overlaps with existing ''deep research'' AI agents/task/benchmark (e.g., OpenAI Deep Research [1], Tongyi DeepResearch [2]) and Data Insight Agent/Task (e.g., InsightBench [3]) that also generate reports with visualization elements (i.e., various analysis chart with explaining text) from web retrieval or multiple data sources (e.g. databases). Requiring HTML output doesn't fundamentally differentiate it, undermining its claimed innovation. For example, if we requires a Deep Research Agent’s output to be restricted to a HTML format by adding this requirement statement in the input of the agents, then the general DeepResearch task becomes this RAViG task.

2. Questionable Task Design: Compared with the general Deep Research task, which requires the LLM to act like a data analyst with searching, reasoning and report generation capability, this RAViG task performs like forcing the LLM to act as both a data analyst and HTML engineer, which is inefficient and a bit unreasonable for real-world applications. These two steps could be effectively solved by separate agents/tasks—one generating a visualized report (not necessarily in HTML, like the general deepresearch task) and another one acting like a coding agent which converting the report into other formats (PDF, HTML, markdown, etc.). Combining them into a single task offers no clear additional benefit.

[1] https://openai.com/index/introducing-deep-research/

[2] https://github.com/Alibaba-NLP/DeepResearch

[3] https://insightbench.github.io/

**Questions:**

I notice that RAViG-Bench does not provide a direct Ground-Truth visual output for each query against which to compare model responses. Instead, it relies heavily on an LLM-as-a-Judge paradigm combined with rule-based checks for evaluation.

Could you discuss the potential reasons/implications of this design? e.g.,, does the absence of a fixed ground-truth and the high dependence on the LLM-based judges make the scores susceptible to judge-specific biases? This concern seems particularly relevant when the judge model is similar in architecture or training data to the models being evaluated.

Furthermore, without a concrete ground-truth, how can the benchmark ensure precise and objective accuracy measurement, rather than just a relative score?

---

> ### Author Response · Authors · 2025-11-20
>
> We thank reviewer Qjiz for the  thoughtful and valuable feedback. We address the raised concerns in detail below.
>
> ------
>
>
> ## **W1: Clarification on Task Novelty and Our Core Contribution**
>
> We agree that generating visual reports is an emerging task, as acknowledged in our introduction with existing systems [1,2]. Our primary innovation, however, is not the task's definition but **the first comprehensive benchmark and automated framework to evaluate it.** Any agent capable of generating an HTML report can be benchmarked by our framework.
>
> ### **Comparison with InsightBench**
>
> 1) ***Input & Task***: InsightBench tests **data analytics** on **structured datasets**. RAViG-Bench tests **information synthesis and web design** from **unstructured, noisy web pages**.
>
> 2) ***Visualization Requirements***:
> The visualization requirement of InsightBench is similar to prior NL2VIS tasks (as described in their paper: *"Each question has a Python code block that generates **a plot** to answer the question, and each plot’s data is summarized as JSON metadata outlining the insights"*).
> In contrast, RAViG-Bench demands the creation of a holistic, visually-rich answer that includes not only charts but also explanatory text, information cards, and distinct content modules (see Appendix F for examples).
>
> 3) ***Evaluation Dimensions***:
> InsightBench evaluates agents at the summary level and the insight level by comparing their outputs to ground-truth summaries and insights, **focusing primarily on the quality of the content and the correctness of the extracted insights**. RAViG-Bench, by contrast, evaluates along three dimensions: functionality, design quality, and content quality. For content, we explicitly consider the logical consistency between text and data (including chart–text alignment), rather than only assessing the model’s understanding of the data itself.
>
> To our knowledge, there is no existing benchmark that jointly *uses real web retrieval*, *targets multi-component visually-rich answers*, and *provides an automatic, multi-modal evaluation* over functionality, design, and content. We have added this comparison to the Appendix A Related Work for clarity.
>
> ----
>
> ## **W2: Clarification on Task Design**
>
> We agree that a modular agent pipeline is a reasonable architecture. However, the RAViG task is designed for a different scenario where its end-to-end approach offers distinct advantages.
>
> 1) ***Different Application Scenarios***: Deep Research targets *complex, in-depth analysis* where a generation time of several minutes is acceptable. RAViG, in contrast, targets *high-volume, everyday search-like queries where users expect fast, visually enriched answers*. This is why our queries are drawn from Natural Questions and then filtered to those that "benefit from visualization" rather than "require deep analysis". In this setting, the key capability is not maximal analytical depth, but the ability to quickly turn noisy web retrievals into a coherent, visually-rich answer that is easy to consume in a chat/search interface.
> 2) ***The Advantages of End-to-End RAG Approach***: We also experimented with the commercial Deep Research features of Gemini and Tongyi on our benchmark queries. For everyday, search-like questions, we found that end-to-end RAViG generation is **comparable to these leading Deep Research systems in both content quality and visual design, while enjoying a substantial latency advantage**. Because the RAViG setting omits multiple intermediate stages such as additional search orchestration and markdown-to-HTML conversion, its end-to-end **runtime is only about 1/10 to 1/20** of Deep Research on the same queries. We have added these comparative results to Appendix J.
>
> ---
>
> [1] Gemini Deep Research, https://gemini.google/overview/deep-research/.
>
> [2] Meta AI, https://metaso.cn/

---

> > ### Author Response · Authors · 2025-11-20
> >
> > ## **Q: Clarification on GT-Free Design and Evaluation Robustness**
> >
> > ### **1) Rationale for a GT-Free Design**
> > The open-ended nature of the RAViG task makes a single, fixed Ground-Truth (GT) not only impractical but also undesirable. As shown in Appendix G, **different LLMs can generate diverse yet equally valid visual designs for the same query**. *Enforcing a single GT would unfairly penalize creative and high-quality alternative responses, thereby limiting a true assessment of a model's capabilities.* Besides, prior work has already explored GT-free evaluation [3,4,5,6] and demonstrated its effectiveness.
> >
> > ### **2) Ensuring Objective Evaluation and Mitigating Biases**
> >
> > 1) ***Principled, Low-Subjectivity Evaluation***: Our framework is *not based on subjective scoring*. For Missing Element, Occlusion, Reasonableness, and Faithfulness, we employ a binary judgment paradigm. The LLM-judge is prompted with detailed criteria to **provide a "Yes" or "No" answer**, requiring its comprehension and reasoning abilities **rather than inviting subjective opinions**. This minimizes the room for judge-specific bias. Furthermore, for highly perceptual metrics like color contrast, we use objective, rule-based methods, which are entirely free of LLM bias. The only metric that involves comparative scoring, Comprehensiveness, is adapted from the well-established WildBench [3].
> > 2) ***Robustness Analysis Against Judge-Specific Biases***: The self-preference analysis in Appendix N supports the reliability of the Comprehensiveness evaluation. In addition, we conducted an additional analysis on the responses from ten models to 100 randomly sampled queries (1,000 responses in total) using GPT-based and Gemini-based judges. As shown in Tab.1 (where “Model A + Model B” denotes using Model A to evaluate design and Model B to evaluate content), the rankings remain stable across judge configurations, and no model family receives systematically favorable treatment.
> >
> > |Judge Configuration|GPT5|Gemini2.5|GPT4O|Qwen3|DeepSeekv3|Doubao1.5|Claude 4|Llama3|Mistral|Llama4|
> > |:---|:---:|:---:|:---:|:---:|:---:|:---:|:---:|:---:|:---:|:---:|
> > |HPS(paper, GPT4O+Gemini2.5)|73.54|51.56|64.77|51.12|51.03|56.68|45.86|58.52|16.19|46.58|
> > |HPS(Gemini2.5+Gemini2.5)|67.71|47.61|63.08|40.86|48.51|51.94|39.71|49.61|15.00|41.64|
> > |HPS(GPT4O+GPT5)|71.67|51.91|67.81|52.61|57.17|55.38|46.01|60.38|17.13|51.56|
> >
> > **Table 1: Robustness Analysis Across Different Judge Configurations.**
> >
> > ---
> >
> > [3] Es, S., et al. "Ragas: Automated evaluation of retrieval augmented generation." EACL. 2024.
> >
> > [4] Lin, Bill Yuchen, et al. "Wildbench: Benchmarking LLMs with challenging tasks from real users in the wild." ICLR Spotlight. 2025.
> >
> > [5] Saad-Falcon, Jon, et al. "Ares: An automated evaluation framework for retrieval-augmented generation systems." NAACL. 2024.
> >
> > [6] RefChecker, https://github.com/amazon-science/RefChecker

---

> > > ### Comment · Reviewer_QJiz · 2025-11-27
> > > **Reply to author's rebuttal**
> > >
> > > Thank the authors for their detailed response and the new exp results. While the explanation regarding the input/output format, visualization requirement, and evaluation metrics difference between Deep Research and the RAViG task is clear, I remain unconvinced by the rebuttal, as it does not adequately address the fundamental questions raised.
> > >
> > > My primary concerns are as follows:
> > >
> > > 1. Lack of Justification for Task Necessity: The rebuttal successfully distinguishes what RAViG is from Deep Research (faster, HTML-output-focused). However, it fails to compellingly argue why we need this as a distinct and standalone task, especially when we already have well-established paradigms like standard RAG (for more accurate factual Q&A) and Deep Research agents (for analysis as a report).
> > > The existence of a "speed gap" does not automatically justify the creation of a new benchmark category. This paper needs a stronger motivation for why this specific niche—fast, visually-rich HTML generation—is a critical research problem in itself, rather than merely an engineering optimization or a sub-capability of a more general agent.
> > >
> > >
> > > 2. Insufficient Differentiation in Evaluation Focus: The authors correctly state that their evaluation focuses on chart quality and text-chart alignment, which they imply is a novel contribution. However, I argue that any competent Deep Research system or evaluation framework for a multi-modal report would also inherently care about and evaluate these aspects. The fact that prior works may have reported only overall metrics does not mean that the concept of evaluating chart-text alignment is new; it simply means that their evaluation methodology was coarser. A truly novel benchmark would need to demonstrate that it captures unique failure modes or capabilities that are specific to the RAViG setting and irrelevant or unmeasurable in the Deep Research context.
> > >
> > > 3. The authors' characterization of Deep Research benchmarks [1] is incomplete. They claim a key differentiator for RAViG is the synthesis of charts, text, and structured content into a holistic report. However, this is the fundamental objective of any Deep Research agent [1][2], which are inherently designed to process unstructured web data and produce such multi-modal outputs. The distinctions raised (e.g., output format, speed) appear to be differences in degree, not a fundamental difference in the core task.
> > >
> > > Based on above reasons, I would like to keep my score.
> > >
> > > [1] https://deepresearch-bench.github.io/
> > > [2] https://platform.openai.com/docs/guides/deep-research

---

> > > > ### Author Response · Authors · 2025-11-28
> > > >
> > > > We thank the reviewer for the additional comments. Below we address the key concerns raised.
> > > >
> > > > ---
> > > >
> > > > ## **R1: On the Necessity and Critical Value of the RAViG Task**
> > > >
> > > > > "However, it fails to compellingly argue why we need this as a distinct and standalone task, especially when we already have well-established paradigms like standard RAG (for more accurate factual Q&A) and Deep Research agents (for analysis as a report). "
> > > >
> > > > The reviewer identifies two established paradigms: standard RAG for factual Q&A and Deep Research agents for in-depth analysis. However, the inherent limitations of these paradigms highlight the necessity of RAViG:
> > > >
> > > > 1.  **Standard RAG:** Falls short in addressing queries that require analysis and insight beyond simple fact retrieval.
> > > > 2.  **Deep Research Agents:** While built for deep analysis, their **minute-scale** latency fails to meet the modern user's core demand for instant information access.
> > > >
> > > > A vast and critical gap exists between these two extremes, encompassing numerous queries that require both data-driven insights and real-time responses. For instance:
> > > > *   **Consumer Decisions:** A user comparing smartphones needs an immediate, side-by-side visual comparison of processor benchmarks, battery tests, and user ratings to make a quick decision, not scattered text or a lengthy technical review.
> > > > *   **Everyday Information Seeking:** Queries like "comparing my city's housing price trends over the past five years to the national average" are analytical and comparative, far exceeding the capabilities of standard RAG, yet do not warrant a full-scale Deep Research task.
> > > >
> > > > > "The existence of a "speed gap" does not automatically justify the creation of a new benchmark category. This paper needs a stronger motivation for why this specific niche—fast, visually-rich HTML generation—is a critical research problem in itself, rather than merely an engineering optimization or a sub-capability of a more general agent."
> > > >
> > > > 1.  **Distinct Research Challenges, Not Engineering Optimization:** The stringent, **second-level** latency constraint fundamentally alters the research problem. It forces models to abandon the iterative, multi-step refinement workflows common in Deep Research. This gives rise to a new scientific question: *How can a model, within a single or very few inference steps, simultaneously comprehend noisy documents, extract data, determine visualization logic, generate correct HTML code, and ensure semantic alignment between text and visuals?* This requires what we term **"Instantaneous Synthesis Intelligence"** a paradigm distinct from the iterative Chain-of-Thought process that underpins Deep Research.
> > > >
> > > > 2.  **Different Task Goals and Application Scenarios:**
> > > >     *   **Deep Research** serves asynchronous, offline analysis for expert users, valuing depth and comprehensiveness.
> > > >     *   **RAViG** serves synchronous, online interaction for the general public, valuing the immediacy and usability of visually-rich, easily digestible outputs.
> > > >
> > > > 3.  **Filling a Critical Academic Vacuum:** What the reviewer perceives as a "niche" is, in fact, **the core battleground for leading AI products like Google AI Overviews [1] and Perplexity [2], instant multimodal answer generation**. Yet, this area currently lacks a standardized academic benchmark. *Our work is not creating an "unnecessary category" but establishing the first scientific, quantifiable foundation for a field of immense and immediate importance*.
> > > >
> > > > [1] Google AI Overviews: https://search.google/ways-to-search/ai-overviews/
> > > >
> > > > [2] Perplexity: https://www.perplexity.ai/

---

> > > > > ### Author Response · Authors · 2025-11-28
> > > > >
> > > > > ## **R2: On the Differentiation and Novelty of the Evaluation Framework**
> > > > >
> > > > > > "However, I argue that any competent Deep Research system or evaluation framework for a multi-modal report would also inherently care about and evaluate these aspects. The fact that prior works may have reported only overall metrics does not mean that the concept of evaluating chart-text alignment is new; it simply means that their evaluation methodology was coarser."
> > > > >
> > > > > We agree that "any competent... system... would also inherently care about and evaluate these aspects." However, this very point underscores the novelty of our work: **making the decisive leap from "inherently caring" to "objectively and automatically measuring."**
> > > > >
> > > > > To the best of our extensive literature review, **no existing public benchmark provides an automated, multi-faceted evaluation framework covering functionality (e.g., rendering errors), design quality, content correctness (especially chart accuracy), and chart-text alignment as ours does.** We respectfully ask the reviewer to specify which prior studies offer a similar automated evaluation scheme, as this would be crucial for us to review and cite.
> > > > >
> > > > > > "A truly novel benchmark would need to demonstrate that it captures unique failure modes or capabilities that are specific to the RAViG setting and irrelevant or unmeasurable in the Deep Research context."
> > > > >
> > > > > The stringent constraints of RAViG introduce and amplify unique failure modes that are less prominent or can be circumvented in Deep Research settings. Our evaluation framework is specifically designed to capture these:
> > > > >
> > > > > 1.  **HTML-Specific Defects:** **Direct code generation** is prone to rendering errors and layout issues, which our framework detects and categorizes.
> > > > > 2.  **End-to-End Data Hallucination:** To meet speed requirements, models may bypass data verification, fabricating data to generate plausible but incorrect charts. Our chart correctness module is designed to identify such hallucinations.
> > > > > 3.  **Text-Chart Semantic Gaps:** In a single-pass generation, models are more susceptible to inconsistencies between textual descriptions and chart data. Our alignment evaluation is highly sensitive to this.
> > > > > 4.  **Heuristic Shortcuts and Expressiveness Degradation:** To optimize for speed, models may default to simpler chart types (e.g., bar charts) at the expense of more expressive visualizations (e.g., using a scatter plot to show correlation).
> > > > >
> > > > > ---
> > > > >
> > > > > ## **R3: On the Fundamental Difference Between RAViG and Deep Research**
> > > > >
> > > > > > "The authors' characterization of Deep Research benchmarks [1] is incomplete. They claim a key differentiator for RAViG is the synthesis of charts, text, and structured content into a holistic report. However, this is the fundamental objective of any Deep Research agent [1][2], which are inherently designed to process unstructured web data and produce such multi-modal outputs."
> > > > >
> > > > > We thank the reviewer for providing specific references. A closer examination of this work confirms that the distinction between RAViG and Deep Research is a **difference in kind, not merely in degree.**
> > > > >
> > > > > 1.  **Reference [1] (Deep Research Bench):** The benchmark itself defines Deep Research as a process involving *"sophisticated multi-step reasoning, comprehensive information synthesis"*. Its evaluation focuses on the final text report's *"Comprehensiveness, Insight/Depth, Instruction-Following, and Readability"*. Crucially, **it lacks a comprehensive evaluation of generated visualizations, similar to InsightBench, focusing instead on the agent's textual and data insights.**
> > > > >
> > > > > 2.  **Reference [2] (OpenAI's Deep Research Guide):** This guide positions the Deep Research Agent as a tool to *"find, analyze, and synthesize hundreds of sources to create a comprehensive report at the level of a research analyst"*. The described agent behaviors—such as "autonomously plans sub-questions" and "synthesize hundreds of sources"—point to a long-duration, exploratory workflow mimicking a human researcher.*In practice, OpenAI's Deep Research Agent does not demonstrate rich, varied visualization generation capabilities.*
> > > > >
> > > > > > "The distinctions raised (e.g., output format, speed) appear to be differences in degree, not a fundamental difference in the core task."
> > > > >
> > > > > *   The core cognitive goal of **Deep Research** is **exploration and argumentation**, producing a "research report" for offline, in-depth analysis. Its value lies in completeness and logical depth.
> > > > > *   The core cognitive goal of **RAViG** is **query and answer**, producing a "visualized answer" for online, instant consumption. Its value lies in immediacy and reducing the user's cognitive load.
> > > > >
> > > > > *Claiming the distinction is merely one of "degree" is akin to arguing that "image classification" and "real-time object detection" are only different in degree.* In computer science, we recognize them as fundamentally distinct fields driven by different user needs, technical stacks, and optimization objectives.

---

> > > > > > ### Author Response · Authors · 2025-11-28
> > > > > >
> > > > > > ## **Conclusion**
> > > > > >
> > > > > > In summary, we wish to clarify our core contributions. The reviewer suggests our evaluation dimensions lack novelty, yet our analysis of all works cited by the reviewer shows they do not support this claim. We firmly believe our automated framework is the first of its kind to systematically measure functionality, design quality, and chart-text alignment. **If prior work with similar automated evaluation capabilities exists, we would be grateful if the reviewer could point us to it, as it would be of great importance to our work.**
> > > > > >
> > > > > > Furthermore, we have detailed how RAViG is a distinct and necessary task, differing from Deep Research in its objectives, applications, and underlying technical challenges. We hope this analysis helps in re-evaluating its independent scientific value.
> > > > > >
> > > > > > Finally, Deep Research agents that generate multimodal reports are not in opposition to our work; rather, they are a key subject for our evaluation framework. Our benchmark provides the first automated, quantitative standard for assessing their visualization quality, thereby contributing to the advancement of the entire field, including Deep Research. We hope our work is seen as a constructive contribution to the community.

---

### Official Review · Reviewer_9hTA · 2025-10-31

**Soundness:** 3
**Presentation:** 4
**Contribution:** 3
**Rating:** 4
**Confidence:** 4

**Summary:**

The authors present a novel benchmark on visually-rich RAG for data summary and visualisation. It includes a dataset of natural language queries that are appropriate for visually-rich summarisation, and an automated LLM-as-judge evaluation protocol that separately assesses technical functionality, aesthetic presentation, and content quality/consistency. They provide a number of performance baselines from existing VLMs, and a reasonable analysis of agreement with human raters.

**Strengths:**

The use case is well-motivated and seems to address a genuine gap in current benchmarks, based on a convincing demonstration of failure modes in existing VLMs.

The dataset construction protocol is sensible, and the listed examples of queries that are suitable for visually-rich RAG appear appropriate. I had my doubts about whether purely automated query filtering would produce an adequate dataset, but the presence of a final human review is welcome.

The LLM-as-judge evaluation method is difficult to trust, so the extensive validation experiments performed are also welcome. Agreement with human raters is encouraging, and in particular, the analysis of self-preference bias is a good inclusion.

**Weaknesses:**

I would like to see a more detailed analysis of how much the protocol is reliant on these specific models, or a more robust protocol that can verifiably work well with open-source models. I would also like to see some analysis of where these models disagree with human evaluators, i.e. whether there are specific or systematic biases, and whether these differ across models.

Minor points that did not affect the review score:
- In the appendix, on page 25 (around lines 1311-1314), the query does not match the image. The query asks about “densely populated countries”, actually used in the next page, but the image is about F1 grand prix races.
- In table 3, the colouring scheme is counter-intuitive. It should not really matter, but my brain insists that green is good (high performance) and red is bad (low performance). Maybe this is a cultural thing, but I suggest reversing the scale.
- Small typos: “Releated work” (line 810) should be “Related work”, “Lable” (lines 1385 and 1392) should be “Label”

**Questions:**

1. The evaluation protocol is said to rely solely on GPT-4o for design quality assessment, and Gemini-2.5-pro for content quality evaluation. Is the quality of the evaluation procedure highly sensitive to these particular VLM/LLM choices?

2. Would the automated evaluation have more agreement with human raters if an ensemble of different models was used, or an ensemble of different prompts?

3. Is there a failsafe method in place if these APIs become inaccessible in the future, during the lifetime of the benchmark?

4. (minor) In 4.2, the section on occlusion (lines 261-268) is not as convincing as the others. Does this approach fail to detect when elements from separate semantic modules (i.e. under different headers) are overlapping? Does it fail if the generated output uses few header tags or none at all?

---

> ### Author Response · Authors · 2025-11-20
>
> We thank reviewer 9hTA for the  thoughtful and valuable feedback. We address the raised concerns in detail below.
>
> ------
>
> ## **W1/Q1/Q3: Model-Dependence and the Viability of Open-Source Models**
> > **W1**: I would like to see a more detailed analysis of how much the protocol is reliant on these specific models, or a more robust protocol that can verifiably work well with open-source models.
> ### **1) Principled Design for Model-Agnosticism**
> Our evaluation framework is designed to be model-agnostic in principle. For all LLM-judged dimensions except Comprehensiveness, the LLM-judge's role is not to provide a subjective score but to perform **a binary "Yes/No" judgment** based on detailed, objective criteria. This task primarily requires strong visual comprehension and reasoning abilities. Therefore, any model possessing these core capabilities can potentially serve as an effective judge. For Comprehensiveness, the original work [1] uses GPT‑4‑turbo‑0429 and Claude‑3‑Haiku; we use Gemini‑2.5-pro and still obtain high human–model agreement.
>
> ### **2) Empirical Analysis with Other Close- and Open-Source Models**
> We also conducted a new experiment to *evaluate the performance of various models as judges*, including leading open-source models such as Qwen3VL-235b-Instruct/Thinking (Abbreviated as Qwen-Instruct/Thinking), InternVL3-78B (the first open-source model on open_vlm_leaderboard[2]), and DeepSeek-v3. We measured their accuracy with the same samples and human-annotated ground truth used in the paper.
> As shown in Tab.1 and 2, **our evaluation protocol is not tightly bound to a single proprietary model**. With sufficiently capable open-source evaluators (e.g., Qwen3-Thinking), the same protocol yields relatively accurate judgments across all dimensions, except that Reasonableness remains the most demanding due to its need for stronger context reasoning.
>
> The results are presented in Tab.1 and 2.
>
> |Category|Model|Missing (FN+FP)|Occlusion (FN+FP)|
> |:---|:---|:---:|:---:|
> |**Open-Source**|Qwen-Instruct|81.5% (37+0)|71.0% (57+1)|
> ||Qwen-Thinking|89.5% (20+1)|78.5% (43+0)|
> ||InternVL3-78B|85.0% (29+1)|54.5% (91+0)|
> |**Closed-Source**|Gemini-2.5-pro|98.5% (3+0)|95.5% (6+3)|
> ||GPT-4o|95.5% (9+0)|93.0% (6+8)|
> ||GPT-5|94.0% (12+0)|93.5% (11+2)|
>
> **Table 1: Accuracy across models for Missing and Occlusion. (FN+FP) represent counts of (False Negatives + False Positives), totaling 200.**
>
> |Category|Model|Reasonableness (FN+FP)|Faithfulness (FN+FP)|
> |:---|:---|:---:|:---:|
> |**Open-Source**|Qwen-Instruct|64.0% (34+1)|82.0% (16+2)|
> ||Qwen-Thinking|74.0% (8+18)|84.0% (15+1)|
> ||Deepseek-v3|50.0% (50+0)|81.0% (17+2)|
> |**Closed-Source**|Gemini-2.5-pro|91.0% (6+3)|94.0% (1+5)|
> ||Gemini-2.5-flash|70.0% (20+10)|87.0% (12+1)|
> ||GPT-5|84.0% (7+9)|95.0% (5+0)|
> ||GPT-4o|49.0% (49+2)|85.0% (15+0)|
>
> **Table 2: Accuracy across models for Reasonableness and Faithfulness. (FN+FP) represent counts of (False Negatives + False Positives), totaling 100.**
>
> ### **3) Justification and Future Directions**
> Based on this analysis, to ensure the highest fidelity for our benchmark at present, we follow the precedent of prior work [1,3,4,5] in selecting the best-performing close-source models as the primary evaluators. However, we agree with the reviewer on reducing API dependency. Developing a fully open-source alternative is indeed a important direction for our future work.
>
> ---
>
> [1] Lin, Bill Yuchen, et al. "Wildbench: Benchmarking LLMs with challenging tasks from real users in the wild." ICLR Spotlight. 2025. Use GPT‑4‑turbo‑0429 and Claude‑3‑Haiku.
>
> [2] https://huggingface.co/spaces/opencompass/open_vlm_leaderboard
>
> [3] Chen, Nan, et al. "VisEval: A benchmark for data visualization in the era of large language models." TVCG. (VIS). 2024. Use GPT-4V.
>
> [4] Bai, Ge, et al. "Mt-bench-101: A fine-grained benchmark for evaluating large language models in multi-turn dialogues." ACL. 2025. Use GPT-4
>
> [5] Liu, Yang, et al. "G-eval: NLG evaluation using gpt-4 with better human alignment." arXiv preprint. 2023. Use GPT-4

---

> > ### Author Response · Authors · 2025-11-20
> >
> > ## **W2: More Analysis on Human-Model Disagreements**
> > > **W2**: I would also like to see some analysis of where these models disagree with human evaluators, i.e. whether there are specific or systematic biases, and whether these differ across models.
> >
> > Thank you for the valuable suggestion. Following the advice, we have added a detailed analysis and some representative human-model inconsistent examples in Appendix H. This analysis first led us to discover that a portion of initial disagreements stemmed from subtle human annotation errors. After a meticulous re-evaluation, we have corrected the ground truth, and the updated model accuracies are reported in Tab. 1 and 2 (response to W1/Q1/Q3).
> >
> > Our analysis of the corrected model failures reveals the following findings:
> > 1) ***Design Dimensions: Capability Gaps Lead to Missed Issues***.
> >    + The primary failure mode in Missing and Occlusion is missing subtle issues (false negatives), which we attribute to capability gaps in visual perception, *not a systematic bias*. Most open-source models **consistently fail** to detect fine-grained defects, while even top-tier proprietary models **share a common bottleneck on extremely difficult cases** (Fig. 18-20 in Appendix H).
> >    + In Occlusion evaluation, Gemini and GPT models can *hallucinate when faced with small fonts or low color contrast*. We observe this tendency somewhat more often in GPT-4o for low contrast (Fig. 21).
> > 2) ***Information Dimensions: Failures in Reasoning and Knowledge***.
> >    + Reasonableness: Assessing reasonbaleness requires strong reasoning capabilities as well as a solid understanding of HTML. Non-reasoning models like Qwen-Instruct and GPT-4o naturally fail. High-performing models can fail due to *knowledge hallucination* (Fig. 22) or *excessive reasoning* (Fig. 23).
> >    + Faithfulness: Models are prone to hallucinations when dealing with *noisy inputs (e.g., similar entities (Fig. 24, left))* or tasks that *require implicit reasoning* (Fig. 24, right).
> >
> > ---
> >
> > ## **W3: Correction of Minor Points**
> > Thank you for pointing out these typo issues. We have addressed all of them in the revised version:
> > 1) *Appendix query–image mismatch*: We corrected the example (Page 37, Fig. 29 now). The updated figure now uses the appropriate query: "Details of all F1 Grand Prix races for the 2025 season."
> > 2) *Tab. 3 (in paper) color scheme*: Following the suggestion, we removed the color encoding and highlight the best score in bold and the worst score with an underline, which avoids the green/red ambiguity.
> > 3) Typos: We fixed "Releated work" to "Related work" (Line 864), and updated all figures containing "Lable" to "Label" (Page 38, Fig. 30 now).

---

> ### Author Response · Authors · 2025-11-20
>
> ## **Q2: Analysis of Model Ensembling**
> We have conducted experiments to investigate the effectiveness of ensembling both models and runs. We evaluated two ensemble strategies: Majority Voting and Optimistic Aggregation (where a single "issue" verdict determines the outcome), and applied them in two distinct settings:
> + *Intra-model ensemble*: Using three runs of the top open-source model (Qwen-thinking, with temperature=1).
> + *Inter-model ensemble*: Using the top-3 closed-source models.
>
> As shown in Tab.3 and 4:
> 1) ***Majority Voting showed limited improvement***. *Because difficult cases often represent a shared bottleneck for most models*, this strategy tended to converge toward the average performance of the constituent models rather than overcoming their common limitations.
>
> 2) ***Optimistic Aggregation was only effective under specific conditions***.
>     + *Effective on low-FP dimensions (e.g., Missing)*: It exploited any model/run's peak performance to catch subtle issues missed by others. This boosts recall and outperforms the best single model, as the cost of a few extra FPs is minimal.
>     + *Ineffective on high-FP dimensions (e.g., Reasonableness)*: It amplified errors from any single model/run's hallucination or excessive reasoning. The resulting surge in FPs negates the benefit of reducing false negatives.
>
> In conclusion, while ensembling holds promise, the optimal strategy is highly dependent on the specific characteristics of the task.
>
> |Type|Model|Missing (FN+FP)|Occlusion (FN+FP)|
> |-------|-------------------------|-----------------|-------------------|
> |**Open-Source**|Qwen-Thinking|89.5% (20+1)|78.5% (43+0)|
> ||Thinking-vote1|91.5% (16+1)|80.5% (39+0)|
> ||Thinking-vote2|86.0% (25+3)|83.5% (33+0)|
> ||**Majority Voting**|90.0% (19+1)|80.5% (39+0)|
> ||**Optimistic Aggregation**|92.0% (13+3)|86.5% (27+0)|
> |**Closed-Source**|Gemini-2.5-pro|98.5% (3+0)|95.5% (6+3)|
> ||GPT-4o|95.5% (9+0)|93.0% (6+8)|
> ||GPT-5|94.0% (12+0)|93.5% (11+2)|
> ||**Majority Voting**|95.5% (9+0)|95.5% (6+3)|
> ||**Optimistic Aggregation**|99.5% (1+0)|93.5% (1+12)|
>
> **Table 3: Accuracy of Models and Ensembles on the Design Dimensions.**
>
> |Type|Model|Reasonableness (FN+FP)|Faithfulness (FN+FP)|
> |-------|-------------------------|-------------------------|----------------------|
> |**Open-Source**|Qwen-Thinking|74.0% (8+18)|84.0% (15+1)|
> ||Thinking-vote1|73.0% (13+14)|82.0% (15+3)|
> ||Thinking-vote2|72.0% (8+20)|87.0% (11+2)|
> ||**Majority Voting**|72.0% (8+20)|86.0% (13+1)|
> ||**Optimistic Aggregation**|73.0% (13+14)|86.0% (10+4)|
> |**Closed-Source**|Gemini-2.5-pro|91.0% (6+3)|94.0% (1+5)|
> ||Gemini-2.5-flash|70.0% (20+10)|87.0% (12+1)|
> ||GPT-5|84.0% (7+9)|95.0% (5+0)|
> ||**Majority Voting**|85.0% (11+4)|91.0% (8+1)|
> ||**Optimistic Aggregation**|84.0% (0+16)|93.0% (0+7)|
>
> **Table 4: Accuracy of Models and Ensembles on the Information Dimensions.**
>
> ---
>
> ## **Q4: Roubustness of Occlusion Detection**
> In our current pipeline, screenshots of different modules are stitched into the final rendered page (see Appendix F), so *any occlusion will necessarily appear in at least one captured region and can still be detected*. When there are few or no header tags, we fall back to a screenshot of the entire page; based on manual inspection of many cases, these header-less layouts tend to be relatively simple, and occlusion is still easy to judge from a single image.

---

### Official Review · Reviewer_2Hgk · 2025-11-01

**Soundness:** 3
**Presentation:** 3
**Contribution:** 3
**Rating:** 6
**Confidence:** 2

**Summary:**

This paper introduces RAViG-Bench, a benchmark for evaluating Retrieval-Augmented Visually-Rich Generation (RAViG) systems. It targets models that must both retrieve multimodal evidence and generate grounded visual outputs. While prior work has separately evaluated retrieval reasoning or visual generation, RAViG-Bench assesses the complete retrieval-to-generation pipeline.
The benchmark defines three complementary evaluation axes, functionality, design quality, and content fidelity, measured through a hybrid approach combining rule-based structural metrics and an LLM-as-a-judge for semantic and stylistic assessment. A complexity-corrected holistic score further normalizes performance across layouts of varying difficulty, discouraging trivial generations.
Ten multimodal generative models, including GPT-5 as the latest, are evaluated under a unified protocol. Results show that current models produce visually plausible yet semantically inconsistent outputs, exposing systematic weaknesses in factual grounding and compositional reasoning. Beyond reporting scores, the paper analyzes failure types and suggests directions for future models that better integrate retrieval accuracy, design coherence, and content faithfulness.

**Strengths:**

The paper is convincing in its motivation. It identifies the lack of a benchmark that jointly evaluates retrieval and visual generation, and addresses this gap with a reproducible and well-structured framework. The proposed three-axis design is coherent and logically motivated. The introduction of new evaluation metrics is particularly commendable, as it reveals cases where models succeed through oversimplified outputs rather than genuine multimodal reasoning. The benchmark’s overall design is practical, reproducible, and aligned with real-world multimodal generation workflows. Its evaluation protocol, illustrated in Fig. 3, is comprehensive, assessing both functional and aesthetic quality beyond textual correctness. The hybrid evaluation scheme combining rule-based validation with an LLM-as-a-judge, achieves a pragmatic balance between objectivity and interpretive depth. The empirical section is substantial. It presents a meaningful taxonomy of failure patterns and articulates clear directions for what stronger models should achieve. The inclusion of recent frontier models under identical zero temperature conditions ensures reproducibility and enables reliable cross-model comparison of multimodal generation capabilities. The paper demonstrates careful engineering, solid coverage of prior literature, and thoughtful reflection on how evaluation can drive progress in multimodal generation research. Section 5.3 provides an exemplary in-depth analysis that dissects error patterns and design trade-offs across model families, revealing nuanced distinctions in retrieval grounding and layout control.

**Weaknesses:**

Despite the paper’s quality, several conceptual limitations prevent RAViG-Bench from being fully convincing as a standard benchmark for visually rich generation.

1. The reliability of the LLM-as-a-judge protocol is not convincingly established.
While the authors report correlations with human judgments, such correlation alone does not guarantee fairness or stability.
It remains unclear whether the same model family acting as both generator and evaluator introduces self-preference bias, or whether version drift and sampling variance were systematically tested.
Since the reported rankings rely heavily on these automatic judgments, a more rigorous quantitative validation of scoring reliability is necessary.


2. Fixing temperature to zero improves reproducibility but oversimplifies the evaluation.
This deterministic setup removes stochastic diversity, which is crucial for understanding how multimodal generators behave in real deployment.
Consequently, the benchmark captures only a single deterministic output per query, making it difficult to assess model robustness or creative range.
A complementary analysis using moderate temperature values would provide a more realistic picture of generative variability.


3. Although the appendix includes numerous rendered-page examples, the paper does not analyze them in sufficient depth.
These visuals appear without detailed textual interpretation, leaving readers uncertain about which specific failure modes or design patterns they represent.
While Section 5.3 offers valuable qualitative insights, its narrative remains largely disconnected from the visual evidence provided.
Incorporating even a small number of representative rendered examples directly into the analysis would enhance both interpretability and transparency.

4. The benchmark primarily measures factual consistency and renderability, yet overlooks higher-level qualities such as design coherence, narrative flow, and alignment with human intent.
These aspects are central to evaluating multimodal generation as creation, not merely as factual assembly.
Without accounting for such qualitative dimensions, the benchmark, though methodologically rigorous, remains somewhat narrow in how it operationalizes visual generation quality.

**Questions:**

Q1. Can the authors provide additional quantitative validation to assess scoring robustness? For example, evaluating cross-family judgments (e.g., Gemini judging GPT outputs) or repeating the evaluation with different model versions would reveal whether ranking stability holds across evaluators.
Such analysis would strengthen confidence that the reported rankings are not artifacts of evaluator bias.


Q2. Would the authors consider an additional experiment that compares deterministic and mildly stochastic decoding settings?
Such a comparison could reveal whether the benchmark rankings remain stable when generation variability is introduced, thereby improving ecological validity.



Q3. Could the authors include a few representative rendered-page examples within the main text or extend Section 5.3 with brief commentary linking each example to observed failure types?
This addition would make the qualitative analysis more transparent and verifiable, without requiring major additional experiments.


Q4. Connecting to Q2, can the authors discuss additional evaluation dimensions to capture creative quality?
Even a small-scale human study or expert annotation subset could substantiate the broader claims about visually rich generation.

---

> ### Author Response · Authors · 2025-11-20
>
> We thank reviewer 2Hgk for the thoughtful and valuable feedback. We address the raised concerns in detail below.
>
> ---
> ## **W1/Q1: Robustness and Bias Analysis of the LLM-as-a-Judge Protocol**
> Our framework is designed to minimize subjectivity. As detailed in the paper, all LLM-judged dimensions (except for Comprehensiveness) rely on a **binary "Yes/No" judgment based on strict criteria, not open-ended scoring**. This design inherently limits the potential for judge-specific bias. Appendix N reports a self-preference analysis for Comprehensiveness, showing no systematic self-preference bias. *The LLM-as-a-judge paradigm itself is also well-established in prior work* [1,2].
>
> As suggested, we re-evaluated the responses from ten models to 100 randomly sampled queries (1,000 responses in total) *using a cross-family LLM-judge configuration*. As shown in Tab.1 ("Model A + Model B" denotes Model A for design, Model B for content):
> 1) ***High Ranking Stability***: The relative model rankings are highly stable across all judge configurations.
> 2) ***No Self-Preference Bias***: When Gemini-2.5 acts as judge, it still scores its own performance significantly below GPT-5. Similarly, GPT-based judges do not artificially inflate the scores of other GPT models.
>
> These results suggest that **our evaluation is not dominated by judge-specific biases, and the comparative ranking of models is robust**.
>
> |Judge Configuration|GPT5|Gemini2.5|GPT4o|Qwen3|DeepSeekv3|Doubao1.5|Claude 4|Llama3|Mistral|Llama4|
> |:-|:-:|:-:|:-:|:-:|:-:|:-:|:-:|:-:|:-:|:-:|
> |HPS(paper, GPT4o+Gemini2.5)|73.54|51.56|64.77|51.12|51.03|56.68|45.86|58.52|16.19|46.58|
> |HPS(Gemini2.5+Gemini2.5)|67.71|47.61|63.08|40.86|48.51|51.94|39.71|49.61|15.00|41.64|
> |HPS(GPT4o+GPT5)|71.67|51.91|67.81|52.61|57.17|55.38|46.01|60.38|17.13|51.56|
>
> **Table 1: Robustness Analysis Across Different Judge Configurations.**
>
> ---
> ## **W2/Q2: Roubustness between Deterministic and Mildly Stochastic Decoding Settings**
> Following the recommendation, we conducted an additional experiment to compare model performance under deterministic decoding (in our paper) versus a mildly stochastic decoding setting (temperature=0.7, top_p=0.9) on random sample of 100 queries.
> As shown in Tab.2:
> 1) ***Overall scores and rankings are stable***: The model rankings demonstrate high stability across both settings. This confirms the robustness and reliability of our evaluation framework.
> 2) ***Larger fluctuations are driven by generation-side variability***: For instance, Gemini-2.5-pro's DSR improved notably. Our analysis traces this to a large reduction in "oversized elements" issues (from 25% to 13%), which comes from stochastic decoding helping the model avoid a deprecated CSS link that it consistently defaults to when temperature=0.
>
> |Metric|GPT5|Gemini2.5|GPT4o|Qwen3|DeepSeekv3|Doubao1.5|Claude 4|Llama3|Mistral|Llama4|
> |:-|:-:|:-:|:-:|:-:|:-:|:-:|:-:|:-:|:-:|:-:|
> |**Temp=0**|||||||||||
> |DSR|0.77|0.61|0.92|0.73|0.79|0.74|0.66|0.91|0.25|0.73|
> |ECQ|95.5|84.5|70.4|70.0|64.6|76.6|69.5|64.3|64.8|63.8|
> |HPS|73.5|51.6|64.8|51.1|51.0|56.7|45.9|58.5|16.2|46.6|
> |**Temp=0.7**|||||||||||
> |DSR|0.76|0.77|0.94|0.66|0.84|0.72|0.59|0.90|0.29|0.73|
> |ECQ|94.1|81.2|68.6|73.9|71.1|74.4|73.7|63.4|55.4|64.0|
> |HPS|71.5|62.5|64.5|48.8|59.8|53.5|43.5|57.1|16.1|46.7|
>
> **Table 2: Comparison of Model Performance under Deterministic (Paper Setting) and Stochastic (Temperature=0.7) Decoding.**
>
> ---
> ## **W3/Q3: Added Analysis and Visual Examples**
> Thank you for the useful suggestion. In the revised version, we have added analyses to Appendix I, and references in Section 5.3 from each discussed failure type to its corresponding rendered example.
>
> ---
> ## **W4/Q4: Evaluation Dimensions of Creative Quality**
> In our evaluation framework, the visual richness score is related to a model's creative quality. Following the suggestion, *we conducted an expert annotation study to test whether this automated metric aligns with human perceptions*. Specifically, three experts rated the visual richness of outputs from 10 models. A Pearson correlation analysis between our visual richness score $VC _ {\text{score}}$ and the averaged human ratings shows a **strong, positive, and statistically significant correlation (r=0.90, p=0.0004)**. This indicates that *our metric effectively captures the structural aspects of visual richness as perceived by humans*. Visualization and more details have been added to Appendix K.
>
> At the same time, we fully agree with your assessment, and as we already acknowledged in the limitation that our current benchmark prioritizes usability aspects first. Automating the evaluation of more subjective qualities like *design coherence* and *narrative flow* remains a key direction for future work.
>
> ----
> [1] Zheng, Lianmin, et al. "Judging LLM-as-a-judge with MT-Bench and Chatbot Arena." NeurIPS. 2023.
>
> [2] Chen, Nan, et al. "VisEval: A benchmark for data visualization in the era of large language models." TVCG. (VIS). 2024.

---

> > ### Comment · Reviewer_2Hgk · 2025-11-27
> >
> > Dear Authors,
> >
> > Thank you for your answers. They address all my questions. Please incorporate these new findings into the manuscript.
> >
> > Best,
> > Reviewer 2Hgk

---

> > > ### Author Response · Authors · 2025-11-28
> > >
> > > **Dear Reviewer 2Hgk,**
> > >
> > > We are delighted to receive your encouraging feedback! It's great to know our replies have resolved your questions.
> > >
> > > As requested, we have now incorporated these new findings into the revised manuscript. Specifically:
> > >
> > > 1) The discussion for W1/Q1 has been added to *Appendix O* and is referenced in the main text (Line 369).
> > > 2) The discussion for W2/Q2 has been added to *Appendix P* and is referenced in the main text (Line 431).
> > > 3) The content for W3/Q3 has been updated in *Appendix I* and *Section 5.3*.
> > > 4) The content for W4/Q4 has been added to *Appendix K*.
> > >
> > > We hope these additions have further strengthened the paper, and we would be very grateful if you could consider offering increased support.
> > >
> > > Thank you again for your time and valuable guidance.
> > >
> > > **Best regards, \
> > > #3310 Authors**

---

### Official Review · Reviewer_ZNZT · 2025-11-01

**Soundness:** 2
**Presentation:** 3
**Contribution:** 2
**Rating:** 6
**Confidence:** 3

**Summary:**

This paper discusses the topic of Retrieval-Augmented Visually-rich Generation (RAViG), which extends RAG by combining textual explanations with multiple visual elements in a structured layout. It identifies a benchmarking gap: existing RAG benchmarks are text-only, and NL2VIS benchmarks focus on charting rather than the RAG paradigm. To address this, the authors propose RAViG-Bench, a comprehensive benchmark built from authentic user queries paired with real-world web retrievals to simulate realistic RAViG scenarios. They also introduce a multi-modal automated evaluation framework that assesses functionality, design quality, and content quality across both textual and visual components. While HTML is used as the working representation, the dataset, evaluation framework, algorithms, and criteria are format-agnostic and can be adapted to other structured formats by modifying input/output modules. Experiments on leading commercial and open-source LLMs reveal notable limitations and significant room for improvement. The current system reliably detects readability-related defects and performs basic visual complexity correction but does not evaluate subjective aesthetics or richer factors like typography and layout, and it excludes judgments about “over-design.” The authors suggest future work should incorporate user-centered aesthetic evaluation and finer-grained visual refinement to assess visual appeal, accessibility, and overall user experience.

**Strengths:**

* Clear problem formulation and novelty
The paper clearly positions RAViG-Bench as the first benchmark to comprehensively evaluate Retrieval-Augmented Visually-rich Generation (RAViG), i.e., reports that integrate textual explanations with multiple visual elements under the RAG paradigm. The contributions (dataset, automated evaluation framework, and empirical study) are well delineated.

* Realism-oriented data construction
Queries are derived from real user search queries and paired with top web retrievals, preserving noise, with human validation applied. This design faithfully simulates realistic RAG conditions where models must ground to multiple noisy sources.

* Alignment between automatic and human evaluation
The paper reports agreement analyses showing that automated evaluation correlates with human judgments, supporting the validity of the framework.

* Broad model coverage and actionable findings
A diverse set of commercial and open-source LLMs is evaluated under uniform settings. Results reveal that functionality is relatively strong but design quality is a major bottleneck, providing concrete guidance for future model improvements.

**Weaknesses:**

* $\gamma_{vc}$ and $VC_{score}$ design choices may be somewhat ad hoc
The visual richness score ($VC_{score}$) aggregates z-scored counts (modules/charts/tables) with fixed weights.
Similarly, $\gamma_{vc}$ is defined using a linear factor of $VC_{score}$ (e.g., $\alpha$ = 0.3).
The generality of these settings has not been fully stress-tested.

* Limited reproducibility due to partial prompt disclosure
For confidentiality reasons, complete prompts are not released, constraining exact replication, ablations on prompt sensitivity, and fine-grained reevaluation.

* As shown in Table 3, the performance of GPT-5 seems very high and nearly perfect according to the evaluation metrics introduced in this paper. It seems that RAViG-Bench is almost solved by current top-tier commercial LLMs. This implies that RAViG-Bench will be difficult to use for distinguishing performance among top-tier commercial LLMs in the future. This may reduce the usefulness of the proposed dataset.

**Questions:**

* The agreement between automated multi-modal evaluation and human assessments as shown in Table 2 seems very high.
It is basically a good point.
However, I suspect that the sampled problems might be extremely easy.
Could the authors elaborate on this point?

* There are many proposed evaluation metrics.
It would be better to clearly show the chance rate that can easily understand the effectiveness of each method on each metric.
Could the authors provide the chance rate for each metric?

---

> ### Author Response · Authors · 2025-11-20
>
> We thank reviewer ZNZT for the  thoughtful and valuable feedback. We address the raised concerns in detail below.
>
> ----
>
> ## **W1: Effectiveness of $VC _ {score}$**
> We introduced $VC _ {score}$ to prevent models from obtaining inflated scores by producing overly simple layouts. The weights in $VC _ {score}$ were chosen to reflect the relative impact of different components on perceived visual richness and their discriminative power across models.
> We acknowledge that these choices were initially heuristic, so we conducted additional analyses to examine their validity and sensitivity:
> ### **1) Correlation with Human Judgment**
> We conducted an human annotation study to evaluate the meaning of $VC _ {score}$. Specifically, three experts rated the visual richness of outputs from 10 models. The results show a **strong positive correlation** between $VC _ {score}$ and human judgment *(Pearson's r = 0.90, P-value = 0.0004)*. This indicates that *our metric effectively captures the structural aspects of visual richness as perceived by humans*. Visualization and details have been added to Appendix K.
> ### **2) Hyperparameter Sensitivity Analysis**
> We further analyzed how model rankings were affected by changes to key hyperparameters $\alpha$ and the internal weights of $VC _ {score}$.
> + *Sensitivity to $\alpha$*: We analyzed the effect of varying $\alpha$ from 0.1 to 1.0. As expected, model rankings showed moderate sensitivity to the penalty's strength. Crucially, this analysis revealed that our chosen value of $\alpha=0.3$ falls within an optimal range (0.1–0.3) where the resulting rankings best align with human preferences.
> + *Robustness to $VC _ {score}$ weights*: We fixed the weight of table $t _ {w}$ and varied the weight of module $m _ {w}$ from 0 to 0.9. We found that model rankings were highly robust to $m _ {w}$, with Kendall's $\tau$ > 0.95 for the vast majority of its tested range. This demonstrates that the overall signal of visual richness is strong enough that the final outcome is not dependent on the specific fixed weights.
>
> The visualized results of the parameter sensitivity analysis have been added to Appendix L.
>
> ----
>
> ## **W2: Reproducibility and Prompt Disclosure**
> We acknowledge the reviewer’s concern about reproducibility. We are committed to full transparency to ensure our work is verifiable and extensible.
>
> 1) *Evaluation Reproducibility*: To ensure our evaluation protocol can be precisely replicated, we commit to releasing **all evaluation prompts, the complete dataset, and our scoring code** upon acceptance. This will allow the community to perform exact replications and fine-grained re-evaluations of our results.
> 2) *Generation Results and System Prompts*: Regarding the system prompts for generation, we are currently under a temporary confidentiality restriction. However, to facilitate immediate result verification, **we will release the complete set of model-generated outputs (inference results)** used in our paper. This enables researchers to directly apply our evaluation framework to our results without needing to replicate the generation step. We will also release the system prompts themselves as soon as these temporary restrictions are lifted.
>
> ----
>
> ## **W3: RAViG-Bench with Top-Tier LLMs**
> We agree that GPT-5 performs very strongly on RAViG-Bench, but it does not "solve" the benchmark, and the benchmark's primary value extends beyond simple ranking.
>
> 1) ***GPT-5 is not perfect***: Under our protocol, GPT-5’s design success rate (DSR) is 81.9%, meaning that nearly 1/5 generated pages still contains at least one design flaw. For a system meant to deliver user-friendly experiences, this is a high failure rate. Its case-level faithfulness is also only 85.34%, indicating that 14.66% of responses contain at least one hallucinated or unsupported claim. In high-stakes domains such as financial analysis or health advice, this level of reliability is still insufficient.
> 2) ***RAViG-Bench is diagnostic, not just for ranking***: Beyond separating top-tier LLMs, the benchmark provides fine-grained error analysis across multiple dimensions. For example, we can pinpoint that GPT-5’s main design failures concentrate on oversized elements and low color contrast, which is actionable for future model and system design improvements. *This diagnostic capability remains useful even at high overall scores and will stay valuable as models improve.*

---

> > ### Author Response · Authors · 2025-11-20
> >
> > ## **Q1: Analysis of High Human–Model Agreement**
> > We would like to clarify that this high agreement is not due to easy samples, but rather is attributed to the design of our evaluation paradigm.
> >
> > 1) The test set was **randomly sampled** (as further explained in Line 361-363) and is **representative of the overall difficulty**. Our analysis even uncovered instances where models correctly identified subtle errors missed by human evaluators, such as missing axis ticks or formatting-induced value mismatches (Fig. 18). Appendix H provides more examples of such challenging cases.
> > 2) **We convert complex multimodal tasks into binary decisions with explicit, detailed criteria**. Under this setting, models only need to apply given rules instead of solving open-ended questions, which reduces ambiguity and makes their behavior better.
> > 3) For dimensions prone to hallucination-prone (e.g., color contrast), we deliberately use deterministic, rule-based evaluation rather than LLM-as-a-Judge.
> >
> > ----
> >
> > ## **Q2: Chance Rate of Evaluation Metrics**
> > We provided chance rate in Tab.2 to clarify the effectiveness of each metric. The chance rate for binary classification metrics (Missing, Occlusion, Overflow, Contrast, Reasonableness) is 50%. For the 3-class Faithfulness metric is 33.3%. The Comprehensiveness score, being a correlation-based metric, does not have an equivalent chance rate; its baseline is 0 (no correlation). Therefore, we have marked it as N/A in the table to avoid confusion.
> >
> > |Dimensions|Missing|Occlusion|Overflow|Contrast|Reasonableness|Comprehensiveness|Faithfulness|
> > |:---|:---:|:---:|:---:|:---:|:---:|:---:|:---:|
> > |**Results**|97.5%|94.0%|94.5%|95.0%|93.0%|0.912 / 0.892|94.0%|
> > |**Chance Rate**|**50.0%**|**50.0%**|**50.0%**|**50.0%**|**50.0%**|**N/A**|**33.3%**|
> >
> > **Table 2: Agreement with Human Assessments and Chance Rate of Evaluation Metrics.**

---

> > > ### Comment · Reviewer_ZNZT · 2025-11-27
> > >
> > > Thank you for addressing the concerns and questions I raised in my initial review.
> > >
> > > W1: Thank you for providing additional analyses to examine the validity and sensitivity of $WC_{score}$. The authors' rebuttal has dispelled my concern.
> > >
> > > W2: Thank you for promising to release all materials (e.g., the prompts, dataset, scoring code, and generation results) necessary to replicate the experiments in this paper, including the system prompts once the temporary restrictions are lifted. This will be very helpful to all researchers interested in this paper.
> > >
> > > W3: I am not fully convinced by the authors' rebuttal regarding the future usefulness of RAViG-Bench, but I understand that this point is not a critical flaw of this paper.
> > >
> > > Q1/Q2: Thank you for clearly answering my questions. I now understand the situation regarding the evaluations. I have no additional questions about them.
> > >
> > > Overall, I will leave my rating unchanged, as my initial rating is already positive for acceptance to the conference.

---

> > > > ### Author Response · Authors · 2025-11-27
> > > >
> > > > **Dear Reviewer ZNZT**,
> > > >
> > > > Thank you very much for your thoughtful follow-up and for taking the time to review our responses. We are glad that our clarifications helped address your earlier concerns, and we truly appreciate your constructive feedback and engagement.
> > > >
> > > > We also hope that the additional analyses and explanations provided during the discussion may help strengthen your confidence in the overall assessment.
> > > >
> > > > Thank you again for your time and effort.
> > > >
> > > > **Best regards,** \
> > > > **#3310 Authors**

---

### Author Response · Authors · 2025-11-20
**Explanation of Revisions**

We sincerely thank all reviewers for their insightful and valuable comments. Following your suggestions, we have revised the PDF and highlighted all changes in blue (figure captions are also marked in blue). Specifically, we have:
1) Corrected the identified typos; the exact locations are specified in the corresponding individual responses.
2) Expanded Appendix A: Related Work with a discussion on deep research agents (Page 18).
3) Added Appendix G: Examples of Visually Rich Answers from Different LLMs for the Same Query (Page 24-27).
4) Added Appendix H: Examples of Human–Machine inconsistent Cases in Design and Content Dimensions (Page 28-33).
5) Expanded Appendix I: Examples of Cases With Content Issues with additional clarifications (Page 34).
6) Added Appendix J: Examples of Responses from Deep Research Agents and RAViG (Page 39-42).
7) Added Appendix K: Human Annotation of Visual Richness (Page 43).
8) Added Appendix L: Sensitivity Analysis of the Score Correction Mechanism (Page 43-44).

We will also refer to these new sections and provide detailed answers in our point-by-point responses below.

---

### Author Response · Authors · 2025-11-27

**Dear Reviewers,**

I hope this message finds you well. As the discussion phase is nearing its end, we would like to kindly check whether our previous responses have addressed your concerns.

We totally understand that this is quite a busy period, and we sincerely appreciate the time and effort you have devoted to reviewing our work. If there are any remaining questions, we would be more than happy to clarify them to the best of our ability.

Thank you again for your time and valuable feedback.

**Best regards,**\
**#3310 Authors**

---

### Author Response · Authors · 2025-12-02
**Summary for Area Chair: Reviewer Concerns and Our Responses**

**Dear Area Chair,**

We sincerely apologize for any additional reviewing burden the new ICLR rebuttal policy may create. To help streamline your review, we provide a concise summary of each reviewer’s main concerns and how we addressed them during the rebuttal.

---

### **Reviewer ZNZT (Reviewer confirmed all concerns were addressed)**
1. **W**: Questioned whether the visual richness score ($ VC_{score} $) and its penalty design are ad hoc and robust.

   **A**: We added human-correlation and hyperparameter sensitivity analyses showing $ VC_{score} $ is both meaningful and stable.
2. **W**: Worried that partial prompt disclosure harms reproducibility.

   **A**: We commit to releasing all evaluation prompts, dataset, scoring code, and generated outputs (and system prompts once allowed).
3) **W**: The benchmark may be too easy for top-tier models such as GPT-5.

   **A**: GPT‑5 is still far from perfect and that the benchmark’s main value is fine-grained diagnostic analysis, not just ranking.
4) **W**: High human-model agreement might be due to easy samples, and requested chance rates for metrics.

   **A**: The test set was randomly sampled and high agreement stems from the binary-decision evaluation design. Provided chance rates for all metrics.

---

### **Reviewer 2Hgk (Reviewer confirmed all concerns were addressed)**
1) **W**: Questioned the robustness and potential bias of the LLM-as-a-judge protocol.

   **A**: We stress our binary, criteria-based design and add cross-family judge experiments showing stable rankings and no self-preference bias.
2) **W**: Worried that using only temperature=0 hides behavior under more realistic stochastic decoding.

   **A**: We run additional temperature=0.7 experiments and show that scores and rankings remain highly consistent.
3) **W**: Asked to better connect failure-type analysis to concrete rendered examples.

   **A**: We link Section 5.3 to representative rendered cases and expand the example-based analysis in the appendix.
4) **W**: Felt the benchmark under-explores creative dimensions like design coherence and narrative flow.

   **A**: We show visual richness correlates strongly with expert ratings and acknowledge more subjective creative metrics as key future work.

---

### **Reviewer 9hTA**
1) **W**: Worried the evaluation is tied to specific proprietary judges and not clearly viable with open-source models.

   **A**: We demonstrate that several strong open-source and alternative closed-source models can also serve as reasonably accurate judges.
2) **W**: Requested analysis of systematic human–model disagreements and potential biases.

   **A**: We re-check annotations, correct ground truth where needed, and categorize remaining model failures to reveal capability-driven, not biased, gaps.
3) **W**: Asked whether model or run ensembling could improve agreement with humans.

   **A**: We test model/run ensembles and find majority voting brings virtually no gains while optimistic aggregation helps only in low-FP regimes.
4) **W**: Questioned the robustness of occlusion detection, especially across modules or without headers.

   **A**: Stitched-region and full-page screenshots ensure any occlusion appears in at least one image and remains detectable.

---

### **Reviewer QJiz**
1) **W**: Argued the RAViG task is not clearly necessary or distinct from standard RAG and Deep Research agents.

   **A**: RAViG targets a distinct, latency-critical regime between RAG and Deep Research, addressing instant, visually-rich answers that current paradigms do not cover.
2) **W**: Felt the benchmark’s evaluation focus is not sufficiently novel and could be subsumed by a generic Deep Research evaluation.

   **A**: Our framework provides the first automated evaluation of functionality, design quality, chart correctness, and text–chart alignment, filling a gap for all visually-rich generation tasks (including Deep Research) while being especially sensitive to failure modes amplified in fast, HTML-style RAViG.

---

We hope this summary is helpful for your final assessment. Thank you for your time and consideration.

**Best regards,** \
**#3310 Authors**

---

### Author Response · Authors · 2025-12-02
**Summary for Area Chair: Overall Feedback and Remaining Concerns**

**Dear Area Chair and Reviewers,**

We sincerely thank the area chair and all reviewers for their valuable evaluations and constructive feedback.
We are encouraged that the reviewers recognized our contributions, which are summarized as follows:

+ Well-motivated work that introduces the first comprehensive benchmark for the valuable RAViG task. *(ZNZT, 2Hgk, 9hTA)*
+ The benchmark’s comprehensive and rigorous design, featuring a realistic dataset and innovative multi-faceted evaluation, was highly praised. *(All Reviewers)*
+ The automated evaluation is proven reliable by high human agreement, and the study offers insightful analysis and actionable directions for future work. *(ZNZT, 2Hgk, 9hTA)*

---

Beyond these recognized strengths, the review process also raised some concerns.  We are pleased that our rebuttal have **fully addressed the concerns of Reviewers ZNZT and 2Hgk**.Their initial questions on metric validity and protocol robustness were addressed by our new experiments, which include a hyperparameter sensitivity analysis (Appendix L), a human correlation study (Appendix K), a cross-family LLM-judge analysis (Appendix O), and a stochastic decoding experiment (Appendix P).

---
Below, we summarize our responses to the remaining key points:

### **Model-Dependence and the Viability of Open-Source Models (9hTA W1/Q1/Q3)**
> 9hTA: "I would like to see a more detailed analysis of how much the protocol is reliant on these specific models, or a more robust protocol that can verifiably work well with open-source models."

Our Rebuttal:
1) We clarified that our framework is principally model-agnostic, as it **relies on objective criteria rather than subjective scoring**.
2) We conducted new empirical analysis with leading close- and open-source (e.g., Qwen and InternVL) models as evaluators. The results confirm that **our protocol is not tied to a specific proprietary model; sufficiently capable open-source evaluators like Qwen3-VL-235B-A22B-Thinking and alternative close-source models also yield relatively accurate judgments**.
3) Therefore, our decision to use the best-performing models ensures the highest fidelity for our benchmark at present, aligning with standard practices in prominent works such as WildBench, VisEval, and G-Eval.

---

### **The Necessity of the RAViG Task (QJiz R1)**
> QJiz: "The existence of a "speed gap" (with Deep Research) does not automatically justify the creation of a new benchmark category. This paper needs a stronger motivation for why this specific niche—fast, visually-rich HTML generation—is a critical research problem in itself, rather than merely an engineering optimization or a sub-capability of a more general agent."

Our Rebuttal:
1) **The stringent, second-level latency constraint of RAViG fundamentally redefines the research problem**. This creates a need for a new paradigm of "Instantaneous Synthesis Intelligence," which is scientifically distinct from the iterative methods of Deep Research, thus posing a unique scientific challenge.
2) What the reviewer dismissed as a "niche" is, in fact, the core battleground for leading industry products like Google AI Overviews and Perplexity. This field currently lacks a standardized academic benchmark. Therefore, **our work is not creating an unnecessary category but establishing the first scientific foundation for a field of immense real-world importance, filling a critical academic vacuum.**

---

### **The Novelty of the Evaluation Framework (QJiz R2)**
> QJiz: "However, I argue that any competent Deep Research system or evaluation framework for a multi-modal report would also inherently care about and evaluate these aspects. The fact that prior works may have reported only overall metrics does not mean that the concept of evaluating chart-text alignment is new; it simply means that their evaluation methodology was coarser."

Our Rebuttal: **The reviewer speculates that similar frameworks exist, yet across two review rounds, none of the references they provided contain systematic, automated evaluation of visualization quality or text-chart alignment**. Our analysis of their citations, in fact, confirms that these prior works focus on textual insights, thereby highlighting the very research gap our benchmark fills. **To the best of our knowledge, our work is therefore the first to bridge this critical gap from conceptual concern ("inherently caring") to concrete, automated measurement.**

---

We thank the area chair for coordinating this discussion and the reviewers again for their insightful feedback, which prompted further empirical analyses that have strengthened our benchmark. We hope our detailed rebuttal and new experiments have addressed all concerns. As the first benchmark to establish a foundation for the critical RAViG task, we believe it offers significant value and hope it has earned your support.

**Best regards,** \
**#3310 Authors**

---

### Meta-Review · Area_Chair_wxhf · 2026-01-08

**Summary:**

There are several concerns raised by the reviewers, which indicate that the paper might fall short for the bar of ICLR.

1. Task Novelty and Necessity: Reviewer QJiz strongly questioned the novelty of the RAViG task, arguing it overlaps significantly with existing "Deep Research" and Data Insight agents. The reviewer posited that requiring HTML output or faster processing times constitutes an engineering optimization or sub-capability rather than a distinct research problem warranting a new benchmark category.

2. Evaluation Methodology and Robustness: Multiple reviewers (2Hgk, 9hTA, QJiz) scrutinized the reliability of the LLM-as-a-Judge protocol. Concerns included potential self-preference bias, the reliance on specific closed-source models (GPT-4o/Gemini), and the lack of a ground truth for visual outputs. Reviewer ZNZT also questioned the ad hoc nature of the visual richness metrics.

3. Experimental Settings: Reviewer 2Hgk noted that using a deterministic setting oversimplified the evaluation by ignoring stochastic diversity and creative range.

Reproducibility: Reviewer ZNZT raised concerns about limited reproducibility due to the non-disclosure of complete system prompts.

Usefulness for Top-Tier Models: Reviewer ZNZT worried that high baseline scores (e.g., GPT-5) might suggest the benchmark is already "solved" or lacks discriminative power for future frontier models.

**Reviewer Concerns:**

Addressed Concerns:

The authors provided a thorough rebuttal that addressed some of the concerns. The authors conducted cross-family judge consistency checks, tested open-source judge models (Qwen, InternVL, DeepSeek), and analyzed self-preference bias, demonstrating the protocol's stability (addressing 2Hgk, 9hTA). An expert annotation study was introduced, showing a strong correlation between the proposed score and human judgments. New experiments using Temperature=0.7 were provided to demonstrate the benchmark's stability under non-deterministic settings. Additional visual examples and detailed error analyses were integrated into the appendix and text to bridge the gap between quantitative metrics and qualitative insights. The authors promised to releasing all evaluation prompts, datasets, and scoring code, alongside the model-generated outputs for reproducibility.

Outstanding Concerns:

The disagreement regarding the necessity of the RAViG task remains unresolved with Reviewer QJiz. While the authors argued that RAViG addresses a distinct "Instantaneous Synthesis" challenge for everyday queries—a gap not filled by Deep Research benchmarks—Reviewer QJiz maintained that this is a difference in degree (speed/format) rather than the nature, and viewed the work as a sub-capability of general agents. Meahwhile, Reviewer ZNZT remained "not fully convinced" regarding the benchmark's long-term ability to distinguish between top-tier models given current high scores.

In sum, the authors made tangible progress to improve the paper. However, the value of the benchmark is still not fully established.

**Reviewer Scores:**

Reviewer ZNZT: This reviewer explicitly maintained their score of 6. They stated that the rebuttal dispelled some of their concerns regarding  but retained a reservation about the benchmark's future longevity for top-tier models.

Reviewer 2Hgk: This reviewer seems satisfied with the rebuttal and discussiopn and requested the new findings be incorporated into the manuscript. Yet, the reviewer didn't raise the score.

Reviewer 9hTA: This reviewer did not participate in the final discussion. The reviewer's current score is 4, and might raise it a bit given the additional results added by the rebuttal.

Reviewer QJiz: This reviewer explicitly maintained their score of 2. They engaged in the discussion but rejected the authors' arguments regarding the distinction between RAViG and Deep Research, maintaining that the paper lacks sufficient motivation for a standalone task.

---

### Decision · Program_Chairs · 2026-01-26

Reject